# A Characterization of List Regression

**Chirag Pabbaraju**                                                   CPABBARA@CS.STANFORD.EDU
*Stanford University*

**Sahasrajit Sarmasarkar**                                          SAHASRAS@STANFORD.EDU
*Stanford University*

**Editors:** Gautam Kamath and Po-Ling Loh

## Abstract

There has been a recent interest in understanding and characterizing the sample complexity of list learning tasks, where the learning algorithm is allowed to make a short list of $k$ predictions, and we simply require one of the predictions to be correct. This includes recent works characterizing the PAC sample complexity of standard list classification and online list classification.

Adding to this theme, in this work, we provide a complete characterization of list PAC *regression*. We propose two combinatorial dimensions, namely the $k$-OIG dimension and the $k$-fat-shattering dimension, and show that they characterize realizable and agnostic $k$-list regression respectively. These quantities generalize known dimensions for standard regression. Our work thus extends existing list learning characterizations from classification to regression.

**Keywords:** List Regression, PAC Learning

## 1. Introduction

The study of machine learning algorithms that output multiple predictions (or a short *list* of predictions) instead of a single prediction has recently been gaining traction (Charikar and Pabbaraju, 2023; Moran et al., 2023; Hanneke et al., 2024). There are several motivations to consider such a setting. In terms of statistical learnability, list learning is an easier task, and can help circumvent prohibitively large sample complexities that might otherwise be necessary with single predictions. Such large sample complexities can arise due to pathologies in the combinatorial structure of the learning task, or simply due to the presence of noisy labels in the training data. Additionally, list-valued predictions are arguably better-suited to many natural tasks that have a notion of "short-listing" built into them, e.g., recommender systems. When the output of the algorithm is mostly a guideline, and will likely undergo a post-processing step by a human, it seems more beneficial to present the human with a few candidate predictions so as to better inform their choice.

Motivated by these reasons, recent works have focused on characterizing the sample complexity of list prediction tasks for a given list size. These include, for example, the works of Charikar and Pabbaraju (2023) and Moran et al. (2023) which respectively extend standard classification and online classification tasks to the list learning setting. In these settings, a learning algorithm is deemed successful on a test point if the true label for that point simply *belongs* to the list of predictions output by the algorithm. Notably, such a characterization has not yet been established for the classical task of real-valued *regression*. Here, we might be willing to tolerate a short list of real-valued outputs for any given test point, so long as the true target output is *close* to at least one of the values in the output.

Consider, for instance, a scenario where the labels in the dataset correspond to measurements of a physical quantity. For many quantities, measurements can be made in different standardized

units. For example, based on geographical region, distance is measured in miles/kilometers, mass is measured in kilograms/pounds, etc. If the measurements correspond to scientific experiments, then details about the lab setup matter. For example, the measurement of the polarization of a wave with a filter depends heavily on the phase at which the filter is initialized. In such instances, if the dataset in question is collected in a crowdsourced manner from different groups of people adhering to differing conventions, there is a discrepancy in the scales of the labels. In the event that information about the different scales in the data is lost, an algorithm can only really hope to output predictions at an assortment of scales. Such list-valued predictions might nevertheless still be useful to a user.

With this context, in the present work, we derive a complete characterization of list regression in the Probably Approximately Correct (PAC) framework (Valiant, 1984). For any target list size $k$, we identify necessary and sufficient conditions under which a given real-valued hypothesis class admits list learnability with $k$ labels, for both the realizable and agnostic settings. Our conditions are phrased in terms of finiteness of novel combinatorial dimensions of the class. We note that each of these settings in standard single-valued (i.e., $k = 1$) regression is characterized by a *separate scale-sensitive* dimension. This is in stark contrast to the setting of classification, where both the realizable and agnostic cases are governed by the same dimension (Vapnik and Chervonenkis, 1974; Blumer et al., 1989; Daniely and Shalev-Shwartz, 2014; Brukhim et al., 2022). While for the realizable case, Attias et al. (2023) recently showed that the *OIG dimension* completely characterizes learnability, for the agnostic case, Alon et al. (1997) had earlier shown that the *fat-shattering dimension* is the relevant quantity. We propose natural generalizations of both these dimensions for list regression, and show that they faithfully characterize their respective learning settings.

Our work is complementary to the line of works on *list-decodable* regression (Karmalkar et al., 2019; Raghavendra and Yau, 2020). While the focus there is to output a list of $\text{poly}(1/\alpha)$ hypotheses given that only an $\alpha$ fraction of the data is clean and the rest may be arbitrarily corrupted, our focus is more on the *combinatorial characterization* of list PAC learnability of hypothesis classes, even with uncorrupted samples. Namely, while the combinatorial structure of classes at a lower order might preclude standard PAC learnability/lead to large sample complexity, we seek to identify higher-order combinatorial structure which might still make it amenable to list learning.

## 1.1. Overview of Results

For concreteness, we focus on real-valued hypothesis classes whose range is bounded in $[0, 1]$, and consider learning with respect to the absolute ($\ell_1$) loss function.[1] Our main theorems are analogous to standard scale-sensitive results from the literature on real-valued ($k = 1$) regression.

For agnostic list regression with lists of size $k$, we show that learnability is characterized by a generalization of the fat-shattering dimension, denoted as the $k$-fat-shattering dimension.

**Theorem 1 (Informal, Agnostic List Regression, see Theorems 12, 26)** *A hypothesis class $\mathcal{H}$ is amenable to agnostic $k$-list regression if and only if its $k$-fat-shattering dimension is finite at all scales.*

Similarly, for realizable list regression with lists of size $k$, we show that learnability is characterized by a generalization of the OIG dimension, denoted as the $k$-*OIG dimension*.

---

1. We believe our results should generalize to other standard loss functions without too much effort.

**Theorem 2 (Informal, Realizable List Regression, see Theorems 40, 43)** *A hypothesis class $\mathcal{H}$ is amenable to realizable $k$-list regression if and only if its $k$-OIG dimension is finite at all scales.*

We remark that our upper and lower bounds for sample complexity in both the realizable and agnostic settings are nearly matching in the asymptotic dependence on the proposed scale-sensitive dimensions, upto polylogarithmic factors.[2] This is in contrast to the existing bounds for list classification (Charikar and Pabbaraju, 2023), where there is a polynomial gap between the upper and lower bounds on sample complexity. While we have not attempted to optimize the dependence of the list size in our bounds (which we can typically assume to be a small constant), this is a natural direction for future study.

Our learning algorithms for both the realizable and agnostic settings follow a common template. First, we utilize finiteness of the relevant dimension in order to construct an initial *weak* learner. Thereafter, using the technique of "minimax-and-sample" (Moran and Yehudayoff, 2016), we boost the weak learner to obtain a procedure that only uses a small (sublinear) fraction of the training data, but labels the entire training data accurately. Such procedures constitute *sample compression schemes*, and are known to generalize beyond the training data. Our lower bounds, on the other hand, follow the textbook template of constructing "maximally hard" learning problems from *shattered* sets of maximal size.

Nevertheless, the existing analyses for standard regression don't immediately generalize to list regression, and we have to rely on different techniques and other novel tools for our analyses. For example, the learning algorithm that works in the standard agnostic case is simply Empirical Risk Minimzation (ERM) (Alon et al., 1997). However, in our setting, it is a priori not clear as to what list hypothesis class one might run ERM on, or how one might consolidate different ERMs from the given hypothesis class. Therefore, inspired from the works of Bartlett and Long (1998); Daskalakis and Golowich (2024), we have to rely on a different recipe that involves constructing a learner for a suitably defined *multiclass partial* hypothesis class (Alon et al., 2022), and thereafter carefully consolidating list-valued predictions for the constructed class. Similarly, for the agnostic lower bound, we are required to show novel higher-order packing number bounds that significantly generalize existing results, and might be of independent interest. For the realizable case, the learning algorithm of Attias et al. (2023) involves the median boosting algorithm (Simon, 1997), which requires ordering predictions given by different weak learners. While real-valued predictions naturally admit an ordering, it is less clear how one would order lists of predictions in order to do the required boosting updates. Hence, we are required to alternatively use the paradigm of minimax boosting mentioned above, together with a different scheme of aggregating weak list-valued predictions. Lastly, for both the realizable and agnostic lower bounds, we require using a simple yet elegant combinatorial claim (Claim 48), which lower bounds the size of the union of (mutually disjoint) sets, given that each of the sets is large.

In the rest of the paper, we incrementally set up the stage and derive all our results in long-form, systematically elaborating on all the points mentioned above and more.

---

2. We note however, that the scales in the dimensions in the upper and lower bounds differ—by a constant factor in the agnostic setting, and by a quadratic factor in the realizable setting.

## 2. Preliminaries and Notation

We denote the data domain by $\mathcal{X}$ and label space by $\mathcal{Y}$. A hypothesis class $\mathcal{H}$ then is simply a subset of $\mathcal{Y}^{\mathcal{X}}$. The focus of this paper is real-valued list regression with respect to the absolute loss function on a bounded interval—thus, in most places, $\mathcal{Y}$ will be the interval $[0,1]$.

Because we care about measuring the loss of a target label against the *best* label from a candidate *list* of $k \geq 1$ labels, the loss function $\ell_{\mathrm{abs}} : \mathcal{Y}^{\leq k} \times \mathcal{Y} \to [0,1]$ between a $k$-list $\mu \in \mathcal{Y}^{\leq k}$ and a target label $y \in \mathcal{Y}$ will be measured as

$$\ell_{\mathrm{abs}}(\mu, y) = \min_{j \in [k]} |\mu_j - y|,$$

where $\mu_j$ denotes the $j^{\text{th}}$ entry in $\mu$, and $[k]$ denotes the set $\{1, 2, \ldots, k\}$. We also use the notation $\mu \ni_\gamma y$, read "$\mu$ $\gamma$-contains $y$", to denote that $\ell_{\mathrm{abs}}(\mu, y) \leq \gamma$, where $\gamma \in (0,1)$. Similarly, we use the terms "$\gamma$-close" and "$\gamma$-far" to denote $\ell_{\mathrm{abs}}(\mu, y) \leq y$ and $\ell_{\mathrm{abs}}(\mu, y) > \gamma$ respectively.

Given a distribution $\mathcal{D}$ on $\mathcal{X} \times \mathcal{Y}$, the loss of a list hypothesis $\mu : \mathcal{X} \to \mathcal{Y}^{\leq k}$ with respect to $\mathcal{D}$ is defined as:

$$\mathrm{err}_{\mathcal{D}, \ell_{\mathrm{abs}}}(\mu) = \mathbb{E}_{(x,y) \sim \mathcal{D}}\, \ell_{\mathrm{abs}}(\mu(x), y).$$

Given a sample $S = \{(x_i, y_i)\}_{i \in [n]} \in \{\mathcal{X} \times \mathcal{Y}\}^n$, the empirical loss of a list hypothesis $\mu$ with respect to $S$ is defined as

$$\hat{\mathrm{err}}_{S, \ell_{\mathrm{abs}}}(\mu) = \frac{1}{n} \sum_{i=1}^{n} \ell_{\mathrm{abs}}(\mu(x_i), y_i).$$

A training sample $S = \{(x_i, y_i)\}_{i \in [n]} \in \{\mathcal{X} \times \mathcal{Y}\}^n$ is realizable by $\mathcal{H}$ if $\exists h^\star \in \mathcal{H}$ such that $h(x_i) = y_i$ for every $i \in [n]$. A distribution $\mathcal{D}$ over $\mathcal{X} \times \mathcal{Y}$ is realizable by $\mathcal{H}$, if there exists $h^\star \in \mathcal{H}$, such that a draw from $\mathcal{D}$ corresponds to drawing $x$ from a marginal distribution $\mathcal{D}_{\mathcal{X}}$ over $\mathcal{X}$, and then setting $y = h^\star(x)$. Finally, we use the notation $\mathcal{H}|_T$ to refer to the restriction of $\mathcal{H}$ to an unlabeled sequence $T \in \mathcal{X}^n$.

### 2.1. PAC Learnability

We formally define the notion of agnostic list regression in the PAC learning framework.

**Definition 3 (Agnostic List Regression)** *For $k \geq 1$, a hypothesis class $\mathcal{H} \subseteq [0,1]^{\mathcal{X}}$ is said to be agnostically $k$-list learnable if there exists a learning algorithm $\mathcal{A}$ and a sample complexity function $m_{\mathcal{A}, \mathcal{H}}^{k, \mathrm{ag}} : (0,1)^2 \to \mathbb{N}$ satisfying the following guarantee: for any $\varepsilon, \delta \in (0,1)$ and any distribution $\mathcal{D}$ over $\mathcal{X} \times [0,1]$, if the algorithm $\mathcal{A}$ is provided with $m \geq m_{\mathcal{A}, \mathcal{H}}^{k, \mathrm{ag}}(\varepsilon, \delta)$ i.i.d. examples drawn from $\mathcal{D}$ (denoted as $S$), then $\mathcal{A}$ produces a $k$-list hypothesis $\mathcal{A}(S)$ such that, with probability at least $1 - \delta$ over the draw of $S$ and the randomness in $\mathcal{A}$,*

$$\mathrm{err}_{\mathcal{D}, \ell_{abs}}(\mathcal{A}(S)) \leq \inf_{h \in \mathcal{H}} \mathrm{err}_{\mathcal{D}, \ell_{abs}}(h) + \varepsilon.$$

Note that the case of $k = 1$ corresponds to the standard definition of agnostic regression (Haussler, 1992; Alon et al., 1997; Bartlett et al., 1994). In this case, agnostic learnability is characterized by a scale-sensitive quantity known as the *fat-shattering* dimension (Alon et al., 1997). We formally

define this quantity, whose finiteness (at all scales) is both sufficient and necessary for agnostic learnability of a hypothesis class. Furthermore, for classes having finite fat-shattering dimension, any ERM is a successful agnostic list regression algorithm.

**Definition 4 ($\gamma$-fat-shattering dimension (Alon et al., 1997))** *For a hypothesis class $\mathcal{H} \subseteq [0,1]^{\mathcal{X}}$ and $\gamma \in (0,1)$, the $\gamma$-fat-shattering dimension $\mathbb{D}_{\gamma}^{\mathrm{fat}}(\mathcal{H})$ is the largest positive integer $d$ such that there exist $x_1, x_2 \ldots, x_d \in \mathcal{X}$ and $s_1, s_2 \ldots s_d \in [0,1]$, such that for all $b \in \{0,1\}^d$, there is some $h_b \in \mathcal{H}$ satisfying $h_b(x_i) \geq s_i + \gamma$ if $b_i = 1$ and $h_b(x_i) \leq s_i - \gamma$ if $b_i = 0$.*

We next formally define the notion of realizable list regression, which only considers learning problems with respect to distributions realizable by the hypothesis class.

**Definition 5 (Realizable List Regression)** *For $k \geq 1$, a hypothesis class $\mathcal{H} \subseteq [0,1]^{\mathcal{X}}$ is said to be (realizably)[3] $k$-list learnable if there exists a learning algorithm $\mathcal{A}$ and a sample complexity function $m_{\mathcal{A},\mathcal{H}}^{k,\mathrm{re}} : (0,1)^2 \to \mathbb{N}$ satisfying the following guarantee: for any $\varepsilon, \delta \in (0,1)$ and any distribution $\mathcal{D}$ over $\mathcal{X} \times [0,1]$ realizable by $\mathcal{H}$, if the algorithm $\mathcal{A}$ is provided with $m \geq m_{\mathcal{A},\mathcal{H}}^{k,\mathrm{re}}(\varepsilon,\delta)$ i.i.d. examples drawn from $\mathcal{D}$ (denoted as $S$), then $\mathcal{A}$ produces a $k$-list hypothesis $\mathcal{A}(S)$ such that, with probability at least $1 - \delta$ over the draw of $S$ and the randomness in $\mathcal{A}$,*

$$\mathrm{err}_{\mathcal{D},\ell_{abs}}(\mathcal{A}(S)) \leq \varepsilon.$$

In the standard case of $k = 1$, the aforementioned fat-shattering dimension (Definition 4) *does not* characterize learnability of a hypothesis class under the realizability assumption (Bartlett et al., 1994, Section 6). Instead, the dimension that characterizes realizable regression was identified by Attias et al. (2023) to be a combinatorial quantity stated in terms of the *One-Inclusion Graph (OIG)* of the class.

**Definition 6 (One-inclusion graph (Haussler et al., 1994; Rubinstein et al., 2009))** *Consider the set $[n]$ and a hypothesis class $\mathcal{H} \subseteq [0,1]^{[n]}$. We define a hypergraph $\mathcal{G}(\mathcal{H}) = (V,E)$ such that $V = \mathcal{H}$. Consider any direction $i \in [n]$ and any mapping $f : [n] \setminus \{i\} \to [0,1]$: together, these induce the hyperedge $e_{i,f} = \{h \in \mathcal{H} : h(j) = f(j), \forall j \in [n] \setminus \{i\}\}$. The edge set $E$ is then defined to be the collection*

$$E = \{e_{i,f} : i \in [n], f : [n] \setminus \{i\} \to [0,1], e_{i,f} \neq \emptyset\}.$$

When every hyperedge in $\mathcal{G}(\mathcal{H})$ is mapped to one of its constituent vertices, we obtain an *orientation* of the hypergraph. Attias et al. (2023) introduced the notion of "scaled outdegree" for such orientations.

**Definition 7 (Orientation and scaled outdegree (Attias et al., 2023))** *An orientation of the one-inclusion graph $\mathcal{G}(\mathcal{H})$ of a hypothesis class $\mathcal{H} \subseteq [0,1]^{\mathcal{X}}$ is a mapping $\sigma : E \to V$ such that $\sigma(e) \in E$ for every $e \in E$. We denote by $\sigma_i(e) \in [0,1]$ the $i^{th}$ entry of the vertex that the edge $e$ is oriented to.*

---

3. We will be explicit about using the term "agnostic" whenever talking about agnostic learnability; otherwise, unless specified, learnability will usually refer to *realizable* learnability.

*For a vertex $v \in V$, which corresponds to some hypothesis $h \in \mathcal{H}$, let $v_i$ be the $i^{th}$ entry of $v$ (which corresponds to $h(i)$), and let $e_{i,v}$ be the hyperedge adjacent to $v$ in the direction $i$. For $\gamma \in (0,1)$, the (scaled) outdegree of vertex $v$ under $\sigma$ is*

$$\mathsf{outdeg}(v; \sigma, \gamma) = |\{i \in [n] : \ell_{abs}(\sigma_i(e_{i,v}), v_i) > \gamma\}|.$$

*The maximum scaled outdegree of $\sigma$ is denoted by $\mathsf{outdeg}(\sigma, \gamma) := \max\limits_{v \in V} \mathsf{outdeg}(v; \sigma, \gamma)$.*

Based on this notion of scaled outdegree, Attias et al. (2023) defined the $\gamma$-OIG dimension of a hypothesis class, and showed that its finiteness is both necessary and sufficient for realizable learnability of a hypothesis class.

**Definition 8 (OIG dimension (Attias et al., 2023))** *Consider a hypothesis class $\mathcal{H} \subseteq [0,1]^{\mathcal{X}}$ and let $\gamma \in (0,1)$. The $\gamma$-OIG dimension of $\mathcal{H}$ is defined as follows:*

$$\mathbb{D}_{\gamma}^{\mathrm{OIG}}(\mathcal{H}) := \sup \{n \in \mathbb{N} : \exists S \in \mathcal{X}^n \text{ such that } \exists \text{ finite subgraph } G = (V, E) \text{ of } \mathcal{G}(\mathcal{H}|_S) \text{ such that}$$

$$\forall \text{ orientations } \sigma, \exists v \in V \text{ such that } \mathsf{outdeg}(v; \sigma, \gamma) > \frac{n}{3}\}. \tag{1}$$

It is worth mentioning that the OIG dimension is linked to the dual definition of "DS dimension" of a hypothesis class—informally, if a hypothesis class DS-shatters a sequence $S$ of size $d$, then for any possible orientation of the projection of the class on $S$, there is a vertex that ends up having large outdegree (Brukhim et al., 2022, Lemma 12).

### 2.2. Sample Compression

We now define the notion of *sample compression schemes* (Littlestone and Warmuth, 1986), which are a useful tool in deriving learning algorithms.

**Definition 9 (List Sample Compression)** *For any domain $\mathcal{X}$ and label set $\mathcal{Y}$, a $k$-list sample compression scheme for the tuple $(\mathcal{X}, \mathcal{Y})$ is a pair $(\kappa, \rho)$, comprising of a compression function $\kappa : (\mathcal{X} \times \mathcal{Y})^* \rightarrow (\mathcal{X} \times \mathcal{Y})^* \times \{0,1\}^*$ and a reconstruction function $\rho : (\mathcal{X} \times \mathcal{Y})^* \times \{0,1\}^* \rightarrow (\mathcal{Y}^k)^{\mathcal{X}}$, satisfying the following property: for any sequence $S \in (\mathcal{X} \times \mathcal{Y})^*$, $\kappa(S)$ evaluates to some tuple $(S', B) \in (\mathcal{X} \times \mathcal{Y})^* \times \{0,1\}^*$, where $S'$ is a sequence of elements of $S$, and the bit string $B$ is some "side information". Given $S = \{(x_i, y_i)\}_{i \in [n]} \in (\mathcal{X} \times \mathcal{Y})^n$, let $(S', B) := \kappa(S)$, where $S' = \{(x_{i_1}, y_{i_1}), \dots, (x_{i_r}, y_{i_r})\}$ for $(i_1, \dots, i_r) \in [n]^r$. Furthermore, denote $S \setminus \kappa(S) := \{(x_j, y_j) \in S : j \notin (i_1, \dots, i_r)\}$. Define $|\kappa(S)| := |S'| + |B|$ to be the sum of the size of $S'$ and the length of $B$. The size of the compression scheme $(\kappa, \rho)$ for $n$-sample datasets is $|\kappa(n)| := \max_{S \in (\mathcal{X} \times \mathcal{Y})^{\leq n}} |\kappa(S)|$.*

In particular, we will make extensive use of the following lemma, which is well-established by this point (e.g., David et al. (2016); Daskalakis and Golowich (2024)), and characterizes the generalization properties of sample compression schemes. A straightforward proof can be obtained, for example, by observing that all the calculations in Sections B.1 and B.2 in Charikar and Pabbaraju (2023) hold true for any loss function $\ell : \mathcal{Y}^k \times \mathcal{Y} \rightarrow [0,1]$.

**Lemma 10 (Essentially Theorem 30.2 in Shalev-Shwartz and Ben-David (2014))** *Consider a domain $\mathcal{X}$ and label set $\mathcal{Y}$, together with a loss function $\ell : \mathcal{Y}^k \times \mathcal{Y} \to [0,1]$. Let $(\kappa, \rho)$ be any sample compression scheme satisfying $|\kappa(n)| \leq n/2$ for all large enough $n$. For any distribution $\mathcal{D}$ over $\mathcal{X} \times [0,1]$, and any $\delta \in (0,1)$, the following holds with probability at least $1 - \delta$ over $S \sim \mathcal{D}^n$:*

$$
\mathrm{err}_{\mathcal{D},\ell}(\rho(\kappa(S))) \leq \hat{\mathrm{err}}_{S,\ell}(\rho(\kappa(S))) + O\left(\sqrt{\hat{\mathrm{err}}_{S\setminus\kappa(S),\ell}(\rho(\kappa(S)))\left(\frac{|\kappa(n)|\log(n) + \log(1/\delta)}{n}\right)}\right)
$$
$$
+ O\left(\frac{|\kappa(n)|\log(n) + \log(1/\delta)}{n}\right).
$$

## 3. Agnostic List Regression

At a high level, the $\gamma$-fat-shattering dimension (Definition 4), which characterizes standard agnostic regression (i.e., $k = 1$), accounts for the possibility of a function being an offset (namely $\gamma$) away on either side of designated anchors on a sequence of points: no matter what label one predicts, the label on the other side of the anchor is going to cause an error (in absolute value) of at least $\Omega(\gamma)$. However, this issue seems to go away if we are allowed to predict two labels that are separated by $\Omega(\gamma)$. Motivated by this, we propose the following definition of $(\gamma, k)$-*fat-shattering* which naturally generalizes $\gamma$-fat-shattering to account for $k \geq 1$ possible predictions, and recovers the $\gamma$-fat-shattering dimension when $k = 1$.

**Definition 11 ($(\gamma, k)$-fat-shattering dimension)** *For $\gamma \in (0, 1)$, a hypothesis class $\mathcal{H} \subseteq [0,1]^{\mathcal{X}}$ $(\gamma, k)$-fat-shatters a sequence $S = (x_1, \ldots, x_d) \in \mathcal{X}^d$ if there exist vectors $c_1, \ldots, c_d \in [0,1]^k$ satisfying* [4]

$$
c_{i,j+1} \geq c_{i,j} + 2\gamma, \forall j \in \{1, \ldots, k-1\}, \forall i \in \{1, \ldots, d\},
$$

*such that $\forall b \in \{0, 1, \ldots, k\}^d$, there exists $h_b \in \mathcal{H}$ satisfying*

- *$h_b(x_i) \leq c_{i,1} - \gamma$ if $b_i = 0$,*

- *$c_{i,b_i} + \gamma \leq h_b(x_i) \leq c_{i,b_i+1} - \gamma$ if $b_i \notin \{0, k\}$,*

- *$c_{i,k} + \gamma \leq h_b(x_i)$ if $b_i = k$.*

*The $(\gamma, k)$-fat-shattering dimension $\mathbb{D}^{\mathrm{fat}}_{\gamma,k}(\mathcal{H})$ is defined to be the size of the largest $(\gamma, k)$-fat-shattered sequence.*

Observe that $\mathbb{D}^{\mathrm{fat}}_{\gamma,k_1}(\mathcal{H}) \geq \mathbb{D}^{\mathrm{fat}}_{\gamma,k_2}(\mathcal{H})$ for $k_1 < k_2$; however, the gap can be unbounded. The following (essentially Example 3 in Charikar and Pabbaraju (2023)) is a simple example of a hypothesis class that has finite $(\gamma, 2)$-fat shattering dimension, but infinite $\gamma$-fat-shattering dimension for every $\gamma \leq 0.05$.

---

4. For $j \in [k]$, $c_{i,j}$ denotes the $j^{\text{th}}$ coordinate of $c_i$.

**Example 1** *Consider $\mathcal{H} \subseteq [0,1]^{\mathbb{N}}$ defined as:*

| $\mathcal{X}$ | $=$ | 1 | 2 | 3 | 4 | 5 | 6 | 7 | 8 | 9 | $\ldots$ |
|---|---|---|---|---|---|---|---|---|---|---|---|

$$\mathcal{H} \quad = \quad \begin{Bmatrix} 0.1 \\ 0.2 \\ 0.3 \end{Bmatrix} \times \begin{Bmatrix} 0.1 \\ 0.2 \\ 0.3 \end{Bmatrix} \times \begin{Bmatrix} 0.1 \\ 0.2 \\ 0.3 \end{Bmatrix} \times \begin{Bmatrix} 0.1 \\ 0.2 \end{Bmatrix} \times \begin{Bmatrix} 0.1 \\ 0.2 \end{Bmatrix} \times \begin{Bmatrix} 0.1 \\ 0.2 \end{Bmatrix} \times \begin{Bmatrix} 0.1 \\ 0.2 \end{Bmatrix} \times \begin{Bmatrix} 0.1 \\ 0.2 \end{Bmatrix} \times \begin{Bmatrix} 0.1 \\ 0.2 \end{Bmatrix} \times \cdots$$

*For any set of points $\{x_i\}_{i \in [n]} \in \mathcal{X}^n$, $\mathcal{H}$ projected onto this set contains all possible patterns in $\{0.1, 0.2\}^n$, and hence the $\gamma$-fat-shattering dimension of $\mathcal{H}$ is infinite for any $\gamma \leq 0.05$. However, the $(\gamma, 2)$-fat-shattering dimension of $\mathcal{H}$ is just 3. We can see that $\mathcal{H}$ $(\gamma, 2)$-fat-shatters the sequence $(1, 2, 3)$ (e.g., with respect to $c_{i,1} = 0.15$ and $c_{i,2} = 0.25$ for $i = 1, 2, 3$). However, any sequence of points that contains an $x$ in $\{4, 5, 6, \ldots\}$ cannot be $(\gamma, 2)$-fat-shattered by $\mathcal{H}$, because there do not exist three functions $h_1$, $h_2$ and $h_3$ that are all distinct at such an $x$.*

Thus, even if the hypothesis class in Example 1 is seemingly "simple", it is not agnostically learnable because of infinite $\gamma$-fat-shattering dimension. We now proceed to state our main theorems characterizing agnostic list learnability of hypothesis classes, which ensure that classes like the one above are agnostically list learnable on account of having finite $(\gamma, k)$-fat-shattering dimension, and also establish that finiteness of this quantity is in fact necessary for agnostic list learnability.

### 3.1. Upper Bound for Agnostic List Regression

**Theorem 12 (Finite $(\gamma, k)$-fat-shattering dimension is sufficient)** *Let $\mathcal{H} \subseteq [0,1]^{\mathcal{X}}$ be a hypothesis class. For any distribution $\mathcal{D}$ over $\mathcal{X} \times [0,1]$ and $\gamma \in (0,1)$, there exists a list learning algorithm $\mathcal{A}$ which takes as input an i.i.d. sample $S \sim \mathcal{D}^n$ and outputs a $k$-list hypothesis $\mathcal{A}(S)$ satisfying*

$$\mathrm{err}_{\mathcal{D}, \ell_{abs}}(\mathcal{A}(S)) \leq \inf_{h \in \mathcal{H}} \mathrm{err}_{\mathcal{D}, \ell_{abs}}(h) + O(k) \cdot \gamma + \widetilde{O}\left( \sqrt{\frac{k^8 \, \mathbb{D}_{\gamma/2, k}^{\mathrm{fat}}(\mathcal{H}) + \log(1/\delta)}{n}} \right)$$

*with probability at least $1 - \delta$.[5] In other words, $m_{\mathcal{A}, \mathcal{H}}^{k, \mathrm{ag}}(\varepsilon, \delta) \leq \widetilde{O}\left( \frac{k^8 \, \mathbb{D}_{\gamma/2, k}^{\mathrm{fat}}(\mathcal{H}) + \log(1/\delta)}{\varepsilon^2} \right)$ for $\gamma = \frac{\varepsilon}{O(k)}$.*

As a corollary of our analysis, we also obtain the following tighter guarantee for realizable list regression.

**Corollary 13 (Realizable list regression $(\gamma, k)$-fat-shattering upper bound)** *Let $\mathcal{H} \subseteq [0,1]^{\mathcal{X}}$ be a hypothesis class. For any distribution $\mathcal{D}$ over $\mathcal{X} \times [0,1]$ realizable by $\mathcal{H}$ and $\gamma \in (0,1)$, there exists a list learning algorithm $\mathcal{A}$ which takes as input an i.i.d. sample $S \sim \mathcal{D}^n$ and outputs a $k$-list hypothesis $\mathcal{A}(S)$ satisfying*

$$\mathrm{err}_{\mathcal{D}, \ell_{abs}}(\mathcal{A}(S)) \leq O(k) \cdot \gamma + \widetilde{O}\left( \frac{k^8 \, \mathbb{D}_{\gamma/2, k}^{\mathrm{fat}}(\mathcal{H}) + \log(1/\delta)}{n} \right)$$

---

5. Here, $\widetilde{O}$ hides polylog factors in $n, k$ and $1/\gamma$.

*with probability at least $1 - \delta$. In other words, $m^{k,\mathrm{re}}_{\mathcal{A},\mathcal{H}}(\varepsilon, \delta) \leq \widetilde{O}\left(\frac{k^8 \; \mathbb{D}^{\mathrm{fat}}_{\gamma/2,k}(\mathcal{H}) + \log(1/\delta)}{\varepsilon}\right)$ for $\gamma = \frac{\varepsilon}{O(k)}$.*

We divide the proof of Theorem 12 into four sections. First, in Section 3.1.1, we construct a *discretized* version of the hypothesis class $\mathcal{H}$ with discretization width $\gamma$. The discretized class is a *multiclass partial* hypothesis class (Alon et al., 2022), whose relevant complexity parameter, namely the $k$-Natarajan dimension (Charikar and Pabbaraju, 2023), is upper-bounded by the $(\gamma/2, k)$-fat-shattering dimension of $\mathcal{H}$. This allows us to obtain a weak learner for the partial class in Section 3.1.2. The next crucial step is to go back from the discretized space to the original space—in Section 3.1.3, we show that by carefully scaling and aggregating the predictions of the weak learner, we are able to obtain accurate, list-valued predictions on the training data in the *original* space. We finally argue in Section 3.1.4 that this entire procedure really only operates on a small fraction of the training data, and hence can be viewed as a sample compression scheme. The upper bound then follows from the generalization properties of compression schemes. The whole procedure is summarized in Algorithm 1.

We make one final remark: while the agnostic learner for standard ($k = 1$) regression is based on ERM, it is not immediately clear as to how one would obtain an ERM-based learner for list regression. In particular, what is a list hypothesis class that we should run ERM on? Alternatively, do we need to collect the predictions of $k$ different risk minimizers from the given class? We discuss more on this later in Section 5. For lack of a straightforward ERM construction, we describe our different, non-ERM agnostic list learner ahead.

### 3.1.1. Construction of a Partial Hypothesis Class $\mathscr{T}_{\gamma,k}(\mathcal{H})$

Given $\mathcal{H} \subseteq [0,1]^{\mathcal{X}}$, we construct a partial hypothesis class $\mathscr{T}_{\gamma,k}(\mathcal{H})$ from it, whose label space is $\{0, 1, \ldots, k, \star\}$. If a hypothesis in a partial hypothesis class labels some $x$ with $\star$, we interpret it as "$h$ is undefined at $x$". We point the reader to the works of Bartlett and Long (1998); Alon et al. (2022) which develop the theory of partial hypothesis classes. A similar construction of a partial hypothesis class from a real-valued class was also originally considered in the works of Bartlett and Long (1998); Long (2001), and more recently in the works of Aden-Ali et al. (2023); Daskalakis and Golowich (2024). While the partial concept classes constructed in these works operate with real-valued thresholds on a binary label space $\{0, 1, \star\}$, a key difference in our construction is that we require *vector-valued thresholds* and operate on the multiclass label space $\{0, 1, \ldots, k, \star\}$.

**Definition 14 ($k$-threshold hypothesis class)** *Consider a hypothesis class $\mathcal{H} \subseteq [0,1]^{\mathcal{X}}$ and $\gamma \in (0,1)$. We define the threshold set $\mathcal{D}_\gamma := \{0, \gamma, 2\gamma, \ldots, \gamma \lfloor 1/\gamma \rfloor\}$, and the $k$-threshold set $\mathcal{D}^{(k)}_\gamma := \{\tau \in \mathcal{D}^k_\gamma : \tau_1 < \tau_2 < \ldots < \tau_k\}$. An element $\tau \in \mathcal{D}^{(k)}_\gamma$ is called a $k$-threshold. For any $\tau \in \mathcal{D}^{(k)}_\gamma$, define the threshold operator $\mathrm{Thr}_\tau : [0,1] \to \{0, 1 \ldots, k, \star\}$ as*

$$
\mathrm{Thr}_\tau(y) := \begin{cases} 0 : & y \leq \tau_1 - \frac{\gamma}{2}, \\ i : & \tau_i + \frac{\gamma}{2} \leq y \leq \tau_{i+1} - \frac{\gamma}{2} \text{ for some } i \in \{1, \ldots, k-1\}, \\ k : & \tau_k + \frac{\gamma}{2} \leq y, \\ \star & \text{otherwise}. \end{cases} \tag{2}
$$

Then, for every $h \in \mathcal{H}$ and $\tau \in \mathcal{D}_\gamma^{(k)}$, we define $h^{\mathrm{thr}} : \mathcal{X} \times \mathcal{D}_\gamma^{(k)} \to \{0, 1, \ldots, k, \star\}$ to be the partial hypothesis that maps $(x, \tau) \mapsto \mathrm{Thr}_\tau(h(x))$. We then define $\mathscr{T}_{\gamma,k}(\mathcal{H}) \subseteq \{0, 1, \ldots, k, \star\}^{\mathcal{X} \times \mathcal{D}_\gamma^{(k)}}$, the $k$-threshold hypothesis class of $\mathcal{H}$, to be the partial hypothesis class

$$\mathscr{T}_{\gamma,k}(\mathcal{H}) := \{h^{\mathrm{thr}} : h \in \mathcal{H}\}.$$

We can bound the complexity of the $k$-threshold class $\mathscr{T}_{\gamma,k}(\mathcal{H})$ in terms of the complexity of $\mathcal{H}$. For this, we recall the definition of the $k$-Natarajan dimension (Daniely et al., 2015; Charikar and Pabbaraju, 2023), and then relate the $k$-Natarajan dimension of $\mathscr{T}_{\gamma,k}(\mathcal{H})$ to the $(\gamma/2, k)$-fat-shattering dimension of $\mathcal{H}$.

**Definition 15 ($k$-Natarajan dimension $\mathbb{D}_k^{\mathrm{Nat}}$)** *A hypothesis class $\mathcal{H} \subseteq \mathcal{Y}^\mathcal{X}$ $k$-Natarajan shatters a sequence $S \in \mathcal{X}^d$ if there exist $(k+1)$-lists $y_i \in \{Y \subseteq \mathcal{Y} \setminus \{\star\} : |Y| = k+1\}$, $i = 1, \ldots, d$ such that $\mathcal{H}|_S \supseteq \prod_{i=1}^d y_i$. The $k$-Natarajan dimension of $\mathcal{H}$, denoted as $\mathbb{D}_k^{\mathrm{Nat}}(\mathcal{H})$, is the largest integer $d$ such that $\mathcal{H}$ $k$-Natarajan shatters some sequence $S \in \mathcal{X}^d$.*

**Lemma 16 (Relating $k$-Natarajan to $(\gamma/2, k)$-fat-shattering)** *Consider any $\mathcal{H} \in [0,1]^\mathcal{X}$ and $\gamma \in (0,1)$. We have that $\mathbb{D}_k^{\mathrm{Nat}}(\mathscr{T}_{\gamma,k}(\mathcal{H})) \leq \mathbb{D}_{\gamma/2,k}^{\mathrm{fat}}(\mathcal{H})$.*

**Proof** Let $d = \mathbb{D}_k^{\mathrm{Nat}}(\mathscr{T}_{\gamma,k}(\mathcal{H}))$, and consider any sequence $(x_1, \tau_1), \ldots, (x_d, \tau_d)$ that is $k$-Natarajan shattered by $\mathscr{T}_{\gamma,k}(\mathcal{H})$, where each $x_i \in \mathcal{X}, \tau_i \in \mathcal{D}_\gamma^{(k)}$. By definition of $\mathscr{T}_{\gamma,k}(\mathcal{H})$ (Definition 14), this means that for each $b \in \{0, 1, \ldots, k\}^d$, there exists $h_b \in \mathcal{H}$ satisfying the conditions for $(\gamma/2, k)$-fat-shattering from Definition 11 as witnessed by the vectors $\tau_1, \ldots, \tau_d$. That is, $\mathcal{H}$ $(\gamma/2, k)$-fat-shatters $x_1, \ldots, x_d$, and hence $d \leq \mathbb{D}_{\gamma/2,k}^{\mathrm{fat}}(\mathcal{H})$. ∎

### 3.1.2. Construction of a Weak Learner for the $k$-Threshold Class $\mathscr{T}_{\gamma,k}(\mathcal{H})$

The bounded $k$-Natarajan dimension of $\mathscr{T}_{\gamma,k}(\mathcal{H})$ allows us to obtain a *weak* list learning algorithm for the class. In particular, we can utilize a standard one-inclusion graph-based list learning algorithm for partial classes having bounded $k$-Natarajan dimension. This is given by the following guarantee:

**Lemma 17 (Weak list learner)** *Consider a partial concept class $\mathcal{H}^{\mathrm{part}} \subseteq \{0, 1, \ldots, k, \star\}^{\mathcal{X}'}$ having $\mathbb{D}_k^{\mathrm{Nat}}(\mathcal{H}^{\mathrm{part}}) < \infty$. There exists a weak list learner $\mathcal{A}_w$ satisfying the following guarantee: for any distribution $\mathcal{D}$ realizable by $\mathcal{H}^{\mathrm{part}}$, for $m \geq 960k^5 \ln(k+1)\mathbb{D}_k^{\mathrm{Nat}}(\mathcal{H}^{\mathrm{part}})$,*

$$\mathrm{Pr}_{S \sim \mathcal{D}^m} \mathrm{Pr}_{(x', y') \sim \mathcal{D}} \left[ \mu_S^k(x') \not\ni y' \right] < \frac{1}{2(k+1)},$$

*where $\mu_S^k = \mathcal{A}_w(S)$ is the $k$-list hypothesis output by $\mathcal{A}_w$.*

The proof of Lemma 17 is given in Section A.1. At a glance, the weak learner $\mathcal{A}_w$ is based on the one-inclusion graph algorithm (Haussler et al., 1994; Rubinstein et al., 2009), which constructs a "small-outdegree" orientation of $\mathcal{H}^{\mathrm{part}}|_S$ to make its predictions—the latter exists by virtue of finite $k$-Natarajan dimension.

---

6. Top-$k(L_1, \ldots, L_l)$ returns the $k$ most frequently occurring labels among the lists $L_1, \ldots, L_l$.

---

**Algorithm 1** Realizable and Agnostic List Regression Algorithms for finite $(\gamma, k)$-fat-shattering classes

---

**Input:** Hypothesis class $\mathcal{H} \subseteq [0,1]^{\mathcal{X}}$, sample $S = \{(x_i, y_i)\}_{i \in [n]} \subset (\mathcal{X} \times [0,1])^n$, discretization parameter $\gamma \in (0,1)$, weak learner $\mathcal{A}_w$ for the partial hypothesis class $\mathscr{T}_{\gamma,k}(\mathcal{H})$

1: **function** REGREALIZABLE$(S, \mathcal{A}_w, \gamma)$
2:      For every $i \in [n]$ and $\tau \in \mathcal{D}_\gamma^{(k)}$, let $y'_{i,\tau} = \mathrm{Thr}_\tau(y_i)$.
3:      Define a dataset $\tilde{S} \subseteq (\mathcal{X} \times \mathcal{D}_\gamma^{(k)} \times \{0, 1, \ldots, k\})^{n'}$ for $n' \leq n \cdot |\mathcal{D}_\gamma^{(k)}|$, as follows:

$$\tilde{S} = \left\{ ((x_i, \tau), y'_{i,\tau}) \, : \, i \in [n], \tau \in \mathcal{D}_\gamma^{(k)}, y'_{i,\tau} \neq \star \right\}.$$

4:      Obtain $l = 6(k+1)\ln(2n')$ subsequences $T_1, \ldots, T_l$ of $\tilde{S}$ each of size $m = 960k^5 \ln(k+1)\mathbb{D}_{\gamma/2,k}^{\mathrm{fat}}(\mathcal{H})$, such that when $\mathcal{A}_w$ is invoked on each of them to yield $\mu_1^k = \mathcal{A}_w(T_1), \ldots, \mu_l^k = \mathcal{A}_w(T_l)$, the list hypothesis $J : \mathcal{X} \times \mathcal{D}_\gamma^{(k)} \to \{0, 1, \ldots, k\}^k$ that maps

$$x' \mapsto \mathsf{Top}\text{-}k(\mu_1^k(x'), \ldots, \mu_l^k(x')),[6]$$

     satisfies $J(x') \ni y'$ simultaneously for every $(x', y') \in \tilde{S}$ (such subsequences exist by Lemma 18).
5:      **return** $H : \mathcal{X} \to [0,1]^k$ which maps $x \mapsto$ MERGELISTS$(J(x,.), r = 6k\gamma + 3\gamma, \gamma)$ (Algorithm 2).
6: **end function**
7: **function** REGAGNOSTIC$(S, \mathcal{A}_w, \gamma)$
8:      Let $h^\star \in \mathcal{H}$ be such that $\hat{\mathrm{err}}_{S,\ell_{\mathrm{abs}}}(h^\star) \leq \inf_{h \in \mathcal{H}} \hat{\mathrm{err}}_{S,\ell_{\mathrm{abs}}}(h) + \gamma$.
9:      Define $\bar{S} = \{(x_i, \hat{y}_i) \, : \, i \in [n]\}$, where $\hat{y}_i = h^\star(x_i)$.
10:      **return** REGREALIZABLE$(\bar{S}, \mathcal{A}_w, \gamma)$.
11: **end function**

---

Lemma 17 instantiated on $\mathscr{T}_{\gamma,k}(\mathcal{H})$ yields a weak learner $\mathcal{A}_w$ for the class, and we can use this weak learner to show the existence of the required subsequences $T_1, \ldots, T_l$ in Line 4 of Algorithm 1. Alternately, one could also use a variant of multi-class boosting (Brukhim et al., 2023, Algorithm 1) to provide a construction of the subsequences $T_1, \ldots, T_l$.

**Lemma 18 (Minimax)** *Let $\mathcal{A}_w$ be the weak learner given by Lemma 17 for the partial concept class $\mathscr{T}_{\gamma,k}(\mathcal{H})$. Let $\tilde{S}$ be the sample constructed by REGREALIZABLE in Line 3 of Algorithm 1. Then, there exist subsequences $T_1, \ldots, T_l$ of $\tilde{S}$ satisfying the condition in Line 4.*

The proof is standard (e.g., see Lemma 7.5 in Charikar and Pabbaraju (2023)), and involves defining an appropriate two-player game, where Player 1's pure strategies are examples $(x', y') \in \tilde{S}$, and Player 2's pure strategies are subsequences $T$ of $\tilde{S}$. Von Neumann's minimax theorem (Neumann, 1928) applied to the game ensures the existence of a distribution over subsequences of $\tilde{S}$, such that when $T_1, \ldots, T_l$ are sampled independently from it, the condition in Line 4 is satisfied with nonzero probability by a Chernoff bound. We defer the details to Section A.2.

### 3.1.3. Bounding the Training Error

We now show how we can use the guarantee given by Lemma 18 to bound the training error of the function $H(\cdot)$ returned by REGREALIZABLE when invoked on a sample $S$ realizable by $\mathcal{H}$. From Line 4 in Algorithm 1, we have a list predictor $J$, that satisfies $J(x') \ni y'$ for every $(x', y') \in \tilde{S}$. Recall that every $(x', y')$ corresponds to some $((x_i, \tau), \text{Thr}_\tau(y_i))$, where $(x_i, y_i) \in S$. The predictor $J$ gives us access to accurate predictions for every $(x_i, \tau)$, and our remaining task is to extract a prediction close to $y_i$ from these. We will show that this can be accomplished by obtaining a carefully scaled sum of the predictions of $J$.

It will be helpful to define the following classifier function $\mathfrak{f}$, which when given an input $a \in [0, 1]$ and a $k$-threshold $\tau$, determines the location of $a$ with respect to $\tau$.

**Definition 19 (Classifier function $\mathfrak{f}(a, \tau)$)** *Given $a \in [0, 1]$ and $k$-threshold $\tau \in \mathcal{D}_\gamma^{(k)}$, $\mathfrak{f}(a, \tau)$ predicts an integer in $\{0, 1, \ldots, k\}$ denoting the number of components in $\tau$ that $a$ is bigger than. Formally,*

$$\mathfrak{f}(a, \tau) := |\{i : a > \tau_i\}|.$$

We now state the following claim (proof is a calculation, and is given in Section A.3), which shows that summing up classifier functions over every $k$-threshold allows us to recover a scaled version of the input, provided that the input is on the discretized grid $\mathcal{D}_\gamma$.

**Claim 20 (Scaled sum)** *Consider $a \in \mathcal{D}_\gamma$ and $k$-threshold $\tau \in \mathcal{D}_\gamma^{(k)}$. Then, we have*

$$a = \gamma \binom{\left\lfloor \frac{1}{\gamma} \right\rfloor}{k - 1}^{-1} \sum_{\tau \in \mathcal{D}_\gamma^{(k)}} \mathfrak{f}(a, \tau).$$

Claim 20 provides us a way to reverse engineer a value of $a$, provided we have evaluations of the classifier function $\mathfrak{f}$ on $a$ with respect to different $k$-thresholds. Crucially, the labels $y'_{i,\tau}$ in the dataset $\tilde{S}$ constructed in Line 2 are, in fact, evaluations of the classifier function $\mathfrak{f}$ on the labels $y_i$! In particular, so long as $\tau$ is a $k$-threshold having none of its components be too close to $y_i$, $y'_{i,\tau} = \text{Thr}_\tau(y_i) = \mathfrak{f}(y_i, \tau)$. We characterize this set of "not-too-close" $k$-thresholds via the following definition:

**Definition 21 (Separated $k$-threshold set $\mathcal{V}(a, \gamma)$)** *Given $a \in [0, 1]$ and $k$-threshold $\tau \in \mathcal{D}_\gamma^{(k)}$, the separated $k$-threshold set $\mathcal{V}(a, \gamma)$ is the set of all $k$-thresholds $\tau$, which satisfy that each component of $\tau$ is at least $\frac{\gamma}{2}$-separated from $a$. Namely,*

$$\mathcal{V}(a, \gamma) := \left\{ \tau' \in \mathcal{D}_\gamma^{(k)} : \min_{j \in [k]} |\tau'_j - a| \geq \frac{\gamma}{2} \right\}. \tag{3}$$

Thus, in Line 2 of Algorithm 1, $y'_{i,\tau}$ equals $\mathfrak{f}(y_i, \tau)$ whenever $\tau$ belongs to the set $\mathcal{V}(y_i, \gamma)$, and $\star$ otherwise. Moreover, the construction of $\tilde{S}$ in Line 3 only includes $\tau \in \mathcal{V}(y_i, \gamma)$.

Now, observe further that for $\tau \in \mathcal{V}(y_i, \gamma)$, by our guarantee on the list predictor $J$, $J(x_i, \tau)$ contains $y'_{i,\tau} = \mathfrak{f}(y_i, \tau)$. If only it were the case that $y_i \in \mathcal{D}_\gamma$, and that $\mathcal{V}(y_i, \gamma) = \mathcal{D}_\gamma^{(k)}$, we could inspect every $J(x_i, \tau)$, locate $\mathfrak{f}(y_i, \tau)$ in it, and then sum up all these values to obtain $y_i$, as

suggested by Claim 20. Unfortunately, $y_i$ might not be on the discretized grid $\mathcal{D}_\gamma$, and many $k$-thresholds might not be included in $\mathcal{V}(y_i, \gamma)$. Nevertheless, we show that there is still merit in this strategy, as formalized by the following lemma:

**Lemma 22** *Let $\mathcal{H} \subseteq [0,1]^{\mathcal{X}}$ be a hypothesis class having $\mathbb{D}_{\gamma/2,k}^{\mathrm{fat}}(\mathcal{H}) < \infty$, and let $\mathcal{A}_w$ be the weak learner for $\mathcal{T}_{\gamma,k}(\mathcal{H})$ given by Lemma 17. Let $S = \{(x_i, y_i)\}_{i \in [n]} \in (\mathcal{X} \times [0,1])^n$ be a sample realizable by $\mathcal{H}$, and consider invoking* $\mathrm{REGREALIZABLE}(S, \mathcal{A}_w, \gamma)$. *Then, there exist indexing functions* $\mathsf{j}_1, \dots, \mathsf{j}_n : \mathcal{D}_\gamma^{(k)} \to [k]$ *that index into the predictions of $J$ on $S$ (Line 4), such that*

*(a)* $[J(x_i, \tau)]_{\mathsf{j}_i(\tau)} = \mathfrak{f}(y_i, \tau)$, $\forall \tau \in \mathcal{V}(y_i, \gamma), \forall i \in [n]$.

*(b)* $\left| \min\left( 1, \gamma \left( \left\lfloor \frac{1}{\gamma} \right\rfloor \atop k-1 \right)^{-1} \sum_{\tau \in \mathcal{D}_\gamma^{(k)}} [J(x_i, \tau)]_{\mathsf{j}_i(\tau)} \right) - y_i \right| \leq (2k+1)\gamma \ \forall i \in [n].$

The proof of this lemma is given in Section A.4. While part (a) readily follows from the guarantee on $J$ given by Lemma 18, part (b) follows by arguing that the $k$-thresholds not included in the set $\mathcal{V}(y_i, \tau)$ are not too many, and the error due to these excluded thresholds is small.

More importantly, the existence of the indexing functions in Lemma 22 suggests the following strategy for obtaining a list prediction on any given $x$: consider the list hypothesis $J(x, \cdot) : \mathcal{D}_\gamma^{(k)} \to \{0, 1, \dots, k\}^k$ given in Line 4, and iterate over every possible indexing function $\mathsf{j} : \mathcal{D}_\gamma^{(k)} \to [k]$: if the indexing function $\mathsf{j}$ satisfies the properties of Lemma 22, keep track of the scaled sum of values indexed into by the indexing function. Finally, return a short list of labels that covers every number that we kept track of. This strategy is formalized in the Merge Lists procedure given in Algorithm 2.

---

**Algorithm 2** The Merge Lists procedure

**Input:** Function $\mathcal{J} : \mathcal{D}_\gamma^{(k)} \to \{0, 1, \dots, k\}^k$, radius $r > 0$, parameter $\gamma \in (0, 1)$.
**Output:** $A \in [0,1]^k$

1: **function** $\mathrm{MERGELISTS}(\mathcal{J}, r, \gamma)$
2:      Initialize $\mathcal{C} = \emptyset$.
3:      **for** every function $\mathsf{j} : \mathcal{D}_\gamma^{(k)} \to [k]$ **do**
4:          Let $\hat{c} := \min\left( 1, \gamma \left( \left\lfloor \frac{1}{\gamma} \right\rfloor \atop k-1 \right)^{-1} \sum_{\tau \in \mathcal{D}_\gamma^{(k)}} [\mathcal{J}(\tau)]_{\mathsf{j}(\tau)} \right).$
5:          **if** $\exists c \in [0,1]$ such that $\{[\mathcal{J}(\tau)]_{\mathsf{j}(\tau)} = \mathfrak{f}(c, \tau), \forall \tau \in \mathcal{V}(c, \gamma)\}$ and $|c - \hat{c}| \leq \frac{r}{3}$ **then**
6:              Append $\hat{c}$ to the set $\mathcal{C}$.
7:          **end if**
8:      **end for**
9:      **return** a subset of $\mathcal{C}$ of $\leq k$ points that $r$-covers $\mathcal{C}$.[7]
10: **end function**

---

The claim below (proof in Section A.5) guarantees that the final step of Merge Lists is valid: there always exists an $r$-cover constituting of at most $k$ points for a suitable value of $r$.

---

7. A subset $\bar{\mathcal{C}} \subseteq \mathcal{C}$ $r$-covers $\mathcal{C}$ if $\forall \hat{c} \in \mathcal{C}, \exists \bar{c} \in \bar{\mathcal{C}}$ s.t. $|\bar{c} - \hat{c}| \leq r$. Such a cover exists by Claim 23.

**Claim 23 (Existence of cover)** *There always exists a subset of $\mathcal{C}$ of at most $k$ points that $r$-covers $\mathcal{C}$ in Line 9 of Algorithm 2 when it is called with $r = 6k\gamma + 3\gamma$.*

Finally, using Lemma 22 and Claim 23, we can show that the list hypothesis $H(\cdot)$ returned by REGREALIZABLE when invoked on a realizable training sample suffers low error on the training data.

**Lemma 24 (Training error for realizable regression)** *Let $\mathcal{H} \subseteq [0,1]^{\mathcal{X}}$ be a hypothesis class having $\mathbb{D}^{\text{fat}}_{\gamma/2,k}(\mathcal{H}) < \infty$, and let $\mathcal{A}_w$ be the weak learner for $\mathscr{T}_{\gamma,k}(\mathcal{H})$ given by Lemma 17. Let $S = \{(x_i, y_i)\}_{i \in [n]} \in (\mathcal{X} \times [0,1])^n$ be a sample realizable by $\mathcal{H}$, and consider invoking REGREALIZABLE$(S, \mathcal{A}_w, \gamma)$. Then, the hypothesis $H(\cdot)$ output by it satisfies*

$$\hat{\text{err}}_{S,\ell_{abs}}(H) \leq \min((8k+4)\gamma, 1).$$

**Proof** From Lemma 22, we have that for each $i \in [n]$, there is an indexing function $\mathfrak{j}_i : \mathcal{D}^{(k)}_\gamma \to [k]$ satisfying $[J(x_i, \tau)]_{\mathfrak{j}_i(\tau)} = \mathfrak{f}(y_i, \tau), \forall \tau \in \mathcal{V}(y_i, \gamma)$, and

$$\left| \min\left(1, \gamma \binom{\left\lfloor \frac{1}{\gamma} \right\rfloor}{k-1}^{-1} \sum_{\tau \in \mathcal{D}^{(k)}_\gamma} [J(x_i, \tau)]_{\mathfrak{j}_i(\tau)} \right) - y_i \right| \leq (2k+1)\gamma.$$

Let $\hat{c}_i = \min\left(1, \gamma \binom{\left\lfloor \frac{1}{\gamma} \right\rfloor}{k-1}^{-1} \sum_{\tau \in \mathcal{D}^{(k)}_\gamma} [J(x_i, \tau)]_{\mathfrak{j}_i(\tau)} \right)$. Observe that for the chosen value of $r = 6k\gamma + 3\gamma$, $\hat{c}_i$ will necessarily belong to the set $\mathcal{C}$ at the end of the for loop in Line 8 of MERGE-LISTS$(J(x_i, \cdot), r, \gamma)$. From Claim 23, we know that this $\hat{c}_i$ is $r$-covered by the list that the procedure returns. Summarily, we obtain that

$$\min_{j \in [k]} |H(x_i)_j - y_i| \leq \min_{j \in [k]} |H(x_i)_j - \hat{c}_i| + |\hat{c}_i - y_i|$$

$$\leq 6k\gamma + 3\gamma + 2k\gamma + \gamma = 8k\gamma + 4\gamma.$$

Finally, note that any $\hat{c}$ in Line 4 in Algorithm 2 is contained in $[0,1]$, and since the list $H(x_i)$ is comprised of some such $\hat{c}$ values, $\min_{j \in [k]} |H(x_i)_j - y_i| \leq 1$. ∎

**Remark 25** *It might appear at this point that we have put in a lot of effort to obtain something clearly suboptimal as compared to simply invoking ERM on the sample (which, assuming realizability, attains 0 training error). The point, as we shall see ahead, is that the algorithm from above can be viewed as a sample compression scheme, and hence generalizes.*

### 3.1.4. Bounding the Generalization Error

With all the machinery above in place, it remains to observe that the execution of REGAGNOSTIC can be instantiated as a sample compression scheme. On an input sample $S$, REGAGNOSTIC first prepares $\bar{S}$ in Line 9, and then invokes REGREALIZABLE on $\bar{S}$. REGREALIZABLE in turn prepares the sample $\tilde{S}$ in Line 3, and then obtains the subsequences $T_1, \ldots, T_l$ in Line 4. Note that hereafter,

the output $H(\cdot)$ of REGREALIZABLE depends *only* on these subsequences, and that the total size of these subsequences is *sublinear* in the size of the original sample $S$. Thus, the mapping from $S$ to $T_1, \ldots, T_l$ can be viewed as the compression function, and the mapping from $T_1, \ldots, T_l$ to $H(\cdot)$ can be viewed as the reconstruction function. Invoking Lemma 10 completes the proof. The formal details are given in Section A.6.

## 3.2. Lower Bound for Agnostic List Regression

Next, we show that finiteness of the $(\gamma, k)$-fat-shattering dimension at all scales is *necessary* for agnostic $k$-list learnability.

**Theorem 26 (Finite $(\gamma, k)$-fat-shattering dimension necessary)** *Let $\mathcal{H} \subseteq [0,1]^{\mathcal{X}}$ be a hypothesis class having $(\gamma, k)$-fat-shattering dimension $\mathbb{D}^{\mathrm{fat}}_{\gamma,k}(\mathcal{H})$, and let $\mathcal{A}$ be any $k$-list regression algorithm for $\mathcal{H}$ in the agnostic setting. Then, for any $\varepsilon, \delta \in (0,1)$ satisfying $\varepsilon, \delta \leq \frac{\gamma}{8(k+1)}$,*

$$m^{k,\mathrm{ag}}_{\mathcal{A},\mathcal{H}}(\varepsilon, \delta) \geq \widetilde{\Omega}\left(\frac{\mathbb{D}^{\mathrm{fat}}_{\gamma,k}(\mathcal{H})}{k^k}\right).$$

Towards proving Theorem 26, we first show in Section 3.2.1 that finiteness of a related but tighter quantity, which we term the $(\gamma, k)$-strong-fat-shattering dimension, is necessary for this task. We then relate both these dimensions to a *higher-order packing number*, which we term the $k$-ary $\gamma$-packing number (Section 3.2.2) of the hypothesis class. While the $(\gamma, k)$-strong-fat-shattering dimension upper bounds the $k$-ary packing number (Section 3.2.3), the $(\gamma, k)$-fat-shattering dimension lower bounds it (Section 3.2.4). In particular, these relations allow us to show that if the $(\gamma, k)$-fat-shattering dimension (of an appropriately discretized version of the hypothesis class) is infinite, then so is the $(\gamma, k)$-strong-fat-shattering dimension (Section 3.2.5). Putting this together with the earlier established lower bound in terms of the $(\gamma, k)$-strong-fat-shattering dimension completes the proof (Section 3.2.6).

### 3.2.1. Stronger Notion of $k$-Fat-Shattering

We will require defining a notion of fat-shattering that is *stronger* than the notion of $(\gamma, k)$-fat-shattering from Definition 11 and is more amenable to proving lower bounds for learnability. This definition generalizes a definition due to Simon (1997).

**Definition 27 ($(\gamma, k)$-strong-fat-shattering dimension)** *A hypothesis class $\mathcal{H} \subseteq [0,1]^{\mathcal{X}}$ $(\gamma, k)$-strongly-fat-shatters a sequence $S = (x_1, x_2, \ldots, x_d) \in \mathcal{X}^d$ if there exist vectors $c_1, c_2, \ldots, c_d \in [0,1]^{k+1}$ satisfying*

$$c_{i,j+1} \geq c_{i,j} + 2\gamma, \forall j \in \{1, \ldots, k\}, \forall i \in \{1, \ldots, d\}$$

*such that $\forall b \in \{1, \ldots, k+1\}^d$, there exists $h_b \in \mathcal{H}$ such that $h_b(x_i) = c_{i,b_i}, \forall i \in \{1, \ldots, d\}$. The $(\gamma, k)$-strong-fat-shattering dimension $\mathbb{D}^{\mathrm{sfat}}_{\gamma,k}(\mathcal{H})$ is defined to be the size of the largest $(\gamma, k)$-strongly-fat-shattered sequence.*

Using the standard lower bound template of considering the uniform distribution on a shattered set (e.g., see Lemma 25 in Bartlett and Long (1998)), we can show that finiteness of the $(\gamma, k)$-strong-fat-shattering dimension is necessary for agnostic list learnability. The proof details are given in Section B.1.

**Lemma 28 (Finite $(\gamma, k)$-strong-fat-shattering dimension necessary)** *Let $\mathcal{H} \subseteq [0, 1]^{\mathcal{X}}$ be a hypothesis class having $(\gamma, k)$-strong-fat-shattering dimension $\mathbb{D}^{\mathrm{sfat}}_{\gamma, k}(\mathcal{H})$, and let $\mathcal{A}$ be any $k$-list regression algorithm for $\mathcal{H}$ in the agnostic setting. Then, for any $\varepsilon, \delta \in (0, 1)$ satisfying $\varepsilon, \delta \leq \frac{\gamma}{2(k+1)}$,*

$$m^{k,\mathrm{ag}}_{\mathcal{A},\mathcal{H}}(\varepsilon, \delta) \geq \Omega\left(\mathbb{D}^{\mathrm{sfat}}_{\gamma, k}(\mathcal{H})\right).$$

### 3.2.2. Higher-Order Packing

We want to replace the $(\gamma, k)$-strong-fat-shattering dimension in Lemma 28 with the $(\gamma, k)$-fat-shattering dimension to obtain Theorem 26—this requires us to relate these two quantities. Note that $\mathbb{D}^{\mathrm{sfat}}_{\gamma, k}(\mathcal{H}) \leq \mathbb{D}^{\mathrm{fat}}_{\gamma, k}(\mathcal{H})$, since $(\gamma, k)$-strong-fat-shattering implies $(\gamma, k)$-fat-shattering. In order to bound $\mathbb{D}^{\mathrm{fat}}_{\gamma, k}(\mathcal{H})$ in terms of $\mathbb{D}^{\mathrm{sfat}}_{\gamma, k}(\mathcal{H})$ from above, we will require relating each of these quantities to a suitable *packing number* of the hypothesis class. Packing numbers generally characterize the largest subset of a set (in a metric space) such that every two members of the set are far away from each other, or rather, are "pairwise separated". For our purposes, we will need to generalize this standard notion of pairwise separation to the notion of "$k$-wise" separation.

**Definition 29 ($k$-wise $\gamma$-separation)** *A hypothesis class $\mathcal{H} \subseteq [0, 1]^{\mathcal{X}}$ is $k$-wise $\gamma$-separated if for any $k$ distinct functions $f_1, \ldots, f_k$ in $\mathcal{H}$, there exists an $x \in \mathcal{X}$ such that $|f_i(x) - f_j(x)| \geq 2\gamma$, $\forall i, j \in [k], i \neq j$.*

**Definition 30 ($k$-ary $\gamma$-packing number)** *The $k$-ary $\gamma$-packing number of a hypothesis class $\mathcal{H} \subseteq [0, 1]^{\mathcal{X}}$, denoted $\mathcal{M}^k_{\infty}(\mathcal{H}, \gamma)$, is the size of the largest set $F \subseteq \mathcal{H}$ that is $(k+1)$-wise $\gamma$-separated.*

### 3.2.3. Relating $k$-Ary Packing to $k$-Strong-Fat-Shattering

We shall first prove an upper bound on the $k$-ary $\gamma$-packing number of a hypothesis class on *finitely many labels* in terms of its $(\gamma, k)$-strong-fat-shattering dimension. The main result of this section is a generalization of the core technical result in Alon et al. (1997), which upper bounds the packing number of a class in terms of its fat-shattering dimension, and may be of independent interest. We also remark that the purpose of such a bound is morally similar to the purpose of the list version of the celebrated Sauer-Shelah-Perles lemma (Charikar and Pabbaraju, 2023).

Let $\mathcal{L} \subset [0, 1]$ be a finite set such that $|\mathcal{L}| = B < \infty$. Following the exposition in Kakade and Tewari, for $k \geq 1, h \geq k + 1, n \geq 1$, define

$$t(h, n) = \max\left\{s : \forall \mathcal{H} \subseteq \mathcal{L}^{\mathcal{X}}, |\mathcal{X}| = n, |\mathcal{H}| = h, \mathcal{H} \text{ is } (k+1)\text{-wise } \gamma\text{-separated}\right.$$
$$\left. \implies \mathcal{H} \ (\gamma, k)\text{-strongly-fat-shatters at least } s \ (X, \boldsymbol{c}) \text{ pairs}\right\}, \tag{4}$$

and define $t(h, n)$ to be $\infty$ if there does not exist any $(k + 1)$-wise $\gamma$-separated $\mathcal{H}$ of size $h$. Note that $t(h, n)$ is non-decreasing in its first argument.

**Claim 31 (Large $t(h, n) \implies$ small $k$-ary packing)** *Fix $\mathcal{L} \subset [0, 1]$ such that $|\mathcal{L}| = B < \infty$, and let $\mathcal{X}$ be a finite domain with $|\mathcal{X}| = n$. Let $y = \sum_{i=1}^{d} \binom{n}{i} \binom{B}{k+1}^i$. If $t(h, n) \geq y$ for some $h$, then any hypothesis class $\mathcal{H} \subseteq \mathcal{L}^{\mathcal{X}}$ that has $(\gamma, k)$-strong fat-shattering dimension $\leq d$ satisfies $\mathcal{M}^k_{\infty}(\mathcal{H}, \gamma) < h$.*

**Proof** Suppose $\mathcal{M}_\infty^k(\mathcal{H}, \gamma) \geq h$. Then, there exists a set $F \subseteq \mathcal{H}$ of size $h$ comprising of $(k+1)$-wise $\gamma$-separated functions. Since $t(h, n) \geq y$, by the definition of $t(h, n)$, this means that $F$ $(\gamma, k)$-strongly-fat-shatters at least $y$ $(X, \boldsymbol{c})$ pairs. But note that the $(\gamma, k)$-strong fat-shattering dimension of $\mathcal{H}$ is at most $d$, and hence the size of the largest sequence that $F$ $(\gamma, k)$-strongly-fat-shatters is at most $d$. Thus, the total number of distinct $(X, \boldsymbol{c})$ pairs that $F$ can $(\gamma, k)$-strongly-fat-shatter is strictly smaller than $\sum_{i=1}^d \binom{n}{i} \binom{B}{k+1}^i$, which is a contradiction. ∎

We proceed with an inductive proof that establishes the growth of $t(h, n)$. A simple yet clever insight that we crucially rely on (Claim 48 in Section B.2) is the following: if we have $k+1$ sets each of size at least $m$, then the sum of the sizes of their union and intersection is at least $\left(\frac{k+1}{k}\right) m$. The $k = 1$ version of this is used to establish the inductive step in Alon et al. (1997), and follows from the standard equality $|A \cup B| = |A| + |B| - |A \cap B|$.

**Lemma 32 (Upper bounding $k$-ary $\gamma$-packing number)** *Fix $\mathcal{L} \subset [0, 1]$ such that $|\mathcal{L}| = B$, and let $\mathcal{X}$ be a finite domain with $|\mathcal{X}| = n$. Let $\mathcal{H} \subseteq \mathcal{L}^{\mathcal{X}}$ be a hypothesis class having $(\gamma, k)$-strong-fat-shattering dimension $\mathbb{D}_{\gamma,k}^{\mathrm{sfat}}(\mathcal{H}) = d$. Then, for $y = \sum_{i=1}^d \binom{n}{i} \binom{B}{k+1}^i$,*

$$\mathcal{M}_\infty^k(\mathcal{H}, \gamma) < (k+1) \left[(k+1)B^{k+1}n\right]^{\left\lceil \log_{\frac{k+1}{k}} y \right\rceil}. \tag{5}$$

**Proof** By Claim 31, it suffices to show that

$$t\left((k+1) \left[(k+1)B^{k+1}n\right]^{\left\lceil \log_{\frac{k+1}{k}} y \right\rceil}, n\right) \geq y. \tag{6}$$

We will show that:

$$t(k+1, n) \geq 1, \qquad \forall n \geq 1, \tag{7}$$

$$t\left((k+1)^2 mnB^{k+1}, n\right) \geq \left(\frac{k+1}{k}\right) t((k+1)m, n-1), \qquad \forall m \geq 1, n \geq 2. \tag{8}$$

For any $(k+1)$-wise $\gamma$-separated $\mathcal{H}$ of size $k+1$, by definition, there must exist an $x$ at which $|f_i(x) - f_j(x)| \geq 2\gamma$ for all $f_i, f_j \in \mathcal{H}$—this means that $\mathcal{H}$ $(\gamma, k)$-strongly-fat-shatters $\{x\}$, giving us (7).

Next, consider any $(k+1)$-wise $\gamma$-separated $\mathcal{H}$ of size $(k+1)^2 mnB^{k+1}$ (if such an $\mathcal{H}$ does not exist, the left hand side of (8) is $\infty$ and the inequality holds). Partition the functions in $\mathcal{H}$ arbitrarily into groups of size $k+1$. For each group $G = \{f_1, \ldots, f_{k+1}\}$, there must exist an $x \in \mathcal{X}$ which witnesses the $(k+1)$-wise $\gamma$-separation of the functions in $G$. Fix such an $x$ and denote it by $\chi(G)$.

Now, for every $x \in \mathcal{X}$, and $c_1, \ldots, c_{k+1} \in \mathcal{L}$ such that $|c_i - c_{i+1}| \geq 2\gamma, \forall i \in [k]$, define

$$\mathrm{bin}(x, c_1, \ldots, c_{k+1}) = \{G = \{f_1, \ldots, f_{k+1}\} : \chi(G) = x, \{f_1(x), \ldots, f_{k+1}(x)\} = \{c_1, \ldots, c_{k+1}\}\}. \tag{9}$$

Namely, $\mathrm{bin}(x, c_1, \ldots, c_{k+1})$ contains all the groups that are $(k+1)$-wise $\gamma$-separated at $x$ according to $c_1, \ldots, c_{k+1}$. We have that the total number of possible bins are at most $n\binom{B}{k+1} \leq nB^{k+1}$. Since the number of groups is $(k+1)mnB^{k+1}$, by the pigeonhole principle, there must exist a bin with at

least $(k+1)m$ groups—let this be $\mathrm{bin}(x^\star, c_1^\star, \ldots, c_{k+1}^\star)$, and arbitrarily trim its size to be exactly $(k+1)m$.

Now, let

$$F_1 = \bigcup_{G \in \mathrm{bin}(x^\star, c_1^\star, \ldots, c_{k+1}^\star)} \{f \in G : f(x^\star) = c_1^\star\}$$

$$\vdots$$

$$F_{k+1} = \bigcup_{G \in \mathrm{bin}(x^\star, c_1^\star, \ldots, c_{k+1}^\star)} \{f \in G : f(x^\star) = c_{k+1}^\star\}.$$

Note that $|F_i| = (k+1)m$ for every $F_i$. Now fix an $F_i$, and note that every function in $F_i$ labels $x^\star$ identically as $c_i^\star$. This means that the functions in $F_i$ must be $(k+1)$-wise $\gamma$-separated at some $x \in \mathcal{X} \setminus \{x^\star\}$. Then, consider the set $F_i|_{\mathcal{X} \setminus \{x^\star\}}$. By the definition of $t(\cdot, \cdot)$, this set $(\gamma, k)$ strongly fat-shatters at least $t((k+1)m, n-1)$ many $(X, \boldsymbol{c})$ pairs—collect these pairs in a set $S_i$. Any pair in $S_i$ is certainly also shattered by $\mathcal{H}$. Now, consider an $(X, \boldsymbol{c})$ pair that is simultaneously shattered by *every* $F_i$. Then, because $c_1, \ldots, c_{k+1}$ differ by at least $2\gamma$ pairwise, observe that $F$ additionally $(\gamma, k)$ strongly fat-shatters the pair $(X \cup x^\star, \boldsymbol{c} \cup \{c_1, \ldots, c_{k+1}\})$. Summarily, we have shown that $F$ $(\gamma, k)$ strongly fat-shatters at least

$$\left| \bigcup_{i=1}^{k+1} S_i \right| + \left| \bigcap_{i=1}^{k+1} S_i \right|$$

many $(X, \boldsymbol{c})$ pairs. Since each $|S_i| \geq t((k+1)m, n-1)$, by Claim 48, we get that this is at least $\left( \frac{k+1}{k} \right) t((k+1)m, n-1))$ many pairs, from which we conclude (8).

Now, let $r$ be such that $n > r \geq 1$. Then, from (7), and repeatedly applying (8), we get

$$t\left( (k+1)^{r+1} \cdot B^{r(k+1)} \cdot n(n-1) \ldots (n-(r-1)) \cdot m, n \right)$$

$$\geq \left( \frac{k+1}{k} \right) \cdot t\left( (k+1)^r \cdot B^{(r-1)(k+1)} \cdot (n-1) \ldots (n-(r-1)) \cdot m, n-1 \right)$$

$$\vdots$$

$$\geq \left( \frac{k+1}{k} \right)^r \cdot t\left( (k+1)m, n-r \right) \geq \left( \frac{k+1}{k} \right)^r.$$

Set $m = 1$ in the above. Since $n^r \geq n(n-1) \ldots (n-(r-1))$ and $t(\cdot, \cdot)$ is non-decreasing in its first argument, we get that for $n > r \geq 1$,

$$t\left( (k+1) \left[ (k+1) B^{k+1} n \right]^r, n \right) \geq \left( \frac{k+1}{k} \right)^r. \tag{10}$$

Thus, if $n > \left\lceil \log_{\frac{k+1}{k}} y \right\rceil$, setting $r = \left\lceil \log_{\frac{k+1}{k}} y \right\rceil$ in (10) gives us (6). Otherwise, if $n \leq \left\lceil \log_{\frac{k+1}{k}} y \right\rceil$, observe that

$$(k+1) \left[ (k+1) B^{k+1} n \right]^{\left\lceil \log_{\frac{k+1}{k}} y \right\rceil} > B^n,$$

which is the maximum number of functions possible from $\mathcal{X} \to \mathcal{L}$. Thus, a $(k+1)$-wise $\gamma$-separated set $F$ of this size does not exist. By definition, $t\left( (k+1) \left[ (k+1)B^{k+1}n \right]^{\left\lceil \log_{\frac{k+1}{k}} y \right\rceil}, n \right) = \infty$, and hence (6) still holds. ∎

### 3.2.4. Relating $k$-Ary Packing to $k$-Fat-Shattering

Next, we lower bound the $k$-ary $\gamma$-packing number of a hypothesis class in terms of its $(\gamma, k)$-fat-shattering dimension. In the case of $k = 1$, the hypotheses that realize a $\gamma$-fat-shattered set are pairwise separated, and hence constitute a packing. However, for $k > 1$, it was not a priori clear to us how to procedurally infer a $k$-ary packing from a $k$-fat-shattered set. As we could not find a result in the literature that fits our needs, we prove an elementary lower bound using the probabilistic method, and include a proof in Section B.3.

**Lemma 33 (Lower Bounding $k$-Ary $\gamma$-Packing Number)** *Let $\mathcal{H} \subseteq [0,1]^{\mathcal{X}}$ be a hypothesis class having $(\gamma, k)$-fat-shattering dimension $\mathbb{D}_{\gamma,k}^{\mathrm{fat}}(\mathcal{H})$. Then,*

$$\mathcal{M}_{\infty}^{k}(\mathcal{H}, \gamma) \geq \exp\left( \Omega\left( \frac{\mathbb{D}_{\gamma,k}^{\mathrm{fat}}(\mathcal{H})}{k^k} \right) \right). \tag{11}$$

### 3.2.5. Relating $k$-Strong-Fat-Shattering Dimension to $k$-Fat-Shattering Dimension

We can now relate the $(\gamma, k)$-fat-shattering and $(\gamma, k)$-strong fat-shattering dimensions of a hypothesis class via its $k$-ary $\gamma$-packing number.

**Lemma 34 (Relating $k$-strong-fat-shattering to $k$-fat-shattering)** *Fix $\mathcal{L} \subset [0,1]$ such that $|\mathcal{L}| = B < \infty$. Let $\mathcal{H} \subseteq \mathcal{L}^{\mathcal{X}}$ be a hypothesis class having $(\gamma, k)$-fat-shattering dimension $\mathbb{D}_{\gamma,k}^{\mathrm{fat}}(\mathcal{H})$ and $(\gamma, k)$-strong-fat-shattering dimension $\mathbb{D}_{\gamma,k}^{\mathrm{sfat}}(\mathcal{H})$. Then,*

$$\mathbb{D}_{\gamma,k}^{\mathrm{sfat}}(\mathcal{H}) \geq \widetilde{\Omega}\left( \frac{\mathbb{D}_{\gamma,k}^{\mathrm{fat}}(\mathcal{H})}{k^k} \right), \tag{12}$$

*where the $\widetilde{\Omega}(\cdot)$ hides* polylog *factors in $k$, $\mathbb{D}_{\gamma,k}^{\mathrm{fat}}(\mathcal{H})$ and $B$.*

The proof simply combines the upper bound on the $k$-ary $\gamma$-packing number from Lemma 32 and the lower bound from Lemma 33, and is given in Section B.4. In particular, Lemma 34 indicates that for finite, fixed $k$ and $B$, if $\mathbb{D}_{\gamma,k}^{\mathrm{fat}}(\mathcal{H}) = \infty$, then $\mathbb{D}_{\gamma,k}^{\mathrm{sfat}}(\mathcal{H}) = \infty$.

### 3.2.6. Inferring Necessity of $k$-Fat-Shattering from $k$-Strong-Fat-Shattering

Now that we have related $\mathbb{D}_{\gamma,k}^{\mathrm{fat}}(\mathcal{H})$ to $\mathbb{D}_{\gamma,k}^{\mathrm{sfat}}(\mathcal{H})$, it remains to translate the lower bound in terms of the $\mathbb{D}_{\gamma,k}^{\mathrm{sfat}}(\mathcal{H})$ from Lemma 28 to a lower bound in terms of $\mathbb{D}_{\gamma,k}^{\mathrm{fat}}(\mathcal{H})$. In order for this, we need to first convert the given class $\mathcal{H}$ to a class on finitely many labels, so as to satisfy the criteria of Lemma 34. We do this by discretizing the members of $\mathcal{H}$ to a grid.

**Definition 35 (Discretized hypothesis class)** *For $\alpha \in (0, 1)$, and $x \in [0, 1]$, define*

$$Q_\alpha(x) = \alpha \left\lfloor \frac{x}{\alpha} \right\rfloor. \tag{13}$$

*For a vector $c \in [0, 1]^k$, define $Q_\alpha(c) = (Q_\alpha(c_1), \ldots, Q_\alpha(c_k))$. For a function $f : \mathcal{X} \to [0, 1]$, define $f^\alpha : \mathcal{X} \to [0, 1]$ such that $f^\alpha(x) = Q_\alpha(f(x))$. Finally, for a function class $\mathcal{H} \subseteq [0, 1]^{\mathcal{X}}$, define the discretized function class $\mathcal{H}^\alpha = \{f^\alpha : f \in \mathcal{H}\}$.*

The following two claims relate the learnability of $\mathcal{H}$ and $\mathcal{H}^\alpha$. The proofs of these essentially follow once we notice that function values do not change by more than $\alpha$ (which is chosen to be smaller than the shattering width $\gamma$) upon discretization, and details are provided in Section B.5.

**Claim 36** *Let $\mathcal{H} \subseteq [0, 1]^{\mathcal{X}}$ be a hypothesis class having $(\gamma, k)$-fat-shattering dimension $d$. Then for any $\alpha \leq \gamma$, the $(\max(\alpha, \gamma - \alpha), k)$-fat-shattering dimension of $\mathcal{H}^\alpha$ is at least $d$.*

**Claim 37** *Let $\mathcal{A}$ be an algorithm that agnostically $k$-list learns $\mathcal{H} \subseteq [0, 1]^{\mathcal{X}}$ with sample complexity $m_{\mathcal{A},\mathcal{H}}^{k,\mathrm{ag}}(\varepsilon, \delta)$. Then, $\mathcal{A}$ also agnostically $k$-list learns $\mathcal{H}^\alpha$, and*

$$m_{\mathcal{A},\mathcal{H}^\alpha}^{k,\mathrm{ag}}(\varepsilon + \alpha, \delta) = m_{\mathcal{A},\mathcal{H}}^{k,\mathrm{ag}}(\varepsilon, \delta).$$

We can now finally prove Theorem 26.

**Proof** [Proof of Theorem 26] Fix $\varepsilon, \delta \leq \frac{\gamma}{8(k+1)}$, and suppose it were the case that

$$m_{\mathcal{A},\mathcal{H}}^{k,\mathrm{ag}}(\varepsilon, \delta) = \tilde{o}\left(\frac{\mathbb{D}_{\gamma,k}^{\mathrm{fat}}(\mathcal{H})}{k^k}\right).$$

Let $\alpha = \frac{\gamma}{16(k+1)} \leq \frac{\gamma}{32}$. From Claim 37 and Claim 36, we get that

$$m_{\mathcal{A},\mathcal{H}^\alpha}^{k,\mathrm{ag}}(\varepsilon + \alpha, \delta) = \tilde{o}\left(\frac{\mathbb{D}_{\gamma,k}^{\mathrm{fat}}(\mathcal{H})}{k^k}\right) = \tilde{o}\left(\frac{\mathbb{D}_{\gamma-\alpha,k}^{\mathrm{fat}}(\mathcal{H}^\alpha)}{k^k}\right).$$

Observe that $\mathcal{H}^\alpha$ is a class on finitely many labels—in particular, it maps $\mathcal{X}$ to $\mathcal{L} = \left\{0, \alpha, 2\alpha, \ldots, \alpha \left\lfloor \frac{1}{\alpha} \right\rfloor\right\}$, so that $B = |\mathcal{L}| = 1 + \alpha \left\lfloor \frac{1}{\alpha} \right\rfloor < \infty$. Then, Lemma 34 implies that $\mathbb{D}_{\gamma-\alpha,k}^{\mathrm{fat}}(\mathcal{H}^\alpha) = \tilde{O}(k^k \cdot \mathbb{D}_{\gamma-\alpha,k}^{\mathrm{sfat}}(\mathcal{H}^\alpha))$. Substituting above, we get

$$m_{\mathcal{A},\mathcal{H}^\alpha}^{k,\mathrm{ag}}(\varepsilon + \alpha, \delta) = o\left(\mathbb{D}_{\gamma-\alpha,k}^{\mathrm{sfat}}(\mathcal{H}^\alpha)\right).$$

But note that

$$\varepsilon + \alpha \leq \frac{3\gamma}{16(k+1)} < \frac{\gamma - \gamma/32}{2(k+1)} \leq \frac{\gamma - \alpha}{2(k+1)},$$

which contradicts Lemma 28. Thus, it must be the case that $m_{\mathcal{A},\mathcal{H}}^{k,\mathrm{ag}}(\varepsilon, \delta) = \tilde{\Omega}\left(\frac{\mathbb{D}_{\gamma,k}^{\mathrm{fat}}(\mathcal{H})}{k^k}\right)$ as required.
∎

## 4. Realizable List Regression

We now shift our attention to the realizable setting. In the setting of standard realizable regression ($k = 1$), the $\gamma$-OIG dimension quantifies the amount of variability possible in the label of a test example $x$, given that we have pinned down labels on a training dataset $x_1, \ldots, x_n$. Informally, a learner gets confused about what to predict on $x$, if there are two hypotheses in the class that are consistent with the training data, but differ on their labels on $x$ by at least $\Omega(\gamma)$—this is what is being captured in Definition 8 by the outdegree of orientations on the one-inclusion graph of $\mathcal{H}|_{\{x_1, \ldots, x_n, x\}}$. However, if we are allowed to predict $k$ labels, for there to be confusion in the prediction on the test point, there ought to be $k + 1$ hypotheses each of which are consistent with the labels on $x_1, \ldots, x_n$, but differ *pairwise* on the test point by an amount $\Omega(\gamma)$. This intuition leads to the following natural generalization of the $\gamma$-OIG dimension, which uses the same notion of $k$-outdegree of an orientation as defined by Charikar and Pabbaraju (2023).

**Definition 38 (List orientation and scaled $k$-outdegree (Charikar and Pabbaraju, 2023))** *A $k$-list orientation of the one-inclusion graph $\mathcal{G}(\mathcal{H})$ of a hypothesis class $\mathcal{H} \subseteq [0,1]^{\mathcal{X}}$ is a mapping $\sigma^k : E \to \{V' \subseteq V : |V'| \leq k\}$ such that $\sigma^k(e) \subseteq E$ for every $e \in E$. We denote by $\sigma_i^k(e) \subseteq [0,1]^{\leq k}$ the list of numbers comprising of the $i^{th}$ entry of each vertex that the edge $e$ is oriented to.*

*For a vertex $v \in V$, which corresponds to some hypothesis $h \in \mathcal{H}$, let $v_i$ be the $i^{th}$ entry of $v$ (which corresponds to $h(i)$), and let $e_{i,v}$ be the hyperedge adjacent to $v$ in the direction $i$. For $\gamma \in (0,1)$, the (scaled) $k$-outdegree of the vertex $v$ under $\sigma^k$ is*

$$\mathsf{outdeg}^k(v; \sigma^k, \gamma) = \left| \{i \in [n] : \sigma_i^k(e_{i,v}) \not\approx_\gamma v_i\} \right|.$$

*The maximum scaled $k$-outdegree of $\sigma^k$ is denoted by $\mathsf{outdeg}^k(\sigma^k, \gamma) := \max_{v \in V} \mathsf{outdeg}^k(v; \sigma^k, \gamma)$.*

**Definition 39 ($(\gamma, k)$-OIG dimension)** *Consider a hypothesis class $\mathcal{H} \subseteq [0,1]^{\mathcal{X}}$ and let $\gamma \in (0,1)$. The $(\gamma, k)$-OIG dimension of $\mathcal{H}$ is defined as follows:*

$$\mathbb{D}_{\gamma,k}^{\mathrm{OIG}}(\mathcal{H}) := \sup \big\{ n \in \mathbb{N} : \exists S \in \mathcal{X}^n \text{ such that } \exists \text{ finite subgraph } G = (V, E) \text{ of } \mathcal{G}(\mathcal{H}|_S) \text{ such that}$$

$$\forall k\text{-list orientations } \sigma^k, \exists v \in V \text{ such that } \mathsf{outdeg}^k(v; \sigma^k, \gamma) \geq \frac{n}{2(k+1)} \big\}. \,^8$$

$$(14)$$

Again, we have that $\mathbb{D}_{\gamma,k_1}^{\mathrm{OIG}}(\mathcal{H}) \geq \mathbb{D}_{\gamma,k_2}^{\mathrm{OIG}}(\mathcal{H})$ for $k_1 < k_2$. In the spirit of Example 1, we now construct a hypothesis class $\mathcal{H}$, that has finite $(\gamma, 2)$-OIG dimension, but infinite $\gamma$-OIG dimension as well as infinite $(\gamma, 2)$-fat-shattering dimension. The former implies that $\mathcal{H}$ is not learnable in the standard realizable setting for $k = 1$. The latter implies that $\mathcal{H}$ is not agnostically 2-list learnable. As we shall see ahead, finiteness of the $(\gamma, 2)$-OIG dimension nevertheless makes $\mathcal{H}$ 2-list learnable in the realizable setting! This shows that 1) there are simple classes that are not learnable in the realizable setting with $k = 1$, but are learnable with $k = 2$, and 2) the $(\gamma, k)$-fat-shattering dimension does not characterize $k$-list learnability in the realizable setting—while finiteness of this quantity is sufficient for learnability (e.g., Corollary 13), it is not necessary. Our example is motivated by a similar example constructed in (Bartlett et al., 1994, Section 6).

---

8. We note that setting $k = 1$ recovers Definition 8—the difference in the constant i.e., $n/3$ versus $n/4$ is inconsequential.

**Example 2** *Consider a hypothesis class $\mathcal{H} \subseteq [0,1]^{\mathcal{X}}$, where $\mathcal{X} = \{x_1, x_2, \ldots, \}$ is a countably infinite domain, defined as follows: for any $b, c \in \{0,1\}^{\mathbb{N}}$, let $h_{b,c}$ be the hypothesis that maps:*

$$h_{b,c}(x_i) = \frac{1}{2} \cdot c_i + \frac{3}{8} \cdot b_i + \frac{1}{16} \sum_{j \in \mathbb{N}} 2^{-j} b_j.$$

*Let $\mathcal{H} = \{h_{b,c} : b, c \in \{0,1\}^{\mathbb{N}}\}$. We first claim that $\mathbb{D}_{\gamma,2}^{\mathrm{OIG}}(\mathcal{H}) \leq 1$ for any $\gamma \in (0, 1/2)$. Observe that once we specify $h_{b,c}(x_i)$ for any $x_i$, this immediately determines both $c_i$, and the entire vector $b$. We can see this by noticing that $c_i = 1 \iff h_{b,c}(x_i) \geq 1/2$ and $c_i = 0 \iff h_{b,c}(x_i) \leq 7/16$. Once $c_i$ is determined, $b_i$ similarly gets determined, which then completely determines all of $b$. So, consider the one-inclusion graph of the projection of $\mathcal{H}$ onto any sequence having more than one distinct $x_i$. Any edge in the graph, say in the direction $x_i$, fixes the evaluation of the hypotheses in that edge on $x_j \neq x_i$. But this fixes the entire vector $b$. Namely, all hypotheses in this edge must have the same $b$, which means that their evaluation on $x_i$ can only be one of two numbers (corresponding to $c_i = 0$ or 1) that differ by exactly $1/2$. We can thus list-orient every edge in the graph to two arbitrary hypotheses in the edge that evaluate to these two different numbers; this ensures that the $k$-outdegree of the graph is 0. Thus, $\mathbb{D}_{\gamma,2}^{\mathrm{OIG}}(\mathcal{H}) \leq 1$.*

*We now claim that $\mathbb{D}_{\gamma,2}^{\mathrm{fat}}(\mathcal{H}) = \infty$ for any $\gamma \leq \frac{1}{16}$. To see this, fix any $n$, and consider $x_1, \ldots, x_n$. Observe that for any $h_{b,c}$, if $c_i = 0, b_i = 0$, then $h_{b,c}(x_i) < \frac{1}{16}$. Else if $c_i = 1, b_i = 1$, then $h_{b,c}(x_i) > \frac{11}{16}$. Otherwise, $h_{b,c}(x_i) \in \left( \frac{6}{16}, \frac{9}{16} \right)$. Thus, we can have $h_{b,c}(x_i)$ evaluate to a number in any of these intervals by appropriately setting $b_i$ and $c_i$, which implies that $\mathcal{H}$ $(\gamma, 2)$-fat-shatters $x_1, \ldots, x_n$ for any $\gamma \leq \frac{1}{16}$.*

*Finally, we also claim that $\mathbb{D}_{\gamma}^{\mathrm{OIG}}(\mathcal{H}) = \infty$ for any $\gamma < 1/2$. Again, fix $n$, and consider $S = x_1, \ldots, x_n$. Observe that $|\mathcal{H}|_S| < \infty$, because the evaluations of hypotheses in $\mathcal{H}$ on $S$ really only depend on $b_{\leq n}, c_{\leq n}$, and there can at most be $2 \cdot 2^n$ different assignments to these. We will show that for any orientation $\sigma$ of $\mathcal{G}(\mathcal{H}|_S) = (V, E)$, there exists a vertex that has outdegree larger than $n/3$. Recall from our argument for $\mathbb{D}_{\gamma,2}^{\mathrm{OIG}}(\mathcal{H})$ above that every vertex $v \in \mathcal{G}(\mathcal{H}|_S)$ is adjacent to exactly one other vertex $v'$ in any fixed direction $i$, corresponding to a common $b$, and different $c_i$. Moreover, $v_i$ and $v_i'$ differ by exactly $1/2$. Thus, if $\sigma$ orients the underlying edge in this direction towards $v$, then $v'$ suffers, and vice-versa. We can hence argue that the average outdegree of vertices in $V$ is lower-bounded as*

$$\frac{1}{|V|} \sum_{v \in V} \mathsf{outdeg}(v; \sigma, \gamma) = \frac{1}{|V|} \sum_{e \in E} 1 = \frac{1}{|V|} \sum_{e \in E} |e|/2 = \frac{1}{2|V|} \cdot n|V| = \frac{n}{2} > \frac{n}{3}.$$

*Because the maximum outdegree is at least the average outdegree, and the above holds for every $n$, $\mathbb{D}_{\gamma}^{\mathrm{OIG}}(\mathcal{H}) = \infty$.*

We proceed to present our main results on how the $(\gamma, k)$-OIG dimension faithfully characterizes realizable $k$-list regression.

### 4.1. Upper Bound for Realizable List Regression

**Theorem 40 (Finite $(\gamma, k)$-OIG dimension is sufficient)** *Let $\mathcal{H} \subseteq [0,1]^{\mathcal{X}}$ be a hypothesis class. For any distribution $\mathcal{D}$ over $\mathcal{X} \times [0,1]$ realizable by $\mathcal{H}$ and $\gamma \in (0,1)$, there exists a list learning*

algorithm $\mathcal{A}$ which takes as input an i.i.d. sample $S \sim \mathcal{D}^n$ and outputs a $k$-list hypothesis $\mathcal{A}(S)$ satisfying

$$\mathrm{err}_{\mathcal{D}, \ell_{abs}}(\mathcal{A}(S)) \leq \gamma + \widetilde{O}\left(\frac{k \cdot \mathbb{D}_{\gamma,k}^{\mathrm{OIG}}(\mathcal{H}) + \log(1/\delta)}{n}\right)$$

with probability at least $1 - \delta$. In other words, $m_{\mathcal{A},\mathcal{H}}^{k,\mathrm{re}}(\varepsilon, \delta) \leq \widetilde{O}\left(\frac{k \cdot \mathbb{D}_{\gamma,k}^{\mathrm{OIG}}(\mathcal{H}) + \log(1/\delta)}{\varepsilon}\right)$ for $\gamma = \Theta(\varepsilon)$.

Towards deriving Theorem 40, we first show that the one-inclusion graph of a hypothesis class $\mathcal{H}$ projected onto any finite sequence of size larger than $\mathbb{D}_{\gamma,k}^{\mathrm{OIG}}(\mathcal{H})$ admits a $k$-list orientation of small $\gamma$-scaled $k$-outdegree. The existence of such an orientation will help us obtain a weak list learning algorithm for $\mathcal{H}$.

**Lemma 41 (Small $k$-out-degree list orientation)** *Let $\mathcal{H} \subseteq [0,1]^{\mathcal{X}}$ be a hypothesis class having $(\gamma, k)$-OIG dimension $\mathbb{D}_{\gamma,k}^{\mathrm{OIG}}(\mathcal{H})$. Fix $n > \mathbb{D}_{\gamma,k}^{\mathrm{OIG}}(\mathcal{H})$ and $S \in \mathcal{X}^n$. Then, there exists a $k$-list orientation of the (possibly infinite) one-inclusion graph $\mathcal{G}(\mathcal{H}|_S) = (V, E)$ such that for every $v \in V$, $\mathsf{outdeg}^k(v; \sigma^k, \gamma) < \frac{n}{2(k+1)}$.*

We note that when $\mathcal{G}|_S$ is finite, the lemma above follows directly from the definition of the $(\gamma, k)$-OIG dimension.[9] When $\mathcal{G}(\mathcal{H}|_S)$ is infinite, we still have the property that every *finite subgraph* of it admits an orientation with small $k$-outdegree. We can therefore use a careful compactness argument to show the existence of the required orientation of the entire graph. The technical details, while similar to those in (Charikar and Pabbaraju, 2023, Lemma 3.1), are slightly more involved, and are given in Section C.1.

We now use the orientation promised by Lemma 41 to construct a weak $k$-list regression algorithm $\mathcal{A}_w$ for $\mathcal{H}$. This weak learner is constructed similarly as in Section 3.1.2, and is based again on the one-inclusion graph algorithm. We can then derive the following lemma by instantiating the classical leave-one-out argument on $\mathcal{A}_w$ (proof in Section C.2).

**Lemma 42 (Weak list learner)** *Let $\mathcal{H} \subseteq [0,1]^{\mathcal{X}}$ be a hypothesis class having $(\gamma, k)$-OIG dimension $\mathbb{D}_{\gamma,k}^{\mathrm{OIG}}(\mathcal{H})$. Fix $n \geq \mathbb{D}_{\gamma,k}^{\mathrm{OIG}}(\mathcal{H})$. Then, there exists a learning algorithm $\mathcal{A}_w$, such that for any distribution $\mathcal{D}$ realizable by $\mathcal{H}$, the following guarantee holds:*

$$\mathrm{Pr}_{S \sim \mathcal{D}^n} \mathrm{Pr}_{(x,y) \sim \mathcal{D}}[\mu_S^k(x) \not\succeq_\gamma y] < \frac{1}{2(k+1)},$$

*where $\mu_S^k = \mathcal{A}_w(S)$ is the $k$-list hypothesis output by $\mathcal{A}_w$.*

The proof of Theorem 40 follows hereon using a similar chain of arguments as in Lemma 18 and Section 3.1.4. Namely, given a training sample $S \sim \mathcal{D}^n$ realizable by $\mathcal{H}$, we use the weak-list learner $\mathcal{A}_w$ above, together with the minimax theorem, to argue the existence of subsequences $T_1, \ldots, T_l$ of $S$, such that a suitable aggregation of the output of $\mathcal{A}_w$ on these subsequences satisfies the required learning guarantee. Crucially again, the total size of the subsequences is only sublinear in $n$, and hence these can be viewed as a compression of $S$. The proof details are given in Section C.3.

---

9. In fact, the definition of the dimension arises in want of this property.

We remark that Attias et al. (2023) used a similar strategy to obtain their upper bound for realizable regression in terms of the $\gamma$-OIG dimension. However, the way they aggregate the predictions of the weak learner $\mathcal{A}_w$ is via the *median boosting* algorithm (Kégl, 2003) instead. This algorithm requires an ordering on the predictions in order to compute the median. While real-valued predictions are naturally ordered, there isn't a natural notion of ordering for lists of numbers. Moreover, a given list can contain both, an accurate prediction as well as a vastly inaccurate prediction. The boosting algorithm would want to assign different weights to these predictions, but we have no way of knowing *which* prediction in the list is accurate. For these reasons, we have to resort to using minimax sampling and a different form of aggregation in our setting.

### 4.2. Lower Bound for Realizable List Regression

**Theorem 43 (Finite $(\gamma, k)$-OIG dimension is necessary)** *Let $\mathcal{H} \subseteq [0,1]^{\mathcal{X}}$ be a hypothesis class having $(\gamma, k)$-OIG dimension $\mathbb{D}_{\gamma,k}^{\mathrm{OIG}}(\mathcal{H})$, and let $\mathcal{A}$ be any $k$-list regression algorithm for $\mathcal{H}$ in the realizable setting. Then, for any $\varepsilon, \delta \in (0,1)$ satisfying $\varepsilon < \Theta(1/k^2)$ and $\delta < \sqrt{\varepsilon}$,*

$$m_{\mathcal{A},\mathcal{H}}^{k,\mathrm{re}}(\varepsilon, \delta) \geq \Omega\left(\frac{\mathbb{D}_{\gamma,k}^{\mathrm{OIG}}(\mathcal{H})}{k\sqrt{\varepsilon}}\right) \quad \text{for } \gamma = \Theta(\sqrt{\varepsilon}).$$

The proof of this theorem is given in Section C.4, and follows the approach in (Attias et al., 2023, Lemma 6). Namely, we consider the one-inclusion graph of $\mathcal{H}$ projected onto a shattered set of size $\mathbb{D}_{\gamma,k}^{\mathrm{OIG}}(\mathcal{H})$, and define an appropriate data distribution on this graph. Thereafter, given any learning algorithm $\mathcal{A}$, we construct a list orientation of the graph that orients edges towards vertices for which the algorithm $\mathcal{A}$ incurs low error. For an orientation defined in this manner, we can lower bound the $k$-outdegree of every vertex in the graph with the expected error of $\mathcal{A}$. The definition of $\mathbb{D}_{\gamma,k}^{\mathrm{OIG}}(\mathcal{H})$ ensures the existence of a vertex with large $k$-outdegree, from which we can conclude that $\mathcal{A}$ is forced to suffer large error. Crucially, in order to show existence of the required orientation above, our proof requires an additional ingredient (Claim 49) which generalizes a claim that we used in Section 3.2.4 above: given $k+1$ events $A_1, \ldots, A_{k+1}$ in a probability space, each having $\Pr[A_i] > c$, and satisfying $\cap_i A_i = \emptyset$, it holds that $\Pr[\cup_i A_i] > \left(\frac{k+1}{k}\right) c$. A proof of this claim is given in Section B.2.

**Remark 44 (Relating $\mathbb{D}_{\gamma,k}^{\mathrm{OIG}}(\mathcal{H})$ and $\mathbb{D}_{\gamma,k}^{\mathrm{fat}}(\mathcal{H})$)** *The lower bound for realizable list regression in Theorem 43 in terms of the $(\gamma, k)$-OIG dimension, and the upper bound in Corollary 13 in terms of the $(\gamma, k)$-fat-shattering dimension, imply that $\mathbb{D}_{\gamma,k}^{\mathrm{OIG}}(\mathcal{H}) \leq \mathrm{poly}(k) \cdot \mathbb{D}_{\Theta(\gamma^2),k}^{\mathrm{fat}}(\mathcal{H})$ upto constants and polylog factors. Note that Example 2 precludes upper bounding $\mathbb{D}_{\gamma,k}^{\mathrm{fat}}(\mathcal{H})$ by any finite function of $\mathbb{D}_{\gamma,k}^{\mathrm{OIG}}(\mathcal{H})$.*

## 5. Conclusion

In this work, we provided a complete characterization of when and how list regression is possible, in terms of combinatorial properties of the hypothesis class in question. We proposed natural generalizations of existing dimensions, namely the $k$-OIG dimension for realizable list regression, and the $k$-fat-shattering dimension for agnostic list regression, and showed that these dimensions individually characterize their respective learning settings. Among the other parameters, the most generous gap in our bounds is admittedly in the dependence on the list size $k$. It will be interesting

to derive more fine-grained bounds that pin down the exact required dependence on the list size. If anything, this would shed light on the extent to which such a dependence gets hidden by constants when $k = 1$.

We conclude with an open question that emerged from our efforts in deriving the upper bound for agnostic list regression.

**Construction of a List Hypothesis Class to Run ERM**

One might wonder whether it is possible to construct an agnostic list learner that is based on ERM whenever the $(\gamma, k)$-fat-shattering dimension of a class is finite at all scales. This question arises naturally, because ERM works as a PAC learner in the case of standard ($k = 1$) agnostic regression whenever the fat-shattering dimension is finite at all scales (Alon et al., 1997). For the purposes of constructing a list learner, it seems plausible that running ERM over an appropriately defined list hypothesis class, which may be some suitable *tensorization* of the given class, does the job. Specifically, we need to construct a class that maps $\mathcal{X}$ to $\mathcal{Y}^k$, where $\mathcal{Y} = [0, 1]$, and prove uniform convergence for such a class. However, despite our efforts, constructing such a class did not appear to be immediately straightforward. A naive $k$-fold Cartesian product of the given class will not work—such a product class contains list functions that are simply $k$ copies of the same function, and hence we cannot hope to prove uniform convergence for the product class if the original class does not allow for uniform convergence.

Before we concretely state the open problem, we define the operation of "flattening" which converts a list hypothesis class to a non-list hypothesis class. This operation is defined in (Hanneke et al., 2024, Remark 11).

**Definition 45 (Flattening a list hypothesis class)** *Given a hypothesis $h : \mathcal{X} \to \mathcal{Y}$ and list-hypothesis $h^{list} : \mathcal{X} \to \mathcal{Y}^k$, we define $h \prec h^{list}$ if $h(x) \in h^{list}(x)$ for every $x \in \mathcal{X}$. Given a list hypothesis class $\mathcal{H}^{list}$, define the flattening operation as follows:*

$$\text{flatten}(\mathcal{H}^{list}) := \{h \in \mathcal{Y}^{\mathcal{X}} : h \prec h^{list} \text{ for some } h^{list} \in \mathcal{H}^{list}\}.$$

We can now formally state the open problem:

**Open Problem 1** *Given any hypothesis class $\mathcal{H} \subseteq [0, 1]^{\mathcal{X}}$ that has finite $(\gamma, k)$-fat shattering dimension for all $\gamma$, can we construct a list hypothesis class $\mathcal{H}^{list} \subseteq ([0, 1]^k)^{\mathcal{X}}$ satisfying the following properties?*

- *$\mathcal{H}^{list}$ contains $\mathcal{H}$: namely, for every $h \in \mathcal{H}$, there exists $h^{list} \in \mathcal{H}^{list}$ satisfying $h \prec h^{list}$.*

- *$\mathcal{H}^{list}$ has finite dimension: namely, flatten$(\mathcal{H}^{list})$ has finite $(\gamma, k)$-fat-shattering dimension for all $\gamma$.*

Presumably, if one can construct a list hypothesis class satisfying the requirements above, then running ERM on such a class might yield an agnostic list learner. As another example, consider the case where $\mathcal{H} = \{0, 1\}^{\mathcal{X}}$. Here, one can simply run ERM on the trivial 2-list class containing a single list hypothesis that maps every $x$ to $\{0, 1\}$. This example is on the other extreme of taking a naive Cartesian product, and suggests that the list hypotheses in the constructed list class should be formed by aggregating hypotheses that exhibit some form of *diversity* in their predictions.

A similar question can also be framed in the classification setup for a hypothesis class on finitely many labels and having a finite $k$-DS dimension. While Hanneke et al. (2024) show that the principle of uniform convergence does continue to hold for list hypothesis classes, their setup assumes that a list hypothesis class (whose flattening has finite $k$-DS dimension) is already given. On the other hand, we assume that we are given a (non-list, standard) hypothesis class to begin with, and wish to construct a list hypothesis class from this, so that we can run ERM on it.

## Acknowledgments

The authors would like to thank Moses Charikar for help with the proof of Claim 48. Chirag is supported by Moses and Gregory Valiant's Simons Investigator Awards.

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

# Appendix A. Proofs from Section 3.1

## A.1. Proof of Lemma 17

We require using the following result from the work of Charikar and Pabbaraju (2023), which states that the one-inclusion graph of a multiclass (total) hypothesis class having small $k$-Natarajan dimension admits a small $k$-outdegree list orientation of its edges. For precise definitions of these quantities, please refer to Section 3 in Charikar and Pabbaraju (2023).

**Theorem 46 (Theorem 8 in Charikar and Pabbaraju (2023))** *Let $\mathcal{H} \in \mathcal{Y}^{\mathcal{X}}$ be a (total) hypothesis class with $k$-Natarajan dimension $\mathbb{D}_k^{\mathrm{Nat}}(\mathcal{H}) < \infty$, and let $|\mathcal{Y}| = p, |\mathcal{X}| < \infty$. Then, there exists a $k$-list orientation $\sigma^k$ of $\mathcal{G}(\mathcal{H})$ with maximum $k$-outdegree*

$$\mathsf{outdeg}^k(\sigma^k) \leq 240 k^4 \mathbb{D}_k^{\mathrm{Nat}}(\mathcal{H}) \ln(p).$$

**Lemma 17 (Weak list learner)** *Consider a partial concept class $\mathcal{H}^{\mathrm{part}} \subseteq \{0, 1, \ldots, k, \star\}^{\mathcal{X}'}$ having $\mathbb{D}_k^{\mathrm{Nat}}(\mathcal{H}^{\mathrm{part}}) < \infty$. There exists a weak list learner $\mathcal{A}_w$ satisfying the following guarantee: for any distribution $\mathcal{D}$ realizable by $\mathcal{H}^{\mathrm{part}}$, for $m \geq 960 k^5 \ln(k+1) \mathbb{D}_k^{\mathrm{Nat}}(\mathcal{H}^{\mathrm{part}})$,*

$$\Pr_{S \sim \mathcal{D}^m} \Pr_{(x', y') \sim \mathcal{D}} \left[ \mu_S^k(x') \not\ni y' \right] < \frac{1}{2(k+1)},$$

*where $\mu_S^k = \mathcal{A}_w(S)$ is the $k$-list hypothesis output by $\mathcal{A}_w$.*

**Proof** Consider the learner $\mathcal{A}_w$ given in Algorithm 3. Fix any $\mathcal{H}^{\mathrm{part}}$-realizable distribution $\mathcal{D}$. For $S \sim \mathcal{D}^m$, let $\mu_S^k = \mathcal{A}_w(S)$ denote the list hypothesis output by $\mathcal{A}_w$ on $S$. We hope to bound from above the probability

$$\Pr_{S \sim \mathcal{D}^m} \Pr_{(x, y) \sim \mathcal{D}} \left[ \mu_S^k(x) \not\ni y \right].$$

---

**Algorithm 3** Weak learner $\mathcal{A}_w$ for $\mathcal{H}^{\text{part}} \subseteq \{0, 1, \ldots, k, \star\}^{\mathcal{X}'}$ based on the one-inclusion graph

---

**Input:** An $\mathcal{H}^{\text{part}}$-realizable sample $S = \big((x_1, y_1), \ldots, (x_m, y_m)\big)$.
**Output:** A $k$-list hypothesis $\mathcal{A}_w(S) = \mu_S^k : \mathcal{X}' \to \{0, 1, \ldots, k\}^k$.

For each $x \in \mathcal{X}'$, the output $h_S(x)$ is computed as follows:

1: Let $\mathcal{X}'_{S,x}$ be the set of all the distinct elements in $\{x_1, \ldots, x_m, x\}$, and consider

$$\mathcal{H}_{S,x}^{\text{tot}} = \big\{ h : \mathcal{X}'_{S,x} \to \{0, 1, \ldots, k\} \mid \{(x, h(x)) : x \in \mathcal{X}'_{S,x}\} \text{ is realizable by } \mathcal{H}^{\text{part}} \big\} .$$

2: **if** $\mathcal{H}_{S,x}^{\text{tot}} = \emptyset$ **then**
3:      Set $\mu_S^k(x) = \{0, 1, \ldots, k-1\}$ arbitrarily.
4: **else**
5:      Find a $k$-list orientation $\sigma^k$ of $\mathcal{G}(\mathcal{H}_{S,x}^{\text{tot}})$ that *minimizes* the *maximum* $k$-outdegree $\text{outdeg}^k(\sigma^k)$.
6:      Consider the edge in the direction of $x$ defined by $S$:

$$e = \{h \in \mathcal{H}_{S,x}^{\text{tot}} : \forall (x_i, y_i) \in S, \ h(x_i) = y_i\}.$$

7:      Set $\mu_S^k(x) = \{h(x) : h \in \sigma^k(e)\}$.
8: **end if**

---

Because $S$ and $(x, y)$ are drawn i.i.d from $\mathcal{D}$, by the leave-one-out argument (Haussler et al., 1994), the above probability is equal to

$$\Pr_{S \sim \mathcal{D}^{m+1}} \Pr_{i \sim \text{Unif}([m+1])} \left[ \mu_{S_{-i}}^k(x_i) \not\ni y_i \right],$$

where $\text{Unif}([m + 1])$ is the uniform distribution on $[m + 1]$, and $\mu_{S_{-i}}^k = \mathcal{A}_w(S \setminus \{(x_i, y_i)\})$. It suffices to upper bound, for every fixed $S = \{(x_1, y_1), \ldots, (x_{m+1}, y_{m+1})\}$ realizable by $\mathcal{H}^{\text{part}}$, the probability

$$\Pr_{i \sim \text{Unif}([m+1])} \left[ \mu_{S_{-i}}^k(x_i) \not\ni y_i \right].$$

For any $i \in [m + 1]$, observe that when Algorithm 3 is instantiated with the input $S_{-i}$, the set $\mathcal{X}'_{S,x}$ and the total class $\mathcal{H}_{S,x}^{\text{tot}}$ it constructs in Line 1 when determining the output $\mu_S^k(x_i)$ are the same. Furthermore, note that since all of $S$ is realizable by $\mathcal{H}^{\text{part}}$, $\mathcal{H}_{S,x}^{\text{tot}} \neq \emptyset$ in Line 2. Thus, the same one-inclusion graph $\mathcal{G}(\mathcal{H}_{S,x}^{\text{tot}})$ and orientation $\sigma^k$ is constructed in Line 5 for every $i \in [m+1]$.

Now, consider the sequence of labels $L = \{y_i\}_{i \leq m+1}$ for every distinct $x_i$ in $S = \{(x_1, y_1), \ldots, (x_{m+1}, y_{m+1})\}$. Because $S$ is realizable by $\mathcal{H}^{\text{part}}$, observe that $L$ constitutes a vertex in $\mathcal{G}(\mathcal{H}_{S,x}^{\text{tot}})$. Denote this common total class by $\mathcal{H}^{\text{tot}}$. Furthermore, observe that $\mu_{S_{-i}}^k(x_i) \not\ni y_i$ iff the list orientation on the edge

adjacent to $L$ in the direction of $x_i$ (call it $e_i$) does not contain $L$. Therefore,

$$\Pr{}_{i \sim \mathrm{Unif}([m+1])} \left[ \mu_{S_{-i}}^k(x_i) \not\ni y_i \right] = \frac{1}{m+1} \sum_{i=1}^{m+1} \mathbb{1}[\sigma^k(e_i) \not\ni L]$$

$$= \frac{\mathsf{outdeg}^k(L; \sigma^k)}{m+1}$$

$$\leq \frac{240 k^4 \mathbb{D}_k^{\mathrm{Nat}}(\mathcal{H}^{\mathrm{tot}}) \ln(k+1)}{m+1}$$

$$\leq \frac{240 k^4 \mathbb{D}_k^{\mathrm{Nat}}(\mathcal{H}^{\mathrm{part}}) \ln(k+1)}{m+1}$$

where the second-to-last inequality follows from Theorem 46, and the last inequality follows because $\mathbb{D}_k^{\mathrm{Nat}}(\mathcal{H}_{S,x}^{\mathrm{tot}}) \leq \mathbb{D}_k^{\mathrm{Nat}}(\mathcal{H}^{\mathrm{part}})$. For the quantity at the end to be less than $\frac{1}{2(k+1)}$, it suffices that

$$\frac{240 k^4 \mathbb{D}_k^{\mathrm{Nat}}(\mathcal{H}^{\mathrm{part}}) \ln(k+1)}{m+1} < \frac{1}{2(k+1)}$$

$$\implies \quad m > 960 k^5 \ln(k+1) \mathbb{D}_k^{\mathrm{Nat}}(\mathcal{H}^{\mathrm{part}}) - 1.$$

∎

### A.2. Proof of Lemma 18

**Lemma 18 (Minimax)** *Let $\mathcal{A}_w$ be the weak learner given by Lemma 17 for the partial concept class $\mathscr{T}_{\gamma,k}(\mathcal{H})$. Let $\tilde{S}$ be the sample constructed by REGREALIZABLE in Line 3 of Algorithm 1. Then, there exist subsequences $T_1, \ldots, T_l$ of $\tilde{S}$ satisfying the condition in Line 4.*

**Proof** First, observe that by construction, the dataset $\tilde{S}$ is realizable by the partial class $\mathscr{T}_{\gamma,k}(\mathcal{H})$.

Now, consider a zero-sum game between two players Max and Minnie, where Max's pure strategies are examples $(x', y') \in \tilde{S}$, and Minnie's pure strategies are subsequences $T$ of $\tilde{S}$ of size $m = 960 k^5 \ln(k+1) \mathbb{D}_{\gamma/2,k}^{\mathrm{fat}}(\mathcal{H})$. For $\mu_T^k = \mathcal{A}_w(T)$, the payoff matrix $M$ is given by $M_{T,(x',y')} = \mathbb{1}[\mu_T^k(x') \not\ni y']$. Let $\mathcal{D}$ be any mixed strategy (i.e., distribution over the examples in $\tilde{S}$) by Max. Note that from Lemma 16, $\mathbb{D}_{\gamma/2,k}^{\mathrm{fat}}(\mathcal{H}) \geq \mathbb{D}_k^{\mathrm{Nat}}(\mathscr{T}_{\gamma,k}(\mathcal{H}))$. Therefore, the sample size $m$ satisfies the condition of Lemma 17, and we have that

$$\Pr{}_{(T,(x',y')) \sim \mathcal{D}^{m+1}}[\mu_T^k(x') \not\ni y'] < \frac{1}{2(k+1)}.$$

But note that the quantity on the LHS above is the expected payoff, when Max plays the mixed strategy $\mathcal{D}$ and Minnie plays the mixed strategy $\mathcal{D}^m$. This means that for any mixed strategy that Max might play, there exists a mixed strategy that Minnie can play which guarantees that the expected payoff is smaller than $\frac{1}{2(k+1)}$. By the Neumann's min-max theorem (Neumann, 1928), there exists a mixed strategy that Minnie can play, such that for every mixed strategy that Max could play (in particular, every pure strategy as well), the expected payoff is smaller than $\frac{1}{2(k+1)}$. This mixed strategy by Minnie is the required distribution $\mathcal{P}$.[10]

---

10. Note that $\mathcal{P}$ can be computed by solving a linear program.

Then, for the independent sequences $T_1, \ldots, T_l$ drawn from $\mathcal{P}$, by a Chernoff bound, where $l = 6(k+1)\ln(2n')$ we have that for every fixed $(x', y') \in \tilde{S}$,

$$\frac{1}{l}\sum_{i=1}^{l}\left[\mathbb{1}[\mu_i^k(x') \not\ni y'] \geq \frac{1}{k+1}\right] \leq \exp\left(-\frac{l}{6(k+1)}\right) < \frac{1}{n'}.$$

A union bound over the $n'$ samples in $\tilde{S}$ ensures the existence of $T_1, \ldots, T_l$ such that for all $(x', y') \in \tilde{S}$ simultaneously, it holds that

$$\sum_{i=1}^{l}\mathbb{1}\left[\mu_i^k(x') \ni y'\right] > \frac{kl}{k+1}.$$

But this necessarily means that $y'$ belongs to the $k$ most frequently occurring labels in the lists $\mu_1^k(x'), \ldots, \mu_l^k(x')$. Because if not, there are $k$ other labels, each occuring in strictly more than $\frac{kl}{k+1}$ lists. Given that $y'$ itself occurs in more than $\frac{kl}{k+1}$ lists, this accounts for more than $k \cdot \frac{kl}{k+1} + \frac{kl}{k+1} = kl$ occurences of labels, which is not possible with $l$ lists of size $k$. Thus, Top-$k$ at the end of Line 4 picks up the correct label simultaneously for every $(x', y') \in \tilde{S}$. ∎

## A.3. Proof of Claim 20

**Claim 20 (Scaled sum)** *Consider $a \in \mathcal{D}_\gamma$ and $k$-threshold $\tau \in \mathcal{D}_\gamma^{(k)}$. Then, we have*

$$a = \gamma \binom{\lfloor\frac{1}{\gamma}\rfloor}{k-1}^{-1} \sum_{\tau \in \mathcal{D}_\gamma^{(k)}} \mathfrak{f}(a, \tau).$$

**Proof** Let us partition $\mathcal{D}_\gamma^{(k)}$ into disjoint sets $\mathcal{D}_\gamma^{(k,r)}$ for every $r \in \{0, 1, \ldots, k\}$, with each set $\mathcal{D}_\gamma^{(k,r)}$ denoting a subset of $\mathcal{D}_\gamma^{(k)}$ such that exactly $r$ components of $\tau$ are smaller than $a$. Formally,

$$\mathcal{D}_\gamma^{(k,r)} := \{\tau : \tau \in \mathcal{D}_\gamma^{(k)} \text{ and } |\{j : \tau_j < a\}| = r\}. \tag{15}$$

Note that $\mathfrak{f}(a, \tau) = r$ for $\tau \in \mathcal{D}_\gamma^{(k,r)}$, and that

$$\left|\mathcal{D}_\gamma^{(k,r)}\right| = \binom{\frac{a}{\gamma}}{r}\binom{\lfloor\frac{1}{\gamma}\rfloor - \frac{a}{\gamma} + 1}{k-r}.$$

Then, we have

$$\sum_{\tau \in \mathcal{D}_\gamma^{(k)}} \mathfrak{f}(a, \tau) = \sum_{r=0}^{k}\sum_{\tau \in \mathcal{D}_\gamma^{(k,r)}} \mathfrak{f}(a, \tau) = \sum_{r=0}^{k} r\binom{\frac{a}{\gamma}}{r}\binom{\lfloor\frac{1}{\gamma}\rfloor + 1 - \frac{a}{\gamma}}{k-r} \tag{16}$$

$$= \frac{a}{\gamma}\sum_{r=0}^{k}\binom{\frac{a}{\gamma} - 1}{r-1}\binom{\lfloor\frac{1}{\gamma}\rfloor + 1 - \frac{a}{\gamma}}{k-r} = \frac{a}{\gamma}\binom{\lfloor\frac{1}{\gamma}\rfloor}{k-1} \tag{17}$$

The last equality follows since the sum is the coefficient of $x^{k-1}$ in $(1+x)^{\frac{a}{\gamma}-1}(1+x)^{\lfloor\frac{1}{\gamma}\rfloor+1-\frac{a}{\gamma}}$. ∎

### A.4. Proof of Lemma 22

**Lemma 22** *Let $\mathcal{H} \subseteq [0,1]^{\mathcal{X}}$ be a hypothesis class having $\mathbb{D}^{\text{fat}}_{\gamma/2,k}(\mathcal{H}) < \infty$, and let $\mathcal{A}_w$ be the weak learner for $\mathscr{T}_{\gamma,k}(\mathcal{H})$ given by Lemma 17. Let $S = \{(x_i, y_i)\}_{i \in [n]} \in (\mathcal{X} \times [0,1])^n$ be a sample realizable by $\mathcal{H}$, and consider invoking $\text{REGREALIZABLE}(S, \mathcal{A}_w, \gamma)$. Then, there exist indexing functions $\mathfrak{j}_1, \ldots, \mathfrak{j}_n : \mathcal{D}^{(k)}_\gamma \to [k]$ that index into the predictions of $J$ on $S$ (Line 4), such that*

(a) $[J(x_i, \tau)]_{\mathfrak{j}_i(\tau)} = \mathfrak{f}(y_i, \tau), \forall \tau \in \mathcal{V}(y_i, \gamma), \forall i \in [n]$.

(b) $\left| \min\left( 1, \gamma \left( \frac{\lfloor \frac{1}{\gamma} \rfloor}{k-1} \right)^{-1} \sum_{\tau \in \mathcal{D}^{(k)}_\gamma} [J(x_i, \tau)]_{\mathfrak{j}_i(\tau)} \right) - y_i \right| \le (2k+1)\gamma \; \forall i \in [n]$.

**Proof** Observe that for any $i \in [n]$, $\tau \in \mathcal{V}(y_i, \gamma)$, $y'_{i,\tau} \ne \star$. Thus, $((x_i, \tau), y'_{i,\tau})$ will be included in the dataset $\tilde{S}$. Note also that $y'_{i,\tau} = \mathfrak{f}(y_i, \tau)$. From Lemma 18, we then have that $J(x_i, \tau) \ni \mathfrak{f}(y_i, \tau)$. Let $\mathfrak{j}_i : \mathcal{D}^{(k)}_\gamma \to [k]$ be the indexing function such that

$$\mathfrak{j}_i(\tau) = \begin{cases} j \text{ where } j \in [k] \text{ is such that } [J(x_i, \tau)]_j = \mathfrak{f}(y_i, \tau) & \text{if } \tau \in \mathcal{V}(y_i, \gamma) \\ 1 & \text{otherwise.} \end{cases} \quad (18)$$

Lemma 22 then follows by construction.

For Lemma 22, define the discretization $\hat{y}_i = \gamma \lfloor \frac{y_i}{\gamma} \rfloor$. From Claim 20, we know that

$$\hat{y}_i = \gamma \left( \frac{\lfloor \frac{1}{\gamma} \rfloor}{k-1} \right)^{-1} \sum_{\tau \in \mathcal{D}^{(k)}_\gamma} \mathfrak{f}(\hat{y}_i, \tau). \quad (19)$$

Then, observe that for any $i \in [n]$,

$$\left| \left( \gamma \left( \frac{\lfloor \frac{1}{\gamma} \rfloor}{k-1} \right)^{-1} \sum_{\tau \in \mathcal{D}^{(k)}_\gamma} [J(x_i, \tau)]_{\mathfrak{j}_i(\tau)} \right) - y_i \right| \le \left| \left( \gamma \left( \frac{\lfloor \frac{1}{\gamma} \rfloor}{k-1} \right)^{-1} \sum_{\tau \in \mathcal{D}^{(k)}_\gamma} [J(x_i, \tau)]_{\mathfrak{j}_i(\tau)} \right) - \hat{y}_i \right| + \underbrace{|\hat{y}_i - y_i|}_{\le \gamma}$$

$$\le \gamma + \gamma \left( \frac{\lfloor \frac{1}{\gamma} \rfloor}{k-1} \right)^{-1} \sum_{\tau \in \mathcal{D}^{(k)}_\gamma} \left| [J(x_i, \tau)]_{\mathfrak{j}_i(\tau)} - \mathfrak{f}(\hat{y}_i, \tau) \right| \qquad \text{(using (19))}$$

$$\overset{(a)}{\le} \gamma + k\gamma \left( \frac{\lfloor \frac{1}{\gamma} \rfloor}{k-1} \right)^{-1} \sum_{\tau \in \mathcal{D}^{(k)}_\gamma} \mathbb{1}\left[ [J(x_i, \tau)]_{\mathfrak{j}_i(\tau)} \ne \mathfrak{f}(\hat{y}_i, \tau) \right]$$

$$\le \gamma + k\gamma \left( \frac{\lfloor \frac{1}{\gamma} \rfloor}{k-1} \right)^{-1} \left[ \sum_{\substack{\tau' \in \mathcal{D}^{(k)}_\gamma: \\ \min_{j \in [k]} |y_i - \tau'_j| \ge \frac{\gamma}{2}}} \mathbb{1}\left[ [J(x_i, \tau')]_{\mathfrak{j}_i(\tau')} \ne \mathfrak{f}(\hat{y}_i, \tau') \right] + \left| \left\{ \tau' \in \mathcal{D}^{(k)}_\gamma : \min_{j \in [k]} |y_i - \tau'_j| < \frac{\gamma}{2} \right\} \right| \right]$$

$$\overset{(b)}{\leq} \gamma + k\gamma \binom{\left\lfloor \frac{1}{\gamma} \right\rfloor}{k-1}^{-1} \left[ \sum_{\substack{\tau' \in \mathcal{D}_\gamma^{(k)}: \\ \min_{j \in [k]} |y_i - \tau'_j| \geq \frac{\gamma}{2}}} \mathbb{1}\left[ [J(x_i, \tau')]_{j_i(\tau')} \neq \mathfrak{f}(\hat{y}_i, \tau') \right] + \binom{\left\lfloor \frac{1}{\gamma} \right\rfloor}{k-1} \right]$$

$$\overset{(c)}{\leq} \gamma + 2k\gamma \binom{\left\lfloor \frac{1}{\gamma} \right\rfloor}{k-1}^{-1} \binom{\left\lfloor \frac{1}{\gamma} \right\rfloor}{k-1} = (2k+1)\gamma.$$

$(a)$ follows because: since $\mathfrak{f}(\hat{y}_i, \tau)$ and all elements in $J(x_i, \tau)$ are in $\{0, 1, \ldots, k\}$ for every $\tau \in \mathcal{D}_\gamma^{(k)}$,

$$\left| [J(x_i, \tau)]_{j_i(\tau)} - \mathfrak{f}(\hat{y}_i, \tau) \right| \leq k \cdot \mathbb{1}[[J(x_i, \tau)]_{j(\tau)} \neq \mathfrak{f}(\hat{y}_i, \tau)].$$

$(b)$ follows because: the cardinality of the set $\left\{ \tau' \in \mathcal{D}_\gamma^{(k)} : \min_{j \in [k]} |y_i - \tau'_j| < \frac{\gamma}{2} \right\}$ is at most $\binom{\left\lfloor \frac{1}{\gamma} \right\rfloor}{k-1}$, since any $\tau'$ satisfying the condition must have one of its components (at one of $k$ indices) fixed to an integer multiple of $\gamma$ in $(y_i - \gamma/2, y_i + \gamma/2)$.

$(c)$ follows because: for $\tau' \in \mathcal{D}_\gamma^{(k)}$ satisfying $\min_{j \in [k]} |y_i - \tau'_j| \geq \frac{\gamma}{2}$, by our choice of $j_i$, we know that $[J(x_i, \tau')]_{j_i(\tau')} = \mathfrak{f}(y_i, \tau')$. Thus, if $[J(x_i, \tau')]_{j_i(\tau')} \neq \mathfrak{f}(\hat{y}_i, \tau')$, then this would imply that $\mathfrak{f}(y_i, \tau') \neq \mathfrak{f}(\hat{y}_i, \tau')$. Since $y_i \geq \hat{y}_i$ and $|y_i - \hat{y}_i| \leq \gamma$, this necessarily means that one of the components of $\tau'$ is $\hat{y}_i$. We can then bound the number of such $\tau'$ as $\binom{\left\lfloor \frac{1}{\gamma} \right\rfloor}{k-1}$.

Finally, observe that since $y_i \in [0, 1]$ and $\gamma \binom{\left\lfloor \frac{1}{\gamma} \right\rfloor}{k-1}^{-1} \sum_{\tau \in \mathcal{D}_\gamma^{(k)}} [J(x_i, \tau)]_{j_i(\tau)} \geq 0$,

$$\left| \min\left( 1, \gamma \binom{\left\lfloor \frac{1}{\gamma} \right\rfloor}{k-1}^{-1} \sum_{\tau \in \mathcal{D}_\gamma^{(k)}} [J(x_i, \tau)]_{j_i(\tau)} \right) - y_i \right| \leq \left| \left( \gamma \binom{\left\lfloor \frac{1}{\gamma} \right\rfloor}{k-1}^{-1} \sum_{\tau \in \mathcal{D}_\gamma^{(k)}} [J(x_i, \tau)]_{j_i(\tau)} \right) - y_i \right|,$$

which completes the proof of Lemma 22. ∎

### A.5. Proof of Claim 23

**Claim 23 (Existence of cover)** *There always exists a subset of $\mathcal{C}$ of at most $k$ points that $r$-covers $\mathcal{C}$ in Line 9 of Algorithm 2 when it is called with $r = 6k\gamma + 3\gamma$.*

**Proof** Suppose that there does not exist a subset of at most $k$ points that $r$-covers $\mathcal{C}$, which means that the internal covering number of $\mathcal{C}$ at radius $r$ is strictly larger than $k$. This means that the internal packing number of $\mathcal{C}$ at radius $r$ is strictly larger than $k$ (since the internal covering number is at most the internal packing number). Alternately, there must exist a set of $k + 1$ points in $\mathcal{C}$ that is pairwise separated by more than $r$: let $\{\hat{c}_1, \ldots, \hat{c}_{k+1}\} \subseteq \mathcal{C}$ be such a set. Note that by way of how $\mathcal{C}$ was constructed, this means that there exists a set of numbers $\{c_1, \ldots, c_{k+1}\}$, which satisfies that, for all $i \in \{1, \ldots, k + 1\}$,

(1) $|\hat{c}_i - c_i| \leq \frac{r}{3}$.

(2) For some function $j_i : \mathcal{D}_\gamma^{(k)} \to [k]$, $[\mathcal{J}(\tau)]_{j_i(\tau)} = \mathfrak{f}(c_i, \tau)$, $\forall \tau \in \mathcal{V}(c_i, \gamma)$.

(1) above implies that $c_1, \ldots, c_{k+1}$ are pairwise separated by an amount strictly larger than $r - 2r/3 = 2k\gamma + \gamma \geq 3\gamma$. Assume without loss of generality that $c_1 < \cdots < c_{k+1}$, and construct $\tau \in \mathcal{D}_\gamma^{(k)}$ such that $\tau_i = \gamma \left\lfloor \frac{c_i + c_{i+1}}{2\gamma} \right\rfloor$ for every $i \in [k]$. Observe that from the preceding argument, $\frac{c_i + c_{i+1}}{2}$ is at least $3\gamma/2$ away from both $c_i$ and $c_{i+1}$, and therefore, $\tau_i = \gamma \left\lfloor \frac{c_i + c_{i+1}}{2\gamma} \right\rfloor$ is at least $\gamma/2$ away from both these numbers as well. In particular, this implies that $\tau \in \mathcal{V}(c_i, \gamma)$ for every $i \in [k+1]$.

Now, consider the set of numbers $\{[\mathcal{J}(\tau)]_{j_1(\tau)}, \ldots, [\mathcal{J}(\tau)]_{j_{k+1}(\tau)}\}$ for the $\tau$ constructed above. Since $\mathcal{J}(\tau)$ is a $k$-list, this set has at most $k$ distinct numbers. But since $\tau \in \mathcal{V}(c_i, \gamma)$ for every $i \in [k+1]$, from (2) above, this set is the same set as $\{\mathfrak{f}(c_1, \tau), \ldots, \mathfrak{f}(c_{k+1}, \tau)\}$. But note that by construction, $\mathfrak{f}(c_i, \tau) = i - 1$, which implies that all the $k+1$ numbers in $\{\mathfrak{f}(c_1, \tau), \ldots, \mathfrak{f}(c_{k+1}, \tau)\}$ are distinct—a contradiction. Thus, there must exist a subset of $\leq k$ points that $r$-cover $\mathcal{C}$. ∎

## A.6. Proof of Theorem 12 and Corollary 13

**Theorem 12 (Finite $(\gamma, k)$-fat-shattering dimension is sufficient)** *Let $\mathcal{H} \subseteq [0, 1]^\mathcal{X}$ be a hypothesis class. For any distribution $\mathcal{D}$ over $\mathcal{X} \times [0, 1]$ and $\gamma \in (0, 1)$, there exists a list learning algorithm $\mathcal{A}$ which takes as input an i.i.d. sample $S \sim \mathcal{D}^n$ and outputs a $k$-list hypothesis $\mathcal{A}(S)$ satisfying*

$$\mathrm{err}_{\mathcal{D}, \ell_{abs}}(\mathcal{A}(S)) \leq \inf_{h \in \mathcal{H}} \mathrm{err}_{\mathcal{D}, \ell_{abs}}(h) + O(k) \cdot \gamma + \widetilde{O}\left(\sqrt{\frac{k^8 \, \mathbb{D}_{\gamma/2, k}^{\mathrm{fat}}(\mathcal{H}) + \log(1/\delta)}{n}}\right)$$

*with probability at least $1 - \delta$.*[11] *In other words, $m_{\mathcal{A}, \mathcal{H}}^{k, \mathrm{ag}}(\varepsilon, \delta) \leq \widetilde{O}\left(\frac{k^8 \, \mathbb{D}_{\gamma/2, k}^{\mathrm{fat}}(\mathcal{H}) + \log(1/\delta)}{\varepsilon^2}\right)$ for $\gamma = \frac{\varepsilon}{O(k)}$.*

**Proof** We will show that the output of REGAGNOSTIC$(S, \mathcal{A}_w, \gamma)$, where $\mathcal{A}_w$ is the weak learner from Lemma 17 satisfies the requirements of the theorem. Let $H : \mathcal{X} \mapsto [0, 1]^k$ denote this output defined by $x \mapsto$ REGAGNOSTIC$(S, \mathcal{A}_w, \gamma)$, which is further given by $x \mapsto$ REGREALIZABLE$(\bar{S}, \mathcal{A}_w, \gamma)$ where $\bar{S}$ is defined in Line 9 of Algorithm 1.

We claim that the execution of REGAGNOSTIC can be instantitated as a sample compression scheme for $(\mathcal{X}, [0, 1])$. For any $S = \{(x_i, y_i)\}_{i \in [n]} \in (\mathcal{X} \times [0, 1])^n$, consider the sample $\bar{S}$ constructed by REGAGNOSTIC in Line 9. Then, let $T_1, \ldots, T_l$ be the subsequences that REGREALIZABLE$(\bar{S}, \mathcal{A}_w, \gamma)$ constructs (after first preparing $\tilde{S}$ from $\bar{S}$) in Line 4. Note that for $t \in [m]$, the $t^{\mathrm{th}}$ example in $T_j$ can be written as $((x_{i_{j,t}}, \tau_{j,t}), b_{j,t})$ for $i_{j,t} \in [n], \tau_{j,t} \in \mathcal{D}_\gamma^{(k)}, b_{j,t} \in \{0, 1, \ldots, k\}$. Let $S_j = \{(x_{i_{j,t}}, y_{i_{j,t}})\}$. We then define $(\kappa, \rho)$ to be the following list sample compression scheme:

- The compression function $\kappa$ maps $S$ to $\kappa(S) = (S_1, \ldots, S_l), ((\tau_{j,t})_{t \in [m]})_{j \in [l]}$.

- The reconstruction function $\rho$ first constructs the labels $b_{j,t} \in \{0, 1 \ldots, k\}$ from each $(x_{i_{j,t}}, y_{i_{j,t}})$ and $\tau_{j,t}$, just as in Line 2. At this point, it has effectively reconstructed the sequences

---

11. Here, $\widetilde{O}$ hides polylog factors in $n, k$ and $1/\gamma$.

$T_1, \ldots, T_l$. It then invokes $\mathcal{A}_w$ on each of these to obtain $\mu_1^k = \mathcal{A}_w(T_1), \ldots, \mu_l^k = \mathcal{A}_w(T_l)$ and thereafter uses these to construct the list map $H : \mathcal{X} \to [0,1]^k$, which it outputs.

Note that each $\tau_{j,t}$ can be encoded with $k \log(1/\gamma)$ bits (the parameter $\gamma \in (0,1)$ is known to $\rho$, so we only need to encode indices into the set $\mathcal{D}_\gamma^{(k)}$). Therefore, the size of the compression is

$$|\kappa(S| = ml + mlk \log(1/\gamma) = O\left(k^7 \log(k) \log(n') \log(1/\gamma) \mathbb{D}_{\gamma/2,k}^{\mathrm{fat}}(\mathcal{H})\right)$$
$$= O\left(k^8 \log(k) \log(n) \log^2(1/\gamma) \mathbb{D}_{\gamma/2,k}^{\mathrm{fat}}(\mathcal{H})\right).$$

Let $\mathcal{A}$ be the learning algorithm that outputs $H = \rho(\kappa(S))$. Then, Lemma 10 guarantees that with probability $1 - \delta/2$ over $S \sim \mathcal{D}^n$,

$$\mathrm{err}_{\mathcal{D},\ell_{\mathrm{abs}}}(H) \leq \hat{\mathrm{err}}_{S,\ell_{\mathrm{abs}}}(H) + O\left(\sqrt{\frac{k^8 \, \mathbb{D}_{\gamma/2,k}^{\mathrm{fat}}(\mathcal{H}) \, \mathrm{polylog}(n,k,1/\gamma) + \log(1/\delta)}{n}}\right). \tag{20}$$

Here, we used that $\hat{\mathrm{err}}_{S\setminus\kappa(S),\ell_{\mathrm{abs}}}(H) \leq 1$. Now note that

$$\hat{\mathrm{err}}_{S,\ell_{\mathrm{abs}}}(H) = \frac{1}{n} \sum_{i=1}^n \left|\min_{j\in[k]} H(x_i)_j - y_i\right|$$
$$\leq \frac{1}{n} \sum_{i=1}^n \left|\min_{j\in[k]} H(x_i)_j - \hat{y}_i\right| + \frac{1}{n} \sum_{i=1}^n |\hat{y}_i - y_i|$$
$$= \hat{\mathrm{err}}_{\bar{S},\ell_{\mathrm{abs}}}(H) + \hat{\mathrm{err}}_{S,\ell_{\mathrm{abs}}}(h^\star)$$
$$\leq \hat{\mathrm{err}}_{\bar{S},\ell_{\mathrm{abs}}}(H) + \inf_{h\in\mathcal{H}} \hat{\mathrm{err}}_{S,\ell_{\mathrm{abs}}}(h) + \gamma, \tag{21}$$

where the last two lines follow by our choice of the labels $\hat{y}_i$ according to $h^\star$ in Line 9. Furthermore, since $\bar{S}$ constructed by REGAGNOSTIC in Line 9 is realizable by $\mathcal{H}$, Lemma 24 guarantees that

$$\hat{\mathrm{err}}_{\bar{S},\ell_{\mathrm{abs}}}(H) \leq (8k+4)\gamma. \tag{22}$$

Combining (20), (21) and (22), we get that with probability at least $1 - \delta/2$,

$$\mathrm{err}_{\mathcal{D},\ell_{\mathrm{abs}}}(H) \leq \inf_{h\in\mathcal{H}} \hat{\mathrm{err}}_{S,\ell_{\mathrm{abs}}}(h) + (8k+5)\gamma \tag{23}$$

$$+ O\left(\sqrt{\frac{k^8 \, \mathbb{D}_{\gamma/2,k}^{\mathrm{fat}}(\mathcal{H}) \, \mathrm{polylog}(n,k,1/\gamma) + \log(1/\delta)}{n}}\right). \tag{24}$$

Now, note that by definition of the infimum, $\exists h' \in \mathcal{H}$ satisfying

$$\mathrm{err}_{\mathcal{D},\ell_{\mathrm{abs}}}(h') \leq \inf_{h\in\mathcal{H}} \mathrm{err}_{\mathcal{D},\ell_{\mathrm{abs}}}(h) + \sqrt{\frac{\log(2/\delta)}{2n}}. \tag{25}$$

Furthermore, for this $h'$, by Hoeffding's inequality, we have that with probability at least $1 - \delta/2$ over $S \sim \mathcal{D}^n$,

$$\hat{\mathrm{err}}_{S,\ell_{\mathrm{abs}}}(h') \leq \mathrm{err}_{\mathcal{D},\ell_{\mathrm{abs}}}(h') + \sqrt{\frac{\log(2/\delta)}{2n}}. \tag{26}$$

Combining (25) and (26), we get that with probability at least $1 - \delta/2$ over $S \sim \mathcal{D}^n$,

$$\inf_{h \in \mathcal{H}} \hat{\text{err}}_{S,\ell_{\text{abs}}}(h) \leq \hat{\text{err}}_{S,\ell_{\text{abs}}}(h') \leq \inf_{h \in \mathcal{H}} \text{err}_{\mathcal{D},\ell_{\text{abs}}}(h) + 2\sqrt{\frac{\log(2/\delta)}{2n}}. \tag{27}$$

∎

**Corollary 13 (Realizable list regression $(\gamma, k)$-fat-shattering upper bound)** *Let $\mathcal{H} \subseteq [0,1]^{\mathcal{X}}$ be a hypothesis class. For any distribution $\mathcal{D}$ over $\mathcal{X} \times [0,1]$ realizable by $\mathcal{H}$ and $\gamma \in (0,1)$, there exists a list learning algorithm $\mathcal{A}$ which takes as input an i.i.d. sample $S \sim \mathcal{D}^n$ and outputs a $k$-list hypothesis $\mathcal{A}(S)$ satisfying*

$$\text{err}_{\mathcal{D},\ell_{abs}}(\mathcal{A}(S)) \leq O(k) \cdot \gamma + \widetilde{O}\left(\frac{k^8 \, \mathbb{D}^{\text{fat}}_{\gamma/2,k}(\mathcal{H}) + \log(1/\delta)}{n}\right)$$

*with probability at least $1 - \delta$. In other words, $m^{k,\text{re}}_{\mathcal{A},\mathcal{H}}(\varepsilon,\delta) \leq \widetilde{O}\left(\frac{k^8 \, \mathbb{D}^{\text{fat}}_{\gamma/2,k}(\mathcal{H}) + \log(1/\delta)}{\varepsilon}\right)$ for $\gamma = \frac{\varepsilon}{O(k)}$.*

**Proof** We will show that the output of REGREALIZABLE$(S, \mathcal{A}_w, \gamma)$, where $\mathcal{A}_w$ is the weak learner from Lemma 17 satisfies the requirements of the theorem. As discussed in the proof of Theorem 12, let $(\kappa, \rho)$ be the sample compression scheme representing the execution of REGREALIZABLE. Recall that the size of the compression is $O\left(k^8 \log(k) \log(n) \log^2(1/\gamma) \mathbb{D}^{\text{fat}}_{\gamma/2,k}(\mathcal{H})\right)$. Let $\mathcal{A}$ be the learning algorithm that outputs $H = \rho(\kappa(S))$. Lemma 10 guarantees that with probability at least $1 - \delta$ over $S$, for some constant $C \geq 1$,

$$\text{er}_{\mathcal{D},\ell_{\text{abs}}}(H) \leq \hat{\text{err}}_{S,\ell_{\text{abs}}}(H) + C \cdot \left(\sqrt{\hat{\text{err}}_{S\setminus\kappa(S),\ell_{\text{abs}}}(H) \cdot \frac{k^8 \, \mathbb{D}^{\text{fat}}_{\gamma/2,k}(\mathcal{H}) \, \text{polylog}(n,k,1/\gamma) + \log(1/\delta)}{n}}\right)$$

$$+ C \cdot \left(\frac{k^8 \, \mathbb{D}^{\text{fat}}_{\gamma/2,k}(\mathcal{H}) \, \text{polylog}(n,k,1/\gamma) + \log(1/\delta)}{n}\right)$$

$$\leq \hat{\text{err}}_{S,\ell_{\text{abs}}}(H) + \frac{|\kappa(S)|}{n} + C \cdot \left(\sqrt{\hat{\text{err}}_{S\setminus\kappa(S),\ell_{\text{abs}}}(H) \cdot \frac{k^8 \, \mathbb{D}^{\text{fat}}_{\gamma/2,k}(\mathcal{H}) \, \text{polylog}(n,k,1/\gamma) + \log(1/\delta)}{n}}\right)$$

$$+ C \cdot \left(\frac{k^8 \, \mathbb{D}^{\text{fat}}_{\gamma/2,k}(\mathcal{H}) \, \text{polylog}(n,k,1/\gamma) + \log(1/\delta)}{n}\right).$$

Now, since the range of $H$ and the labels in $S$ are in $[0,1]$, observe that

$$\hat{\text{err}}_{S\setminus\kappa(S),\ell_{\text{abs}}}(H) \leq \hat{\text{err}}_{S,\ell_{\text{abs}}}(H) + \frac{|\kappa(S)|}{n}.$$

Substituting in the above, this gives us

$$\text{er}_{\mathcal{D},\ell_{\text{abs}}}(H) \leq t + C\sqrt{t\Delta} + C\Delta,$$

where $t = \widehat{\mathrm{err}}_{S,\ell_{\mathrm{abs}}}(H) + \frac{|\kappa(S)|}{n}$, $\Delta = \frac{k^8 \, \mathbb{D}^{\mathrm{fat}}_{\gamma/2,k}(\mathcal{H}) \, \mathrm{polylog}(n,k,1/\gamma) + \log(1/\delta)}{n}$. Using the AM-GM inequality, we have that $\sqrt{C^2 \Delta \cdot t} \leq \frac{C^2 \Delta + t}{2}$, which gives us that

$$\mathrm{er}_{\mathcal{D},\ell_{\mathrm{abs}}}(H) \leq 2t + 2C^2 \Delta.$$

Substituting back the values of $t$ and $\Delta$, and recalling that $C$ is a constant, we get that

$$\mathrm{er}_{\mathcal{D},\ell_{\mathrm{abs}}}(H) \leq 2\widehat{\mathrm{err}}_{S,\ell_{\mathrm{abs}}}(H) + 2\frac{|\kappa(S)|}{n} + \widetilde{O}\left(\frac{k^8 \, \mathbb{D}^{\mathrm{fat}}_{\gamma/2,k}(\mathcal{H}) + \log(1/\delta)}{n}\right)$$

$$= 2\widehat{\mathrm{err}}_{S,\ell_{\mathrm{abs}}}(H) + \widetilde{O}\left(\frac{k^8 \, \mathbb{D}^{\mathrm{fat}}_{\gamma/2,k}(\mathcal{H}) + \log(1/\delta)}{n}\right).$$

Finally, Lemma 24 guarantees that $\widehat{\mathrm{err}}_{S,\ell_{\mathrm{abs}}}(H) \leq (8k+4)\gamma$, which completes the proof. ∎

## Appendix B. Proofs from Section 3.2

### B.1. Proof of Lemma 28

**Lemma 28 (Finite $(\gamma,k)$-strong-fat-shattering dimension necessary)** *Let $\mathcal{H} \subseteq [0,1]^{\mathcal{X}}$ be a hypothesis class having $(\gamma,k)$-strong-fat-shattering dimension $\mathbb{D}^{\mathrm{sfat}}_{\gamma,k}(\mathcal{H})$, and let $\mathcal{A}$ be any $k$-list regression algorithm for $\mathcal{H}$ in the agnostic setting. Then, for any $\varepsilon, \delta \in (0,1)$ satisfying $\varepsilon, \delta \leq \frac{\gamma}{2(k+1)}$,*

$$m^{k,\mathrm{ag}}_{\mathcal{A},\mathcal{H}}(\varepsilon, \delta) \geq \Omega\left(\mathbb{D}^{\mathrm{sfat}}_{\gamma,k}(\mathcal{H})\right).$$

**Proof** Fix $\varepsilon, \delta \leq \frac{\gamma}{2(k+1)}$. Let $\mathbb{D}^{\mathrm{sfat}}_{\gamma,k}(\mathcal{H}) = d$ for ease of notation, and suppose it were the case that $m^{k,\mathrm{ag}}_{\mathcal{A},\mathcal{H}}(\varepsilon, \delta) < d/2$. Then, given a sample $S$ of size $m = d/2$ drawn i.i.d. from any distribution $D$ on $\mathcal{X} \times [0,1]$, with probability at least $1 - \delta$ over the draw of $S$, the output list hypothesis $\mu^k_S = \mathcal{A}(S)$ satisfies

$$\mathbb{E}_{(x,y)\sim D}[\min_l |\mu^k_S(x)_l - y|] \leq \inf_{f \in \mathcal{H}} \mathbb{E}_{(x,y)\sim D}[|f(x) - y|] + \varepsilon.$$

In particular, we have that

$$\mathbb{E}_{S\sim D^m} \mathbb{E}_{(x,y)\sim D}[\min_l |\mu^k_S(x)_l - y|] \leq \inf_{f \in \mathcal{H}} \mathbb{E}_{(x,y)\sim D}[|f(x) - y|] + \varepsilon + \delta$$

$$\leq \inf_{f \in \mathcal{H}} \mathbb{E}_{(x,y)\sim D}[|f(x) - y|] + \frac{\gamma}{k+1}. \qquad (28)$$

Now, let $T = (x_1, \ldots, x_d) \in \mathcal{X}^d$ be a sequence that is $(\gamma,k)$-strongly-fat-shattered by $\mathcal{H}$ according to vectors $c_1, \ldots, c_d \in [0,1]^{k+1}$. Let $F = \{f_b \in \mathcal{H} : b \in [k+1]^d\}$ be the set of $(k+1)^d$ functions that realize this shattering with respect to $c_1, \ldots, c_d$ (see Definition 27).

For each $b \in [k+1]^d$, let $D_b$ be the distribution on $\mathcal{X} \times [0,1]$ such that the marginal distribution of the first component is the uniform distribution on the shattered sequence $T$, and the distribution

of the second component conditioned on the first component being $x$ has all its mass on $f_b(x)$. Concretely,

$$D_b(x, y) = \begin{cases} \frac{1}{d} & \text{if } x \in T, \ y = f_b(x), \\ 0 & \text{otherwise.} \end{cases}$$

Note that for any $D_b$, $\inf_{f \in \mathcal{H}} \mathbb{E}_{(x,y) \sim D_b}[|f(x) - y|] = 0$ and is attained by $f_b$. We will show that there is some $D_b$ for which $\mathbb{E}_{(x,y) \sim D_b}[\min_l |\mu_S^k(x)_l - y|]$ is large.

Consider the random experiment of drawing $b$ uniformly at random from $[k+1]^d$, and then drawing a sample $S$ of size $m$ i.i.d. from $D_b$, and then measuring the error of $\mu_S^k$ with respect to $D_b$. Conditioned on any $S = \{(x_{i_1}, y_{i_1}), \ldots, (x_{i_m}, y_{i_m})\}$, observe that $b_{i_1}, \ldots, b_{i_m}$ get determined from $S$. However, the values $b_j$ for $j \notin \{i_1, \ldots, i_m\}$ are still uniform in $[k+1]$.[12] Furthermore, $\mu_S^k$ depends only on $S$, and is independent of the random choice of these $b_j$ values. Thus, for each $x_j$ such that $j \notin \{i_1, \ldots, i_m\}$, fixing all the other randomness (including internal randomness of $\mathcal{A}$) and considering only the randomness in $b_j$, the expected error of $\mu_S^k$ is

$$\frac{1}{k+1} \left( \min_l |\mu_S^k(x_j)_l - c_{j,1}| + \cdots + \min_l |\mu_S^k(x_j)_l - c_{j,k+1}| \right).$$

But note that the number of terms in the summation is $k+1$, so by the pigeonhole principle, the same $l$ attains the minimum at some two summands. Thus, for this $l$, the above quantity is at least

$$\frac{1}{k+1} \left( |\mu_S^k(x_j)_l - c_{j,a}| + |\mu_S^k(x_j)_l - c_{j,b}| \right) \geq \frac{1}{k+1} |c_{j,a} - c_{j,b}| \geq \frac{2\gamma}{k+1}. \tag{29}$$

Factoring in the randomness of $\mathcal{A}$ and the draw of $x_j$ from $D_b$, we get that the expected error of $\mu_S^k$ over the draw of $b$ and $x_j$ is at least

$$\left( 1 - \frac{m}{d} \right) \cdot \frac{2\gamma}{k+1} > \frac{\gamma}{k+1}.$$

The above bound holds for every fixed $S$, and hence it also holds in expectation over $S$. In particular, this implies that there exists a $b \in [k+1]^d$ (and a corresponding $D_b$) for which

$$\mathbb{E}_{S \sim D_b^m} \mathbb{E}_{(x,y) \sim D_b} \left[ \min_l |\mu_S^k(x)_l - y| \right] > \frac{\gamma}{k+1}.$$

But this contradicts our agnostic learning guarantee from (28), which asserts that

$$\mathbb{E}_{S \sim D_b^m} \mathbb{E}_{(x,y) \sim D_b} \left[ \min_l |\mu_S^k(x)_l - y| \right] \leq \frac{\gamma}{k+1}.$$

Thus, it must be the case that $m_{\mathcal{A},\mathcal{H}}^{k,\mathrm{ag}}(\varepsilon, \delta) \geq d/2 = \Omega(\mathbb{D}_{\gamma,k}^{\mathrm{sfat}}(\mathcal{H}))$ as required. ∎

**Remark 47** *Comparing to Theorem 26 from Bartlett and Long (1998), we note that the accuracy (i.e., $\varepsilon$) for which we can show a lower bound scales inversely with $k$. Intuitively, this factor arises because the learning algorithm can output $k$ labels, and we measure error only with respect to the best label—this is reflected in the calculation in (29).*

---

12. This is not the case if we work with the $(\gamma, k)$-fat-shattering dimension instead, where $S$ could determine $b$ entirely (for example, if every $f_b$ has its own set of labels).

### B.2. Bound on Sum of Union and Intersection

We prove two claims that lower bound the sum of the measures of the union and intersection of a collection of sets, given that each of these sets have a decent enough measure. We expect these claims to possibly be known, but since we could not find a good reference, we prove them here. We state the first claim simply in terms of sizes of the sets, while we state the second one in terms of arbitrary probability measures on the sets. We remark that the former can be derived from the latter by considering the uniform distribution on finite sets; nevertheless, because we are able to show the former via a simple application of the pigeonhole principle, which illustrates the main idea better, we choose to include both the proofs.

**Claim 48** *For $k \geq 1$, let $S_1, \ldots, S_{k+1}$ be $k + 1$ sets such that $|S_i| \geq m, \forall i \in [k + 1]$. Then, we have that*

$$\left| \bigcup_{i=1}^{k+1} S_i \right| + \left| \bigcap_{i=1}^{k+1} S_i \right| \geq \left( \frac{k+1}{k} \right) m.$$

**Proof** Let $T = \bigcap_{i=1}^{k+1} S_i$, and let $|T| = x$. Note that

$$\left| \bigcup_{i=1}^{k+1} S_i \right| + \left| \bigcap_{i=1}^{k+1} S_i \right| = \left| \bigcup_{i=1}^{k+1} S_i \setminus T \right| + 2x. \tag{30}$$

Consider the sets $S_1 \setminus T, \ldots, S_{k+1} \setminus T$. Then, we have that $|S_i \setminus T| \geq m - x, \forall i \in [k + 1]$. Further, observe that

$$\bigcap_{i=1}^{k+1} S_i \setminus T = \emptyset.$$

Thus, every element in the union $\bigcup_{i=1}^{k+1} S_i \setminus T$ is excluded from at least one $S_i \setminus T$. By the pigeonhole principle, some $S_i \setminus T$ excludes at least a $\frac{1}{k+1}$ fraction of the elements in the union. That is, for such an $S_i \setminus T$,

$$m - x \leq |S_i \setminus T| \leq \left( 1 - \frac{1}{k+1} \right) \left| \bigcup_{i=1}^{k+1} S_i \setminus T \right|$$

$$\implies \quad \left| \bigcup_{i=1}^{k+1} S_i \setminus T \right| \geq \left( \frac{k+1}{k} \right) (m - x).$$

Substituting in (30), we get

$$\left| \bigcup_{i=1}^{k+1} S_i \right| + \left| \bigcap_{i=1}^{k+1} S_i \right| \geq \left( \frac{k+1}{k} \right) m + x \left( \frac{k-1}{k} \right) \geq \left( \frac{k+1}{k} \right) m,$$

where the last inequality follows because $k \geq 1$. ∎

**Claim 49** *For $k \geq 1$, let $A_1, \ldots, A_{k+1}$ be $k+1$ events in a probability space such that $\Pr[A_i] > c$, $\forall i \in [k+1]$. Then, we have that*

$$\Pr\left[\bigcup_{i=1}^{k+1} A_i\right] + \Pr\left[\bigcap_{i=1}^{k+1} A_i\right] > \left(\frac{k+1}{k}\right) c.$$

**Proof** Let $T = \bigcap_{i=1}^{k+1} A_i$ and let $\Pr[T] = x$. Then, we have that

$$\Pr\left[\bigcup_{i=1}^{k+1} A_i\right] + \Pr\left[\bigcap_{i=1}^{k+1} A_i\right] = \Pr\left[\bigcup_{i=1}^{k+1} A_i \setminus T\right] + 2x. \tag{31}$$

Let $C_i = A_i \setminus T$. Define $B_0 = \bigcup_{j=1}^{k+1} C_j$, and $B_i = \bigcap_{j=1}^{i} C_j$ for $i \in [k+1]$. Then, because the sets $C_i$ have empty common intersection (implying that every element in the union is excluded by at least one set), observe that the union of the sets decomposes as

$$\bigcup_{i=1}^{k+1} C_i = \bigcup_{i=1}^{k+1} B_{i-1} \setminus C_i.$$

Furthermore, observe that every pair of sets involved in the union on the right side above is disjoint. Thus, we have that

$$\Pr\left[\bigcup_{i=1}^{k+1} C_i\right] = \sum_{i=1}^{k+1} \Pr[B_{i-1} \setminus C_i]. \tag{32}$$

Recall that we are given that $\Pr[A_i] > c$ for every $i \in [k+1]$. This implies that $\Pr[C_i] > c - x$ for every $i \in [k+1]$. We claim the following:

$$\exists j \in [k+1] \text{ such that } \Pr\left[\left(\bigcup_{i=1}^{k+1} C_i\right) \setminus C_j\right] \geq \frac{1}{k+1} \cdot \Pr\left[\bigcup_{i=1}^{k+1} C_i\right] \tag{33}$$

Suppose (33) were not true: this would mean that for every $j \in [k+1]$,

$$\Pr\left[\left(\bigcup_{i=1}^{k+1} C_i\right) \setminus C_j\right] < \frac{1}{k+1} \cdot \Pr\left[\bigcup_{i=1}^{k+1} C_i\right]. \tag{34}$$

But note that for every $j \in [k+1]$, since $B_{j-1} \subseteq \bigcup_{i=1}^{k+1} C_i$,

$$\Pr[B_{j-1} \setminus C_j] \leq \Pr\left[\left(\bigcup_{i=1}^{k+1} C_i\right) \setminus C_j\right]. \tag{35}$$

Combining (32), (35) and (34), we get

$$\Pr\left[\bigcup_{j=1}^{k+1} C_j\right] = \sum_{j=1}^{k+1} \Pr[B_{j-1} \setminus C_j] \leq \sum_{j=1}^{k+1} \Pr\left[\left(\bigcup_{i=1}^{k+1} C_i\right) \setminus C_j\right] < \Pr\left[\bigcup_{i=1}^{k+1} C_i\right]$$

which is a contradiction. Hence, (33) is true, and for the $j \in [k+1]$ satisfying it, we get

$$c - x < \Pr[C_j] = \Pr\left[\left(\bigcup_{i=1}^{k+1} C_i\right) \cap C_j\right] \leq \frac{k}{k+1}\Pr\left[\bigcup_{i=1}^{k+1} C_i\right]$$

$$\implies \quad \Pr\left[\bigcup_{i=1}^{k+1} C_i\right] > \left(\frac{k+1}{k}\right)(c-x).$$

Substituting this back in (31), we get

$$\Pr\left[\bigcup_{i=1}^{k+1} A_i\right] + \Pr\left[\bigcap_{i=1}^{k+1} A_i\right] > \left(\frac{k+1}{k}\right)(c-x) + 2x = \left(\frac{k+1}{k}\right)c + x\left(\frac{k-1}{k}\right) \geq \left(\frac{k+1}{k}\right)c,$$

where the last inequality follows because $k \geq 1, x \geq 0$. $\blacksquare$

## B.3. Proof of Lemma 33

**Lemma 33 (Lower Bounding $k$-Ary $\gamma$-Packing Number)** *Let $\mathcal{H} \subseteq [0,1]^{\mathcal{X}}$ be a hypothesis class having $(\gamma, k)$-fat-shattering dimension $\mathbb{D}_{\gamma,k}^{\text{fat}}(\mathcal{H})$. Then,*

$$\mathcal{M}_{\infty}^{k}(\mathcal{H}, \gamma) \geq \exp\left(\Omega\left(\frac{\mathbb{D}_{\gamma,k}^{\text{fat}}(\mathcal{H})}{k^k}\right)\right). \tag{11}$$

**Proof** We will show that

$$\mathcal{M}_{\infty}^{k}(\mathcal{H}, \gamma) \geq \left\lfloor \left(\frac{k+1}{3}\right) \exp\left(\frac{\mathbb{D}_{\gamma,k}^{\text{fat}}(\mathcal{H}) \cdot (k+1)!}{(k+1)^{k+2}}\right) \right\rfloor.$$

Let $(x_1, \ldots, x_d) \in \mathcal{X}^d$ be a $(\gamma, k)$ fat-shattered sequence, and let $F \subseteq \mathcal{H}$ be the set of functions witnessing this shattering according to $c_1, \ldots, c_d$, so that $|F| = (k+1)^d$. We can view this as follows: for each $x_i$, we can choose any one of $k+1$ buckets, given by the regions $[0, c_{i,1} - \gamma], [c_{i,1} + \gamma, c_{i,2} - \gamma], \ldots, [c_{i,k} + \gamma, 1]$, and after having made a choice of bucket for each $x_i$, $F$ contains a function that respects this choice of buckets.

Suppose we populate a subset $F' \subseteq F$ randomly as follows. Initially, $F'$ is empty. For every $x_i$, we will choose one of the $k+1$ buckets for that $x_i$ uniformly at random, until we have chosen a bucket for each of $x_1, \ldots, x_d$. We will then add the function from $F$ respecting this assignment of buckets to $F'$. We will add $m$ such randomly chosen functions to $F'$. We want the property that for every $k+1$ functions in $F'$, there is an $x_i$ such that we chose a different bucket at that $x_i$ for each of these $k+1$ functions. Furthermore, we want $m$ to be as large as possible while respecting this property.

For any fixed $k+1$ functions that we added to $F'$, the probability that we chose a different bucket for each of them on a *fixed* $x_i$ is $\frac{(k+1)!}{(k+1)^{k+1}}$. Thus, the probability that we did not choose a different bucket for each of them on this $x_i$ is $1 - \frac{(k+1)!}{(k+1)^{k+1}}$. The probability that we did not choose a different bucket for each of them at *any* of $x_1, \ldots, x_d$ is thus

$$\left(1 - \frac{(k+1)!}{(k+1)^{k+1}}\right)^d \leq \exp\left(-\frac{d(k+1)!}{(k+1)^{k+1}}\right).$$

By a union bound over all the $\binom{m}{k+1}$ choices of a tuple of $k+1$ functions in $F'$, we obtain that the probability that there exists a tuple of $k+1$ functions that is not $(k+1)$-wise $\gamma$-separated at any of $x_1, \ldots, x_d$ is at most

$$\binom{m}{k+1} \exp\left(-\frac{d(k+1)!}{(k+1)^{k+1}}\right) \leq \left(\frac{em}{k+1}\right)^{k+1} \exp\left(-\frac{d(k+1)!}{(k+1)^{k+1}}\right)$$

$$= \exp\left((k+1)\ln\left(\frac{em}{k+1}\right) - \frac{d(k+1)!}{(k+1)^{k+1}}\right)$$

$$< 1$$

$$\text{so long as} \quad m \leq \left\lfloor \left(\frac{k+1}{3}\right) \exp\left(\frac{d(k+1)!}{(k+1)^{k+2}}\right) \right\rfloor.$$

Thus, for $m = \left\lfloor \left(\frac{k+1}{3}\right) \exp\left(\frac{d(k+1)!}{(k+1)^{k+2}}\right) \right\rfloor$, there is a positive probability of constructing an $F'$ that is $(k+1)$-wise $\gamma$-separated, which in particular means that there exists a $(k+1)$-wise $\gamma$-separated set of this size. ∎

## B.4. Proof of Lemma 34

**Lemma 34 (Relating $k$-strong-fat-shattering to $k$-fat-shattering)** *Fix $\mathcal{L} \subset [0,1]$ such that $|\mathcal{L}| = B < \infty$. Let $\mathcal{H} \subseteq \mathcal{L}^{\mathcal{X}}$ be a hypothesis class having $(\gamma, k)$-fat-shattering dimension $\mathbb{D}^{\text{fat}}_{\gamma,k}(\mathcal{H})$ and $(\gamma, k)$-strong-fat-shattering dimension $\mathbb{D}^{\text{sfat}}_{\gamma,k}(\mathcal{H})$. Then,*

$$\mathbb{D}^{\text{sfat}}_{\gamma,k}(\mathcal{H}) \geq \widetilde{\Omega}\left(\frac{\mathbb{D}^{\text{fat}}_{\gamma,k}(\mathcal{H})}{k^k}\right), \tag{12}$$

*where the $\widetilde{\Omega}(\cdot)$ hides polylog factors in $k, \mathbb{D}^{\text{fat}}_{\gamma,k}(\mathcal{H})$ and $B$.*

**Proof** Let $d_{\text{s}} = \mathbb{D}^{\text{sfat}}_{\gamma,k}(\mathcal{H})$ and $d_{\text{w}} = \mathbb{D}^{\text{fat}}_{\gamma,k}(\mathcal{H})$. We will show that

$$d_{\text{s}} > \frac{d_{\text{w}}}{\ln^2(d_{\text{w}}(k+1)B^{k+1})} \cdot \frac{(k+1)!\ln\left(\frac{k+1}{k}\right)}{12(k+1)^{k+3}}.$$

Let $S = (x_1, \ldots, x_{d_{\text{w}}})$ be a sequence that is $(\gamma, k)$ fat-shattered by $\mathcal{H}$, and consider the class $\mathcal{H}|_S$. Note that $\mathcal{H}|_S$ has $(\gamma, k)$-fat-shattering dimension at least $d_{\text{w}}$ and $(\gamma, k)$-strong-fat-dimension at most $d_{\text{s}}$. From Lemma 33, we know that

$$\mathcal{M}^k_\infty(\mathcal{H}|_S, \gamma) \geq \left\lfloor \left(\frac{k+1}{3}\right) \exp\left(\frac{d_{\text{w}}(k+1)!}{(k+1)^{k+2}}\right) \right\rfloor. \tag{36}$$

Also, from Lemma 32, we have that for $y = \sum_{i=1}^{d_{\text{s}}} \binom{d_{\text{w}}}{i}\binom{B}{k+1}^i$,

$$\mathcal{M}^k_\infty(\mathcal{H}|_S, \gamma) < (k+1)\left[(k+1)B^{k+1}d_{\text{w}}\right]^{\left\lceil \log_{\frac{k+1}{k}} y \right\rceil}. \tag{37}$$

Combining (36) and (37), we get that

$$(k+1)\left[(k+1)B^{k+1}d_{\mathrm{w}}\right]^{\left\lceil \log_{\frac{k+1}{k}} y\right\rceil} > \left\lfloor \left(\frac{k+1}{3}\right)\exp\left(\frac{d_{\mathrm{w}}(k+1)!}{(k+1)^{k+2}}\right)\right\rfloor$$

$$\implies \quad \ln(2(k+1)) + \left\lceil \log_{\frac{k+1}{k}} y\right\rceil \ln\left((k+1)B^{k+1}d_{\mathrm{w}}\right) > \ln\left(\frac{k+1}{3}\right) + \frac{d_{\mathrm{w}}(k+1)!}{(k+1)^{k+2}}$$

$$\implies \quad 2\left\lceil \log_{\frac{k+1}{k}} y\right\rceil \ln\left((k+1)B^{k+1}d_{\mathrm{w}}\right) > \frac{d_{\mathrm{w}}(k+1)!}{(k+1)^{k+2}}$$

But note that

$$y = \sum_{i=1}^{d_{\mathrm{s}}} \binom{d_{\mathrm{w}}}{i}\binom{B}{k+1}^i \leq d_{\mathrm{s}}\cdot d_{\mathrm{w}}^{d_{\mathrm{s}}}\cdot B^{d_{\mathrm{s}}(k+1)}$$

$$\implies \quad \left\lceil \log_{\frac{k+1}{k}} y\right\rceil \leq 2\cdot\left(\frac{\ln(d_{\mathrm{s}}) + d_{\mathrm{s}}\ln(d_{\mathrm{w}}) + d_{\mathrm{s}}(k+1)\ln(B)}{\ln\left(\frac{k+1}{k}\right)}\right)$$

$$\leq \frac{6(k+1)}{\ln\left(\frac{k+1}{k}\right)}\cdot d_{\mathrm{s}}\cdot\ln((k+1)B^{k+1}d_{\mathrm{w}})$$

where we used that $d_{\mathrm{s}} \leq d_{\mathrm{w}}$. Plugging in the previous display, we get the desired bound. ∎

## B.5. Proofs of Claim 36 and Claim 37

**Claim 36** *Let $\mathcal{H} \subseteq [0,1]^{\mathcal{X}}$ be a hypothesis class having $(\gamma,k)$-fat-shattering dimension $d$. Then for any $\alpha \leq \gamma$, the $(\max(\alpha,\gamma-\alpha),k)$-fat-shattering dimension of $\mathcal{H}^{\alpha}$ is at least $d$.*

**Proof** Let $x,y \in [0,1]^2$ be such that $|x-y| \geq \gamma$, and assume without loss of generality that $x > y$. Then,

$$\left|\alpha\left\lfloor\frac{x}{\alpha}\right\rfloor - \alpha\left\lfloor\frac{y}{\alpha}\right\rfloor\right| = \alpha\left(\left\lfloor\frac{x}{\alpha}\right\rfloor - \left\lfloor\frac{y}{\alpha}\right\rfloor\right) \geq \alpha\left\lfloor\frac{x-y}{\alpha}\right\rfloor \geq \alpha\left\lfloor\frac{\gamma}{\alpha}\right\rfloor \geq \max(\alpha,\gamma-\alpha).$$

Then, consider any sequence $(x_1,\ldots,x_m)$ that is $(\gamma,k)$-fat-shattered by $\mathcal{H}$, and let $F \subseteq \mathcal{H}$ witness this shattering with respect to $c_1,\ldots,c_m$. The calculation above then shows that $F^{\alpha}$ witnesses the $(\max(\alpha,\gamma-\alpha),k)$-fat-shattering of the same sequence with respect to $Q_{\alpha}(c_1),\ldots,Q_{\alpha}(c_m)$, which proves the claim. ∎

**Claim 37** *Let $\mathcal{A}$ be an algorithm that agnostically $k$-list learns $\mathcal{H} \subseteq [0,1]^{\mathcal{X}}$ with sample complexity $m_{\mathcal{A},\mathcal{H}}^{k,\mathrm{ag}}(\varepsilon,\delta)$. Then, $\mathcal{A}$ also agnostically $k$-list learns $\mathcal{H}^{\alpha}$, and*

$$m_{\mathcal{A},\mathcal{H}^{\alpha}}^{k,\mathrm{ag}}(\varepsilon+\alpha,\delta) = m_{\mathcal{A},\mathcal{H}}^{k,\mathrm{ag}}(\varepsilon,\delta).$$

**Proof** Fix any distribution on $\mathcal{X}\times[0,1]$. We have that with probability at least $1-\delta$ over a sample $S$ of size $m(\varepsilon,\delta)$ drawn i.i.d from $D$, the hypothesis $\mu^k = \mathcal{A}(S)$ output by $\mathcal{A}$ satisfies

$$\mathbb{E}_{(x,y)\sim D}[\min_l |\mu^k(x)_l - y|] \leq \inf_{f\in\mathcal{H}} \mathbb{E}_{(x,y)\sim D}[|f(x)-y|] + \varepsilon.$$

But note that for any $f \in \mathcal{H}$, and fixed $(x, y)$, because $|f(x) - f^\alpha(x)| \le \alpha$,

$$|f(x) - y| \le |f^\alpha(x) - y| + \alpha,$$

and therefore

$$\mathbb{E}_{(x,y) \sim D}[\min_l |\mu^k(x)_l - y|] \le \inf_{f \in \mathcal{H}} \mathbb{E}_{(x,y) \sim D}[|f^\alpha(x) - y|] + \alpha + \varepsilon$$
$$= \inf_{f^\alpha \in \mathcal{H}^\alpha} \mathbb{E}_{(x,y) \sim D}[|f^\alpha(x) - y|] + \alpha + \varepsilon.$$

■

## Appendix C. Proofs from Section 4

### C.1. Proof of Lemma 41

**Lemma 41 (Small $k$-out-degree list orientation)** *Let $\mathcal{H} \subseteq [0,1]^{\mathcal{X}}$ be a hypothesis class having $(\gamma, k)$-OIG dimension $\mathbb{D}^{\mathrm{OIG}}_{\gamma,k}(\mathcal{H})$. Fix $n > \mathbb{D}^{\mathrm{OIG}}_{\gamma,k}(\mathcal{H})$ and $S \in \mathcal{X}^n$. Then, there exists a $k$-list orientation of the (possibly infinite) one-inclusion graph $\mathcal{G}(\mathcal{H}|_S) = (V, E)$ such that for every $v \in V$, $\mathsf{outdeg}^k(v; \sigma^k, \gamma) < \frac{n}{2(k+1)}$.*

**Proof** First, since $n > \mathbb{D}^{\mathrm{OIG}}_{\gamma,k}(\mathcal{H})$, it must be the case that for every *finite* subgraph $(V', E')$ of $\mathcal{G}(\mathcal{H}|_S)$, there exists a $k$-list orientation $\sigma^k$ such that for every $v \in V'$, $\mathsf{outdeg}^k(v; \sigma^k, \gamma) < \frac{n}{2(k+1)}$. To show that there exists a $k$-list orientation that works for the whole graph, we will use a compactness argument.

Formally, let $\mathcal{Z}$ be the set of pairs $\mathcal{Z} = \{(v, e) \in V \times E : v \in e\}$. For $z = (v, e) \in \mathcal{Z}$, define the discrete topology on the set $X_z = \{0, 1\}$ with $\tau_z = 2^{X_z} = \{\emptyset, \{0\}, \{1\}, \{0, 1\}\}$ as the open sets. Note that $(X_z, \tau_z)$ is compact since $X_z$ is finite. By Tychonoff's theorem, $\mathcal{K} = \prod_{z \in \mathcal{Z}} X_z$ is compact with respect to the product topology. The base open sets in the product topology are sets of the form $\prod_{z \in \mathcal{Z}} S_z$ where only a finite number of the $S_z$s are not equal to $X_z$.

We want to think of the members of $\mathcal{K}$ as possible orientations of $\mathcal{G}(\mathcal{H}|_S)$. Then, define the following "per-vertex good" sets. For any $v \in V$, let

$$A_v = \left\{ \kappa \in \mathcal{K} : |\{e \ni v : \kappa_{(v,e)} = 0\}| < \frac{n}{2(k+1)} \right\}. \tag{38}$$

Intuitively, these are all the orientations for which the number of edges adjacent to $v$ that are not "satisfying" it are fewer than $\frac{n}{2(k+1)}$, where we think of an edge *satisfying* a vertex if it is either oriented to this vertex, or oriented to a vertex that is $\gamma$-close to this vertex.

Next, we define the following "per-edge good" sets. For any $v \in V$ and any $j \in [n]$, let $e_j$ be the edge adjacent to $v$ in the direction $j$. For any $\kappa \in \mathcal{K}$, let $C_{\kappa,v,j} = \{u(j) : u \in e_j, \kappa_{(u,e_j)} = 1\}$. We say that $C_{\kappa,v,j}$ is "$k$-list coverable" if there exist $x_1, \ldots, x_k \in C_{\kappa,v,j}$ such that for any $x \in C_{\kappa,v,j}$, $|x - x_i| \le \gamma$. Then,

$$B_{v,j} = \{\kappa \in \mathcal{K} : C_{\kappa,v,j} \text{ is } k\text{-list coverable}\}. \tag{39}$$

That is, this set simply checks the condition that among all the vertices being supposedly satisfied by the edge, there is in fact a subset of $k$ vertices that can serve as the vertices to which the edge may be oriented.

Now, we claim that $A_v$ is a closed set in the product topology. Fix any $t$ such that $\frac{n}{2(k+1)} \leq t \leq n$. Next, fix some $t$ edges $E_t = \{e_{i_1}, \ldots, e_{i_t}\}$ adjacent to $v$. Let $\mathcal{Z}_1 = \{(v, e) : e \in E_t\}$ and $\mathcal{Z}_2 = \mathcal{Z} \setminus \mathcal{Z}_1$. Then,

$$\bar{A}_v = \bigcup_{\frac{n}{2(k+1)} \leq t \leq n} \bigcup_{E_t = \{e_{i_1}, \ldots, e_{i_t}\}} \underbrace{\prod_{z \in \mathcal{Z}_1} \{0\} \times \prod_{z \in \mathcal{Z}_2} \{0, 1\}}_{\text{base open set}}. \tag{40}$$

Note that every term inside the double-union is a base open set, because every $\mathcal{Z}_1$ is finite. Because an arbitrary union of open sets is open, $\bar{A}_v$ is open, and hence $A_v$ is a closed set.

Next, we claim that every $B_{v,j}$ is also a closed set. Fix any $t \geq k+1$. Fix any $U_t = \{u_1, \ldots, u_t\} \subseteq e_j$ such that the set $\{u_1(j), \ldots, u_t(j)\}$ is not $k$-list coverable. Let $\mathcal{Z}_1 = \{(u, e_j) : u \in U_t\}$ and $\mathcal{Z}_2 = \mathcal{Z} \setminus \mathcal{Z}_1$.

$$\bar{B}_{v,j} = \bigcup_{t \geq k+1} \bigcup_{U_t = \{u_1, \ldots, u_t\}} \underbrace{\prod_{z \in \mathcal{Z}_1} \{1\} \times \prod_{z \in \mathcal{Z}_2} \{0, 1\}}_{\text{base open set}}. \tag{41}$$

Again, every term inside the double-union is a base open set. Thus, $\bar{B}_{v,j}$ is open, which means that $B_{v,j}$ is closed. This means that $\Sigma_v = A_v \cap \bigcap_{j \in [n]} B_{v,j}$ is also closed.

We will now argue that for every finite $V' \subseteq V$, $\bigcap_{v \in V'} \Sigma_v$ is non-empty. This will establish the finite intersection property for the collection of closed sets $\{\Sigma_v\}_{v \in V}$, which will let us claim that $\bigcap_{v \in V} \Sigma_v$ is non-empty. Let $(V', E')$ be the finite subgraph induced by $V'$ on $\mathcal{G}(\mathcal{H}|_S)$. From our argument at the start of the proof, we know that there exists a $k$-list orientation $\sigma^k$ of $(V', E')$ that has maximum $k$-outdegree smaller than $\frac{n}{2(k+1)}$. We can interpret this $\sigma^k$ as defining a $\kappa \in \bigcap_{v \in V'} \Sigma_v$. Namely, for every $v \in V'$, we can find the edges $e$ edjacent to it that do not contribute to its $k$-outdegree — there must be at least $n - \frac{n}{2(k+1)}$ many of these. We set $\kappa_{(v,e)} = 1$ for these edges, and $\kappa_{(v,e)} = 0$ for the edges that do contribute to its $k$-outdegree. Observe that $\kappa$ then satisfies the condition (38) for $A_v$. Now, we need to check the condition for $B_{v,j}$ — let $e_j$ be the edge adjacent to $v$ in the direction $j$. For any vertex $u \in V \setminus V'$ that also belongs to $e_j$, we simply set $\kappa_{(u,e_j)} = 0$. In this way, $\kappa_{(u,e_j)} = 1$ only for the vertices in $V'$. Since $\sigma^k$ is a valid orientation, and by the way we set $\kappa$ to 1 only for the vertices that are satisfied by the edge, we thus satisfy $B_{v,j}$ for all $v$ and $j$. For $(u, e)$ pair that is left over, we can set $\kappa_{(u,e)}$ arbitrarily, and these do not really affect membership of $\kappa$ in $\bigcap_{v \in V'} \Sigma_v$. Thus, we have shown the existence of a valid $\kappa$ in $\bigcap_{v \in V'} \Sigma_v$, which means in particular that this set is non-empty.

This further implies by our preceding argument that $\bigcap_{v \in V} \Sigma_v$ is non-empty. Then, let $\kappa^\star \in \bigcap_{v \in V} \Sigma_v$. We will construct a $k$-list orientation from $\kappa^\star$. For every vertex $v \in V$, and every edge $e_j$ adjacent to it (in direction $j$) for which $\kappa^\star_{(v,e_j)} = 1$, if $e_j$ has already not been list-oriented, we obtain the set $C_{\kappa^\star, v, j} = \{u(j) : u \in e_j, \kappa^\star_{(u,e_j)} = 1\}$. By the per-edge condition (39) on $B_{v,j}$, we know that $C_{\kappa^\star, v, j}$ is $k$-list coverable. We orient the edge $e_j$ to the $k$-list that covers $C_{\kappa^\star, v, j}$. This edge will then satisfy every vertex $u \in e_j$ for which $\kappa^\star_{(u,e_j)} = 1$. By the per-vertex condition (38) on $A_v$, we also know that at least $n - \frac{n}{2(k+1)}$ edges adjacent to $v$ will satisfy it, and hence its

---

**Algorithm 4** Weak learner $\mathcal{A}_w$ for $\mathcal{H} \subseteq [0,1]^{\mathcal{X}}$ based on the one-inclusion graph

---

**Input:** An $\mathcal{H}$-realizable sample $S = \big((x_1, y_1), \ldots, (x_n, y_n)\big)$, scale parameter $\gamma$.
**Output:** A $k$-list hypothesis $\mathcal{A}_w(S) = \mu_S^k : \mathcal{X} \to \{Y \subseteq [0,1] : |Y| \leq k\}$.

For each $x \in \mathcal{X}$, the $k$-list $\mu_S^k(x)$ is computed as follows:

1: Consider the class of all patterns over the *unlabeled data* $\mathcal{H}|_{(x_1,\ldots,x_n,x)} \subseteq [0,1]^{n+1}$.
2: Find a $k$-list orientation $\sigma^k$ of $\mathcal{G}(\mathcal{H}|_{(x_1,\ldots,x_n,x)})$ that *minimizes* the *maximum* scaled $k$-outdegree $\mathrm{outdeg}^k(\sigma^k, \gamma)$.
3: Consider the edge in direction $x$ defined by $S$:
$$e = \{h \in \mathcal{H}|_{(x_1,\ldots,x_n,x)} : \forall i \in [n] \ h(x_i) = y_i\}.$$
4: Set $\mu_S^k(x) = \{h(x) : h \in \sigma^k(e)\}$.

---

$k$-outdegree would be smaller than $\frac{n}{2(k+1)}$. If we do this for every vertex and every direction that remains, we would have ensured that every vertex has $k$-outdegree smaller than $\frac{n}{2(k+1)}$. Finally, if any edge remains unoriented, we can orient it arbitrarily to any $\leq k$ vertices it is adjacent to, and we have obtained the list orientation we desire. ∎

## C.2. Proof of Lemma 42

**Lemma 42 (Weak list learner)** *Let $\mathcal{H} \subseteq [0,1]^{\mathcal{X}}$ be a hypothesis class having $(\gamma, k)$-OIG dimension $\mathbb{D}_{\gamma,k}^{\mathrm{OIG}}(\mathcal{H})$. Fix $n \geq \mathbb{D}_{\gamma,k}^{\mathrm{OIG}}(\mathcal{H})$. Then, there exists a learning algorithm $\mathcal{A}_w$, such that for any distribution $\mathcal{D}$ realizable by $\mathcal{H}$, the following guarantee holds:*

$$\Pr_{S \sim \mathcal{D}^n} \Pr_{(x,y) \sim \mathcal{D}}[\mu_S^k(x) \not\ni_\gamma y] < \frac{1}{2(k+1)},$$

*where $\mu_S^k = \mathcal{A}_w(S)$ is the $k$-list hypothesis output by $\mathcal{A}_w$.*

**Proof** The argument is similar to the proof of Lemma 17, and is based on the one-inclusion graph algorithm. In particular, we will show that Algorithm 4 satisfies the requirements of the lemma.

Because each sample is drawn i.i.d. from $\mathcal{D}$, we can apply the classical leave-one-out symmetrization argument to claim

$$\Pr_{S \sim \mathcal{D}^n} \Pr_{(x,y) \sim \mathcal{D}}[\mu_S^k(x) \not\ni_\gamma y] = \Pr_{S \sim \mathcal{D}^{n+1}} \Pr_{i \sim \mathrm{Unif}([n+1])}[\mu_{S_{-i}}^k(x_i) \not\ni_\gamma y_i],$$

where $\mathrm{Unif}([n+1])$ is the uniform distribution on $[n+1]$, and $\mu_{S_{-i}}^k = \mathcal{A}_w(S \setminus \{(x_i, y_i)\})$. It then suffices to show that for every fixed sample $S = \{(x_1, y_1), \ldots, (x_{n+1}, y_{n+1})\}$ of size $n+1$,

$$\Pr_{i \sim \mathrm{Unif}([n+1])}[\mu_{S_{-i}}^k(x_i) \not\ni_\gamma y_i] < \frac{1}{2(k+1)}.$$

Let $y = (y_1, \ldots, y_{n+1})$ be a vertex in the one-inclusion graph $\mathcal{G}(\mathcal{H}|_S)$. Note that this vertex exists due to the realizability assumption. For any $i \in [n+1]$, observe that when Algorithm 4 is asked to

make a prediction on $x_i$ given the labeled sample $S_{-i}$, it constructs the same one-inclusion graph $\mathcal{G}(\mathcal{H}|_S)$ and hence the same outdegree-minimizing orientation $\sigma^k$. Let $e_i$ be the edge adjacent to vertex $y$ in the direction $i$ in this oriented graph, and consider the list of labels $\mu(e_i) = \{h(x_i) : h \in \sigma^k(e_i)\}$. Observe that $\mu_{S_{-i}}^k(x_i) \not\ni_\gamma y_i$ if and only if $\mu(e_i)$ does not $\gamma$-contain $y_i$. Namely,

$$
\begin{aligned}
\Pr_{i \sim \mathrm{Unif}([n+1])}[\mu_{S_{-i}}^k(x_i) \not\ni_\gamma y_i] &= \frac{1}{n+1} \sum_{i=1}^{n+1} \mathbb{1}[\mu_{S_{-i}}^k(x_i) \not\ni_\gamma y_i] \\
&= \frac{1}{n+1} \sum_{i=1}^{n+1} \mathbb{1}[\mu(e_i) \not\ni_\gamma y_i] \\
&= \frac{\mathsf{outdeg}^k(y; \sigma^k, \gamma)}{n+1}.
\end{aligned}
$$

Finally, by Lemma 41, $\mathsf{outdeg}^k(y; \sigma^k, \gamma) < \frac{n+1}{2(k+1)}$, which completes the proof. ∎

### C.3. Proof of Theorem 40

**Theorem 40 (Finite $(\gamma, k)$-OIG dimension is sufficient)** *Let $\mathcal{H} \subseteq [0,1]^{\mathcal{X}}$ be a hypothesis class. For any distribution $\mathcal{D}$ over $\mathcal{X} \times [0,1]$ realizable by $\mathcal{H}$ and $\gamma \in (0,1)$, there exists a list learning algorithm $\mathcal{A}$ which takes as input an i.i.d. sample $S \sim \mathcal{D}^n$ and outputs a $k$-list hypothesis $\mathcal{A}(S)$ satisfying*

$$
\mathrm{err}_{\mathcal{D}, \ell_{abs}}(\mathcal{A}(S)) \leq \gamma + \widetilde{O}\left( \frac{k \cdot \mathbb{D}_{\gamma,k}^{\mathrm{OIG}}(\mathcal{H}) + \log(1/\delta)}{n} \right)
$$

*with probability at least $1-\delta$. In other words, $m_{\mathcal{A}, \mathcal{H}}^{k, \mathrm{re}}(\varepsilon, \delta) \leq \widetilde{O}\left( \frac{k \cdot \mathbb{D}_{\gamma,k}^{\mathrm{OIG}}(\mathcal{H}) + \log(1/\delta)}{\varepsilon} \right)$ for $\gamma = \Theta(\varepsilon)$.*

**Proof** Let $n_0 = \mathbb{D}_{\gamma,k}^{\mathrm{OIG}}(\mathcal{H})$ for ease of notation. The theorem statement is not interesting for $n \leq \mathbb{D}_{\gamma,k}^{\mathrm{OIG}}(\mathcal{H})$, because the right side is larger than 1 in this case. Hence, let $n > n_0$, and let $S$ be any labeled sequence of size $n$ realizable by $\mathcal{H}$. We claim that there exists a distribution $\mathcal{P}$ over subsequences $T$ of $S$ of size $n_0$ (with repetitions allowed) such that for every $(x, y) \in S$,

$$
\Pr_{T \sim \mathcal{P}}[\mu_T^k(x) \not\ni_\gamma y] < \frac{1}{2(k+1)},
$$

where $\mu_T^k = \mathcal{A}_w(T)$ for the weak learner $\mathcal{A}_w$ promised by Lemma 42. Towards this, consider a zero-sum game between Max and Minnie, where Max's pure strategies are examples $(x, y) \in S$, and Minnie's pure strategies are sequences $S^{n_0}$. The payoff matrix $L$ is given by $L_{T,(x,y)} = \mathbb{1}[\mu_T^k(x) \not\ni_\gamma y]$. Now, let $\mathcal{D}$ be any mixed strategy (i.e., distribution over the examples in $S$) by Max. From Lemma 42, we know that

$$
\Pr_{T \sim \mathcal{D}^{n_0}} \Pr_{(x,y) \sim \mathcal{D}}[\mu_T^k(x) \not\ni_\gamma y] < \frac{1}{2(k+1)}.
$$

But note that the quantity on the LHS above is the expected payoff, when Max plays the mixed strategy $\mathcal{D}$ and Minnie plays the mixed strategy $\mathcal{D}^{n_0}$. Namely, for any mixed strategy that Max can

play, there exists a mixed strategy that Minnie can play which guarantees that the expected payoff smaller than $\frac{1}{2(k+1)}$. By the min-max theorem, there exists a mixed strategy that Minnie can play, such that for any mixed strategy that Max could play (in particular, any pure strategy as well), the expected payoff is smaller than $\frac{1}{2(k+1)}$. This mixed strategy is the required distribution $\mathcal{P}$.

Now, suppose we draw $l = 6(k+1)\log(2n)$ sequences $T_1, \ldots, T_l$ independently from $\mathcal{P}$. By a Chernoff bound, we have that for any $(x, y) \in S$,

$$\Pr\left[\frac{1}{l}\sum_{i=1}^{l} \mathbb{1}[\mu_{T_i}^k(x) \not\ni_\gamma y] \geq \frac{1}{k+1}\right] \leq \exp\left(-\frac{l}{6(k+1)}\right) < \frac{1}{n}.$$

A union bound over the $n$ examples in $S$ then implies that there exist sequences $T_1, \ldots, T_l$ such that for all $(x, y) \in S$ simultaneously, it holds that

$$\sum_{i=1}^{l} \mathbb{1}[\mu_{T_i}^k(x) \ni_\gamma y] > \frac{kl}{k+1}.$$

Namely, for each $(x, y) \in S$, the target label $y$ is $\gamma$-contained in more than $\frac{kl}{k+1}$ of the list predictions on $x$. Now, concatenate all the labels in the $l$ lists $\mu_{T_1}^k(x), \ldots, \mu_{T_l}^k(x)$ to obtain a giant list of size at most $N = kl$; sort this list. Finally, obtain the $\frac{1}{k+1}^{\text{th}}$ quantiles in this sorted list and collect them in a list $\mu(x)$ of size $k$. Namely, consider the elements at indices $\left\lceil\frac{N}{k+1}\right\rceil, \left\lceil\frac{2N}{k+1}\right\rceil, \ldots, \left\lceil\frac{kN}{k+1}\right\rceil$ in the sorted list.

The claim is that $y \in_\gamma \mu(x)$. To see this, note that there are at least $\left\lceil\frac{N}{k+1}\right\rceil$ "good" labels in the sorted list that are all $\gamma$-close to $y$. The only bad case that we need to worry about is when these good labels all get trapped in a single region *between* two consecutive quantiles (and not including any). But note that the number of elements between two consecutive quantiles is at most $\left\lceil\frac{N}{k+1}\right\rceil - 1$, and hence this is not possible. In particular, at least one of the $k$ elements in $\mu(x)$ is sandwiched in between some pair of good labels in the sorted list, and is hence itself $\gamma$-close to $y$.

The subsequences $T_1, \ldots, T_l$ constitute a sample compression. Namely, consider the following sample compression scheme: for each $S$, $\kappa(S)$ simply outputs the concatenation of the sequences $T_1, \ldots, T_l$ constructed above (of total size $s = O(n_0 k \log(n))$), and $\rho$ runs the above procedure of aggregation on this concatenation: the output list of $\rho$ on each $x \in S$ is then guaranteed to $\gamma$-contain the true label $y$. Thus, $(\kappa, \rho)$ constitute a list sample compression scheme for $(\mathcal{X}, [0, 1])$. We can then instantiate Lemma 10, with the loss function $\ell_{\text{thr}} : [0, 1]^k \times [0, 1] \to [0, 1]$, where

$$\ell_{\text{thr}}(\mu, y) = \mathbb{1}[\mu \not\ni_\gamma y],$$

to obtain that with probability $1 - \delta$ over $S$,

$$\text{err}_{\mathcal{D}, \ell_{\text{thr}}}(\rho(\kappa(S))) \leq O\left(\frac{n_0 k \log^2(n) + \log(1/\delta)}{n}\right).$$

Here, we used that $\hat{\text{err}}_{S, \ell_{\text{thr}}}(\rho(\kappa(S)))$ and $\hat{\text{err}}_{S \setminus \kappa(S), \ell_{\text{thr}}}(\rho(\kappa(S)))$ are both 0 for the compression scheme $(\kappa, \rho)$. But now, note that for any $k$-list hypothesis $\mu$ and $x \geq 0$, $\text{err}_{\mathcal{D}, \ell_{\text{thr}}}(\mu) \leq x$ implies

that $\mathrm{err}_{\mathcal{D}, \ell_{\mathrm{abs}}} \leq (1 - x)\gamma + x \leq \gamma + x$, and hence with probability at least $1 - \delta$ over $S$,

$$\mathrm{err}_{\mathcal{D}, \ell_{\mathrm{abs}}}(\rho(\kappa(S))) \leq \gamma + O\left(\frac{n_0 k \log^2(n) + \log(1/\delta)}{n}\right).$$

The required learning algorithm $\mathcal{A}$ maps $S \mapsto \rho(\kappa(S))$, completing the proof. ∎

### C.4. Proof of Theorem 43

**Theorem 43 (Finite $(\gamma, k)$-OIG dimension is necessary)** *Let $\mathcal{H} \subseteq [0,1]^{\mathcal{X}}$ be a hypothesis class having $(\gamma, k)$-OIG dimension $\mathbb{D}_{\gamma,k}^{\mathrm{OIG}}(\mathcal{H})$, and let $\mathcal{A}$ be any $k$-list regression algorithm for $\mathcal{H}$ in the realizable setting. Then, for any $\varepsilon, \delta \in (0, 1)$ satisfying $\varepsilon < \Theta(1/k^2)$ and $\delta < \sqrt{\varepsilon}$,*

$$m_{\mathcal{A}, \mathcal{H}}^{k, \mathrm{re}}(\varepsilon, \delta) \geq \Omega\left(\frac{\mathbb{D}_{\gamma,k}^{\mathrm{OIG}}(\mathcal{H})}{k\sqrt{\varepsilon}}\right) \quad \text{for } \gamma = \Theta(\sqrt{\varepsilon}).$$

**Proof** Let $n_0 = \mathbb{D}_{2\gamma,k}^{\mathrm{OIG}}(\mathcal{H})$ for ease of notation. By the definition of $\mathbb{D}_{2\gamma,k}^{\mathrm{OIG}}(\mathcal{H})$, we know that there exists a sequence $S \in \mathcal{X}^{n_0}$ such that there exists a finite subgraph $G = (V, E)$ of the one-inclusion graph $\mathcal{G}(\mathcal{H}|_S)$ that satisfies the following property: for any $k$-list orientation $\sigma^k$ of the edges in $E$, there exists a vertex $v \in V$ that has $\mathrm{outdeg}^k(v; \sigma^k, 2\gamma) \geq \frac{n_0}{2(k+1)}$.

Given any learning algorithm $\mathcal{A}$, we will define a corresponding $k$-list orientation $\sigma_{\mathcal{A}}^k$ of the edges in $G$ such that the $k$-outdegree of every vertex with respect to this orientation is upper bounded by the expected error of $\mathcal{A}$. Since we know by the property above that there exists a vertex $v$ whose $k$-outdegree is bounded from below, we will have obtained the required lower bound.

First, for every vertex $v = (v_1, \ldots, v_{n_0}) \in V$, where $v_1, \ldots, v_{n_0}$ correspond to the labels of some function in $\mathcal{H}$ on $x_1, \ldots, x_{n_0}$, let $P_v$ be the distribution

$$P_v((x_1, v_1)) = 1 - 24(k+1)\sqrt{\varepsilon}, \qquad P_v((x_i, v_i)) = \frac{24(k+1)\sqrt{\varepsilon}}{n_0 - 1} \text{ for } i = 2, \ldots, n_0. \quad (42)$$

Namely, $P_v$ assigns a massive chunk of its mass to $(x_1, v_1)$, and uniformly distributes the remaining tiny amount of mass over $(x_2, v_2) \ldots, (x_{n_0}, v_{n_0})$.

Now, fix a training set size $m = \frac{n_0}{72(k+1)\sqrt{\varepsilon}}$; this is the lower bound we are going for. Let $e_t \in E$ be any edge in the direction $t \in [n_0]$. For any $u \in e_t$, define

$$p_{e_t}(u) = \Pr_{S \sim P_u^m}\left[\mu^k(x_t) \not\approx_\gamma u_t \mid (x_t, u_t) \notin S\right], \quad (43)$$

where $\mu^k = \mathcal{A}(S)$ is the $k$-list hypothesis output by $\mathcal{A}$ on the training set $S$. In words, $p_{e_t}(u)$ measures the probability (according to $P_u$) that $\mathcal{A}$ has a large error on the hidden test point $x_t$ (where $t$ is the direction of the edge $e_t$), when the training set corresponds to the edge $e_t$. The $k$-list orientation $\sigma_{\mathcal{A}}^k$ that we will construct will be a function of these probabilities $p_{e_t}(u)$—we will orient edges towards vertices having small $p_{e_t}(u)$.

For an edge $e_t \in E$ in the direction $t$, define the set

$$C_{e_t} = \left\{u \in e_t : p_{e_t}(u) < \frac{1}{k+1}\right\}. \quad (44)$$

If $C_{e_t} = \emptyset$, we set $\sigma_{\mathcal{A}}^k(e_t)$ to be an arbitrary $\leq k$-sized subset of $e_t$.

Otherwise, we claim that there exists a subset $\mathcal{B} \subseteq C_{e_t}$ having size at most $k$, such that for every $u \in C_{e_t}$, $\exists c \in \mathcal{B}$ such that $|c_t - u_t| \leq 2\gamma$. To see this, first note that for any $u \in e_t$ (and hence any $u \in C_{e_t}$), the distribution $P_u^m$ conditioned on $(x_t, u_t) \notin S$ is the same. This is because of the way we have defined the distributions in (42), and the fact that every $u \in C_{e_t}$ is identically everywhere other than at $x_t$. Summarily, the random variable $\mu^k(x_t)$ follows a common conditional distribution $D$ for any $u \in C_{e_t}$. If it were the case that $C_{e_t}$ were not $2\gamma$-coverable by a subset of size $\leq k$, then there must exist $u^{(1)}, \ldots, u^{(k+1)} \in C_{e_t}$ such that every $|u_t^{(i)} - u_t^{(j)}| > 2\gamma$. Let $A_i$ be the event (with respect to the common conditional distribution $D$) that $\mu^k(x_t) \ni_\gamma u_t^{(i)}$. Then, because of the previous sentence, $\cap_{i=1}^{k+1} A_i = \emptyset$. Furthermore, on account of membership in $C_{e_t}$, for every $i \in [k+1]$, we have that $\Pr_D[A_i] > \frac{k}{k+1}$. Instantiating Claim 49 with the events $A_1, \ldots, A_{k+1}$, we would then obtain that

$$\Pr_D[\cup_{i=1}^{k+1} A_i] > 1,$$

which is not possible. Thus, the required cover $\mathcal{B} \subseteq C_{e_t}$ of size at most $k$ must exist, and we set $\sigma_{\mathcal{A}}^k(e_t) = \mathcal{B}$.

Now, we can upper bound the $k$-outdegree of any vertex $v$ in this orientation. Concretely, denoting $e_1, \ldots, e_{n_0}$ to be the edges adjacent to $v$ in directions 1 through $n_0$, we have

$$\text{outdeg}^k(v; \sigma_{\mathcal{A}}^k, 2\gamma) = \sum_{t \in [n_0]} \mathbb{1}[\text{all values in } \sigma_{\mathcal{A}}^k(e_t) \text{ are more than } 2\gamma \text{ away from } v_t].$$

Note that if $p_{e_t}(v) < \frac{1}{k+1}$, $v \in C_{e_t}$, and $\sigma_{\mathcal{A}}^k(e_t)$ will contain an element within $2\gamma$ of $v_t$. Thus, the indicator is only active when $p_{e_t}(v) > \frac{k}{k+1}$, which means that

$$\text{outdeg}^k(v; \sigma_{\mathcal{A}}^k, 2\gamma) = \sum_{t \in [n_0]} \mathbb{1}\left[p_{e_t}(v) > \frac{k}{k+1}\right] \leq 1 + \sum_{t=2}^{n_0} \mathbb{1}\left[p_{e_t}(v) > \frac{k}{k+1}\right]$$

$$\leq 1 + \frac{k+1}{k} \sum_{t=2}^{n_0} \Pr_{S \sim P_v^m}\left[\mu^k(x_t) \not\ni_\gamma u_t \mid (x_t, u_t) \notin S\right]$$

$$= 1 + \frac{k+1}{k} \sum_{t=2}^{n_0} \frac{\Pr_{S \sim P_v^m}\left[\{\mu^k(x_t) \not\ni_\gamma u_t\} \wedge \{(x_t, u_t) \notin S\}\right]}{\Pr_{S \sim P_v^m}[(x_t, v_t) \notin S]}.$$

But notice that by the definition of $P_v$ (42), we have that for $t \geq 2$,

$$\Pr_{S \sim P_v^m}[(x_t, v_t) \notin S] = \left(1 - \frac{24(k+1)\sqrt{\varepsilon}}{n_0 - 1}\right)^m \geq 1 - \frac{24(k+1)m\sqrt{\varepsilon}}{n_0 - 1}$$

$$= 1 - \frac{n_0}{3n_0 - 3} \geq \frac{1}{3}.$$

where we used that $(1+x)^r \geq 1 + rx$ for any $x \geq -1$, and substituted $m = \frac{n_0}{72(k+1)\sqrt{\varepsilon}}$. Substituting this bound above, we get

$$
\begin{aligned}
\mathrm{outdeg}^k(v; \sigma_{\mathcal{A}}^k, 2\gamma) &\leq 1 + \frac{3(k+1)}{k} \sum_{t=2}^{n_0} \Pr_{S \sim P_v^m}\left[ \{\mu^k(x_t) \not\approx_\gamma u_t\} \wedge \{(x_t, u_t) \notin S\}\right] \\
&\leq 1 + \frac{3(k+1)}{k} \sum_{t=2}^{n_0} \Pr_{S \sim P_v^m}\left[ \mu^k(x_t) \not\approx_\gamma u_t\right] \\
&= 1 + \frac{3(k+1)}{k} \left(\frac{n_0 - 1}{24(k+1)\sqrt{\varepsilon}}\right) \sum_{t=2}^{n_0} \left(\frac{24(k+1)\sqrt{\varepsilon}}{n_0 - 1}\right) \Pr_{S \sim P_v^m}\left[ \mu^k(x_t) \not\approx_\gamma u_t\right] \\
&\leq 1 + \frac{3(k+1)}{k} \left(\frac{n_0 - 1}{24(k+1)\sqrt{\varepsilon}}\right) \mathbb{E}_{(x,y) \sim P_v} \mathbb{E}_{S \sim P_v^m}\left[ \mathbb{1}\left[\mu^k(x) \not\approx_\gamma y\right]\right] \\
&= 1 + \frac{3(k+1)}{k} \left(\frac{n_0 - 1}{24(k+1)\sqrt{\varepsilon}}\right) \mathbb{E}_{S \sim P_v^m} \Pr_{(x,y) \sim P_v}\left[ \mu^k(x) \not\approx_\gamma y\right]
\end{aligned}
$$

Finally, using that there exists some $v^\star$ that has $\mathrm{outdeg}^k(v; \sigma_{\mathcal{A}}^k, 2\gamma) \geq \frac{n_0}{2(k+1)}$ because of our choice of $S$ and $G$, we get

$$
\begin{aligned}
\mathbb{E}_{S \sim P_v^m} \Pr_{(x,y) \sim P_v}\left[ \mu^k(x) \not\approx_\gamma y\right] &\geq \frac{24(n_0 - 2k - 2)k(k+1)\sqrt{\varepsilon}}{6(k+1)(k+1)(n_0 - 1)} \\
&\geq 2\sqrt{\varepsilon},
\end{aligned} \tag{45}
$$

where we used that $\frac{k}{k+1} \geq \frac{1}{2}$ for $k \geq 1$, and that $\frac{n_0 - 2k - 2}{n_0 - 1} \geq \frac{1}{2}$ for $n_0$ large enough.

But now, by definition, when $\mathcal{A}$ uses $m' \geq m_{\mathcal{A},\mathcal{H}}^{k,\mathrm{re}}(\varepsilon, \delta)$ samples, with probability at least $1 - \delta$ over the draw of $S \sim P_{v^\star}^{m'}$, its error is at most $\varepsilon$. Namely, with probability at least $1 - \delta$, its output $\mu^k$ satisfies

$$
\begin{aligned}
&\mathrm{err}_{P_{v^\star}, \ell_{\mathrm{abs}}}[\mu^k] = \mathbb{E}_{(x,y) \sim P_{v^\star}}[\ell_{\mathrm{abs}}(\mu^k(x), y)] \leq \varepsilon \\
\implies\quad &\Pr_{(x,y) \sim P_{v^\star}}[\mu^k(x) \not\approx_{\sqrt{\varepsilon}} y] \leq \sqrt{\varepsilon} \qquad \text{(Markov's inequality)} \\
\implies\quad &\mathbb{E}_{S \sim P_v^{m'}} \Pr_{(x,y) \sim P_v}\left[ \mu^k(x) \not\approx_{\sqrt{\varepsilon}} y\right] \leq (1 - \delta)\sqrt{\varepsilon} + \delta < \sqrt{\varepsilon} + \delta < 2\sqrt{\varepsilon},
\end{aligned}
$$

Therefore, for (45) above to not be a contradiction, it must be the case that for $\gamma = \sqrt{\varepsilon}$,

$$
m_{\mathcal{A},\mathcal{H}}^{k,\mathrm{re}}(\varepsilon, \delta) \geq \frac{n_0}{72(k+1)\sqrt{\varepsilon}} = \Omega\left(\frac{\mathbb{D}_{2\gamma,k}^{\mathrm{OIG}}(\mathcal{H})}{k\sqrt{\varepsilon}}\right).
$$

∎

