# OpenReview forum: "A Characterization of List Regression"
_algorithmiclearningtheory.org/ALT/2025/Conference — ALT 2025_

### Official Review · Reviewer_hvHa · 2024-11-07
**Review of the paper *A characterization of list regression***

**Rating:** 5
**Confidence:** 1

**Review:**

This paper introduces the learning framework of *list regression* and introduces two combinatorial dimensions, generalizing, respectively, the fat-shattering dimension by Alon, Ben-David, Cesa-Bianchi and Haussler and the OIG dimension by Attias, Hanneke, Kalavasisi, Karbasi and Velegkas, to characterize, respectively, the agnostic case and the realizable one.

List regression can be seen as a particular instance of a *list learning task*. For such kind of tasks, the learning algorithm, on input a given test point, is allowed to output a short list of values for that point and it is declared successful whenever at least one of them is correct. The study of list learning frameworks for **classification** has recently received some attention: here, the learner outputs a list of possible labels for the given test point, and succeeds in case its true label belongs to that list. List classification is, indeed, a very natural framework to study, with wide applicability (just think of recommendation systems). In the case of regression, on the other hand, we deal with objective functions $h \colon \mathcal{X} \to [0,1]$. Thus, in the list regression framework, the learner, when receiving a test point $x$, must output a list $(y_1, \dots, y_k)$ of values and its prediction is consider successful whenever some of the $y_i$'s is sufficiently close to $h(x)$. Although this may be theoretically interesting, such framework doesn't seem as much natural to consider, and the authors do not really provide clear motivations for their work.

As mentioned in the beginning, the work done in the paper can be summarized as follows:

- The authors propose a generalization of fat-shattering dimension, which they call $k$-fat-shattering dimension: usual fat-shattering dimension is recovered in case we set $k=1$ in their definition. Furthermore, they prove that finiteness (at every scale) of such dimension characterizes learnability in the agnostic list regression framework, by some learner which outputs a list of $k$ values.

- Similarly, they generalize OIG dimension to the so-called $k$-OIG dimension: again, the standard definition corresponds to the latter in case $k=1$. They hence show that a class can be learned in the realizable list regression framework by some learner which outputs a list of $k$ values if and only if its $k$-OIG dimension is finite at every scale.

- Both in the agnostic and the realizable settings, list regression is strictly easier that standard regression: this is shown by analyzing examples inspired, respectively, by the work of Charikar and Pabbaraju and that of Bartlett, Long and Williamson.

**PROS**

Although I was not able to check the details of their argument, the authors develop a seemingly interesting theoretical machinery, reworking techniques from different areas of learning theory. Therefore, they seem to have an accurate knowledge of the state of the art.

**CONS**

- I honestly found the paper very hard to follow. It is also very hard to review it, as, in particular, the first 12 pages of the paper do not really contain any outline of the main arguments. I strongly suggest to the authors to modify the setup of the paper in such a way that the first 12 pages contain both the definitions of $k$-fat-shattering and $k$-OIG dimension an informative (yet, possibly, intuitive) general outline of both the main results.

- The whole framework of list regression does not seem very natural to consider: indeed, the motivating example given in the introduction appears quite artificial and uninteresting. Is there a more natural or interesting example which can better motivate this learning setting?

- It is not really clear to me how this paper relates with the framework of list-decodable regression. I hope that the authors are willing to elaborate further on this point.

**Paper Award:**

No

---

> ### Author Response · Authors · 2024-11-25
> **Response to review**
>
> Thank you so much for your time in reviewing our paper!
>
> The only reason we did not define our dimensions in the results section was because we felt that the definitions were a little technically verbose, and we did not want to distract the reader from the main results themselves, which was the characterization of learnability in terms of the dimensions. However, your suggestion is well-taken: to give the reader a sense of our constructions, we will possibly try and motivate the construction of the k-fat-shattering dimension from the fat-shattering dimension right in the results section itself, since it seems like the easier of the two. We will also try to include slightly more detailed summaries, in terms of technical tools used, in the first 12 pages.
>
> To elaborate on the differences with list-decodable regression: In list-decodable regression, the setting is that of **linear** regression, where in a dataset of $n$ samples, $\alpha n$ samples $(x_i, y_i)$ follow the model $y_i = \langle x_i, l^* \rangle$ (small noise may be permitted as well), whereas the remaining $(1-\alpha)n$ samples are **adversarially** chosen. The goal is to produce a small set $L= \{l_1,\dots,l_m \} $ of parameters, where $m =O(\frac{1}{\alpha})$, such that at least one parameter in this set is sufficiently close to the true parameter $l^*$.
>
> In contrast, our focus is on PAC learning over a **general** hypothesis class, and not just linear separators. Moreover, unlike list-decodable regression, we do not require the learnt hypothesis to belong to the hypothesis class itself. In fact, for multi-class PAC learning, it has been shown [1] that PAC learning is not possible by returning hypotheses from the underlying hypothesis class.
>
> Additionally, while the presence of corrupted samples in list-decodable regression is the main technical challenge, the main complexity in our setting is through the complexity of the hypothesis class itself. That is, because we do not only consider the class of linear functions,  but arbitrary function classes, it may be so that regression without lists is information-theoretically not possible at all, even in the absence of adversarial corruptions! (see Example 1). Finally, the number of labels ($k$) that the learner can output is a fixed parameter in our model, whereas in list decodable regression, the number of hypothesis is a polynomial function of $1/ \alpha$, where $\alpha$ is the fraction of uncorrupted data.
>
> **References**:
>
> [1] Amit Daniely and Shai Shalev-Shwartz. Optimal learners for multiclass problems.

---

### Official Review · Reviewer_M6FU · 2024-11-10
**Review of the paper "A Characterization of List Regression"**

**Rating:** 7
**Confidence:** 3

**Review:**

This paper studies the problem of PAC list regression. For each of the agnostic and realizable settings, the authors propose a family of dimensions at different scales for hypothesis classes and establish upper and lower bounds on the PAC sample complexity in terms of those dimensions, which shows that a hypothesis class is list regression PAC learnable iff the dimension is finite at all scales. Examples are presented to demonstrate the separation of the proposed dimensions.

The problem studied is fundamental in learning theory. The contribution on PAC learnability and sample complexity is original and significant. I appreciate the idea of reducing to multiclass learning via discretization used in Algorithm 1. The writing of the paper is clear with enough motivation and introduction to both the problem and the analysis.

However, the upper bounds and lower bounds of the sample complexity are valid in different settings and therefore cannot be compared at a fixed scale $\gamma$. In agnostic setting, the sample complexity upper bound in Theorem 12 requires $\epsilon> k^{O(k)}\gamma$ while the lower bound in Theorem 26 requires $\epsilon\le \gamma/8(k+1)$. For fixed $\gamma$, the two requirements cannot be satisfied simultaneously even if $k$ is viewed as a constant. For $\epsilon\le \gamma/8(k+1)$, the term $k^{O(k)}\gamma$ in the derived error rate upper bound is lower bounded by $k^{O(k)}\epsilon>\epsilon$ for $k>1$, which makes it impossible to upper bound the sample complexity at $\epsilon$ using the error rate upper bound.
Similarly in Theorem 40 and 43, the upper bound requires $\gamma=\Theta(\epsilon)$ while the lower bound requires $\gamma=\Theta(\sqrt{\epsilon})$.
I understand that the claim on the learnability in Theorem 1 and 2 is correct, but the claim that the dependence of the sample complexity on the dimension is optimal needs more explanation as dimensions under different $\gamma$ appear in the upper and lower bounds.

For the presentation of the paper, I believe it would be better if the authors can include more technical overview for the realizable setting in the first 12 pages. I also recommend the authors to include the reference to the proposed dimensions in Theorem 1 and 2 as well as Section 2.1 for readers convenience and for a comparison to the existing dimensions. Besides, I would appreciate it if the authors could comment on the possibility or difficulty in improving the dependence on list size $k$ under the current analysis.

Finally, there are some typos.

(1) In page 2, the "an" in the sentence "only an $\alpha$ fraction of the data is clean an the rest may be arbitrarily corrupted" should be "and".

(2) In Def. 4, $h(x_i)$ should be $h_b(x_i)$.

(3) In Def. 7 and 38, the outdeg should be defined to be the cardinality of the set given.

(4) In line 8 of Algorithm 1, $h$ should be used instead of $h^\star$ when taking $\inf_{h\in \mathcal{H}}$.

**Paper Award:**

No

---

> ### Author Response · Authors · 2024-11-25
> **Response to review**
>
> We sincerely appreciate your careful review, and we are glad that you like our paper! Thanks a lot for pointing out the typos as well--we will fix those in the revision. With regards to your other points:
>
> Indeed, we acknowledge your observation that our sample complexity bounds are not exactly optimal with respect to having the same scales in the complexity parameters in the upper and lower bounds. We will update our manuscript in the next version to make this clear.  Our intention for mentioning optimality was simply to highlight that the asymptotic dependence on the scale-sensitive dimension was the same (linear, upto log factors) in the upper and lower bounds. However, we will clarify that the scales in the dimensions are different.
>
> We would like to however mention that we have been able to significantly improve our upper bound for agnostic regression (Theorem 12), and improve the $k^{O(k)}\cdot \gamma$ term to simply $O(k)\cdot \gamma$ (which we believe, should be the right bound). In particular, we are able to make tighter the inequalities labeled (b) and (c) in the proof of Lemma 22 in A.4 via more careful counting. Thus, with our improved upper bound, for a fixed error $\epsilon$, the difference in scales in the fat-shattering dimension for the upper and lower bound is now simply a constant factor.
>
> We would also like to note that such a constant factor difference in the scales also appears in the original $k=1$ analyses: for agnostic regression with the absolute loss, the best-known upper bound and lower bound differ in the scales by a factor of 2 [Theorems 21 and 26, BL98], and there is also a constant factor gap in the scales for the squared loss [ABDCBH97, Theorems 4.1 and 4.2].
>
> Even in the case of realizable regression, in the work of [AHK+23], we can see in Theorem 2 in Appendix C, that the lower bound and upper bound for the **cutoff** PAC sample complexity differ in the scales of the $\gamma$ OIG dimension by a factor of 2 (it is because of this constant factor difference that they refer to their algorithm as **almost** optimal). However, when one translates from the cutoff PAC sample complexity to the (normal) PAC sample complexity using Markov's inequality (Lemma 1 in AHK+23], the gap widens further, and one needs $\gamma=\Theta(\epsilon)$ for the upper bound, and $\gamma=\Theta(\sqrt{\epsilon})$ for the lower bound. This square root gap shows up for the same reason in the list setting as well.
>
> While we have already been able to improve the list size dependence in the agnostic upper bound, we would also like to provide a couple lines of intuition as to why the $k^k$ factor shows up (in the denominator) in the lower bound. For the agnostic lower bound, we can observe that if we were to work with the strong-fat-shattering dimension instead of fat-shattering dimension, then we get an exact $\Omega(strong-fat-dimension)$ lower bound (Lemma 28). The $k^k$ factor arises in translating from strong-fat to fat-shattering dimension, which requires us to go through the $k$-ary packing number analysis.
>
> Lastly, thank you for the pointer about including a brief technical discussion about the techniques used for the realizable setting. We will address this in the revision, and also include references to the definitions. The only reason we did not define these dimensions in the results section was because we felt that the definitions were a little technically verbose, and we did not want to distract from the main results themselves, which was the characterization of learnability in terms of the dimensions. Nevertheless, to give the reader a sense of our constructions, we can possibly try and motivate the construction of the k-fat-shattering dimension from the fat-shattering dimension right in the results section itself, since it seems like the easier of the two.
>
> **References**
>
> [ABDCBH97] Noga Alon, Shai Ben-David, Nicol`o Cesa-Bianchi, and David Haussler. Scale-sensitive
> dimensions, uniform convergence, and learnability.
>
> [AHK+23] Idan Attias, Steve Hanneke, Alkis Kalavasis, Amin Karbasi, and Grigoris Velegkas. Optimal
> learners for realizable regression: Pac learning and online learning.
>
> [BL98] Peter L Bartlett and Philip M Long. Prediction, learning, uniform convergence, and
> scale-sensitive dimensions.

---

### Official Review · Reviewer_G8M5 · 2024-11-11
**Review: A Characterization of List Regression**

**Rating:** 7
**Confidence:** 4

**Review:**

The paper considered the problem of k-list regression which is basically a problem of online regression (in this case with absolute loss) where the learner is allowed to make a list of k predictions and the one with the smallest loss is in effect considered. This problem can be framed as a general statistical learning problem with specific structure for the loss considered. The main challenge is to use this structure (of min over the list of losses) to obtain characterization of learnability. The paper provides characterization of learnability for both agnostic and realizable setting. The k = 1 case is the classic regression with absolute loss case and characterization for both agnostic and realizable setting are known for this case in prior literature.

The paper seems theoretically sound. I am not jumping up and down because I find the problem somewhat niche and not one that many people care about. That said, the paper is well written technically and the results are strong.

**Paper Award:**

No

---

> ### Author Response · Authors · 2024-11-25
> **Response to review**
>
> We thank you for your positive review, and appreciate your time in reviewing our paper.

---

### Author Rebuttal · Authors · 2024-11-25

We thank all the reviewers for their helpful comments and time in reviewing our paper. We have addressed each review individually as a comment below.

---

### Meta-Review · Area_Chair_mJWo · 2024-12-13

**Recommendation:** Accept
**Confidence:** 5

**Metareview:**

The reviewers agree that this work makes a solid contribution.
They also express some reservations about the structure and presentation of the paper; the authors have responded saying they will work to improve this aspect in the final version.

I find the paper interesting.  The formulation of the problem is novel, and seems to me to be the natural extension of regression to the list prediction setting (i.e., the loss of the list prediction is the minimum of the individual losses).  Before reading their formulation, the appropriate definition was not obvious to me, so this itself is a nice contribution (and will likely inform future works on other settings involving a non-binary loss function).  They completely characterize learnability, in both the realizable and agnostic settings, making an important contribution to the developing literatures on both regression and list learning.  The techniques and arguments contain several nontrivial innovations. Overall, I find this work worthy of publication at ALT.

**Paper Award:**

No